# VoMP: Predicting Volumetric Mechanical Property Fields

**Rishit Dagli**[1,2]  **Donglai Xiang**[1]  **Vismay Modi**[1]  **Charles Loop**[1]  **Clement Fuji Tsang**[1]
**Anka He Chen**[1]  **Anita Hu**[1]  **Gavriel State**[1]  **David I.W. Levin**[1,2]  **Maria Shugrina**[1]

[1]NVIDIA  [2]University of Toronto
`https://research.nvidia.com/labs/sil/projects/vomp`

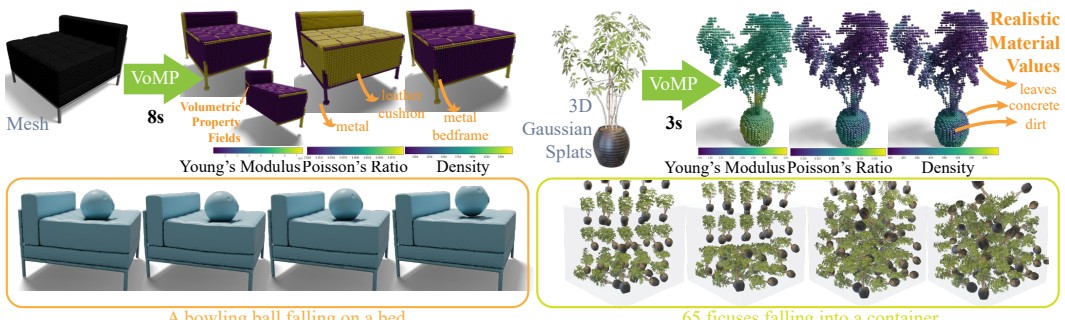

Figure 1: **VoMP** predicts physically accurate volumetric mechanical property fields across 3D representations in just a few seconds (top), enabling their use in realistic deformable simulations (bottom).

## ABSTRACT

Physical simulation relies on spatially-varying mechanical properties, often laboriously hand-crafted. VoMP is a feed-forward method trained to predict Young's modulus ($E$), Poisson's ratio ($\nu$), and density ($\rho$) throughout *the volume* of 3D objects, in any representation that can be rendered and voxelized. VoMP aggregates per-voxel multi-view features and passes them to our trained Geometry Transformer to predict per-voxel material latent codes. These latents reside on a space of physically plausible materials, which we learn from a real-world dataset, guaranteeing the validity of decoded per-voxel materials. To obtain object-level training data, we propose an annotation pipeline combining knowledge from segmented 3D datasets, material databases, and a vision-language model, along with a new benchmark. Experiments show that VoMP estimates accurate volumetric properties, far outperforming prior art in accuracy and speed.

## 1 INTRODUCTION

Accurate physics simulation is a critical part of modern design and engineering, for example, in workflows like creating Digital Twins (virtual replicas of real systems) (Grieves & Vickers, 2017), Real-2-Sim (generating digital simulation from the real world) (NVIDIA, 2019), and Sim-2-Real (transferring policies trained in simulation to real-world deployment) (Rudin et al., 2021). However, setting up reliable simulations remains labor-intensive, partially due to the necessity to provide accurate mechanical properties *throughout the volume* of every object, namely the spatially-varying Young's Modulus ($E$), Poisson's ratio ($\nu$), and density ($\rho$). Common 3D capture methods (Kerbl et al., 2023) and 3D repositories (Deitke et al., 2023) rarely contain such annotations, forcing artists and engineers to guess or copy-paste coarse material presets in a subjective, time-consuming process. We focus on automatic prediction of these parameters, addressing important limitations of prior art.

We propose VoMP, *the first feed-forward model trained to estimate simulation-ready mechanical property fields* ($E, \nu, \rho$) *within the volume of 3D objects across representations*. Rather than specializing on inputs like Gaussian Splats (Shuai et al., 2025; Xie et al., 2024), our method works

for any geometry that can be voxelized and rendered from turnaround views, including meshes, Gaussian Splats, NeRFs and SDFs (Fig. 1). Unlike virtually all prior works, VoMP is fully feed-forward, requiring no per-object optimization of feature fields (Zhai et al., 2024; Shuai et al., 2025) or run-time aggregation of Vision-Language Model (VLM) (Lin et al., 2025a) or Video Model (Lin et al., 2025b) supervision. Uniquely among others, VoMP outputs true mechanical properties (a.k.a. material parameters), like those measured in the real world. Many existing pipelines target fast, approximate simulators, resulting in simulator-specific parameters (Zhang et al., 2025; Huang et al., 2024b) that may not transfer reliably across frameworks (Fig. 2), whereas our result is directly compatible with any accurate simulator. Finally, unlike prior art, our method is designed to assign materials throughout the object volume, which is critical for simulation fidelity.

To enable learning physically valid mechanical properties, we first train a latent space on a database of real-world values $(E, \nu, \rho)$ using a variational auto-encoder MatVAE (§3). To predict mechanical property fields for 3D objects, our method first voxelizes the input geometry and aggregates multi-view image features across the voxels (§4.1). This process accepts many representations[1] and is fast, unlike optimization used in concurrent work (Le et al., 2025). We pass the voxel features through the Geometry Transformer (§4.2), trained to output per-voxel material latents. The MatVAE latent space decouples learning material assignments for objects from learning what materials are valid, ensuring that the final volumetric properties $(E, \nu, \rho)$ decoded by MatVAE are physically valid, even in the case of interpolation. To create material property fields for training, we propose a pipeline (§5) combining the knowledge from part-segmented 3D assets, material databases, visual textures, and a VLM. Our experiments (§6) show that VoMP estimates simulation-ready spatially-varying mechanical properties across a range of object classes and representations, resulting in realistic elastodynamic simulations. We evaluate our method on an existing mass prediction benchmark and contribute a new material estimation benchmark (§6.3), consistently outperforming prior art (Shuai et al., 2025; Lin et al., 2025a; Zhai et al., 2024). In summary, our contributions are:

- The first (to our knowledge) method to estimate object mechanical material property fields that *(1)* is a trained feed-forward model with minimal preprocessing, *(2)* generalizes across 3D representations, *(3)* predicts physically valid properties that can be used with an accurate simulator, and *(4)* predicts mechanical properties *within the volume* of objects (§4).

- The first (to our knowledge) mechanical properties latent space (§3).

- An automatic data annotation pipeline and a new benchmark for volumetric physics materials (§5).

- Thorough evaluation through high-fidelity simulations and quantitative metrics on existing and new benchmarks, significantly outperforming the prior art (§6).

## 2  RELATED WORK

### 2.1  BACKGROUND

All algorithms for continuum-based simulation of solids and liquids require material models as input. The material, or constitutive, model is the function that determines the force response of a class of materials (e.g., rubbers, snow, water) to internal strains and strain rates. To produce the correct constitutive behavior for a given material, the model requires an accurate set of corresponding material parameters for every point in the simulated volume. For locally isotropic material models, Young's modulus ($E$, in the 1D linear regime, the proportionality constant between stress and strain), Poisson's ratio ($\nu$, the negative ratio of transverse to axial strain under uniaxial loading) and density ($\rho$, unit mass per volume) are ubiquitous. Given an accurate and valid triplet ($E$, $\nu$, $\rho$) along with a reasonable material model, a consistent numerical simulation can produce accurate predictions of an object's behavior under load. Measured, real-world parameters



Figure 2: **Simulator differences** when dropping a solid sphere with $(E, \nu, \rho) = (10^4 Pa, 0.3, 10^3 \,\text{kg/m}^3)$ with XPBD (Macklin et al., 2016) and MPM (Sulsky et al., 1994) vs. more accurate FEM.

---

[1] We describe available methods for meshes, SDFs, and NeRFs, and present a method for Splats in §6.1.

are portable to any consistent simulation algorithm (we use high-resolution Finite Element Methods). Further, they are portable across any material model that relies on density, Young's modulus and Poisson's ratio, or derived quantities, such as shear or bulk modulus (e.g., Neo-Hookean, St. Venant–Kirchhoff, As-Rigid-As-Possible, Co-Rotated Elastic, Mooney–Rivlin, and Ogden models). On the other hand, many physics simulation algorithms are not implemented or applied in a consistent fashion, favoring speed over accuracy (Macklin et al., 2016; Sulsky et al., 1994). In these cases, material parameters must be modified to avoid inaccurate behavior (Fig. 2).

## 2.2 INFERRING MECHANICAL PROPERTIES OF STATIC OBJECTS

Our goal is to predict volumetric mechanical properties given only shape and appearance, a challenging inverse problem, which research suggests humans learn good intuition about (Adelson, 2001; Fleming, 2014; Fleming et al., 2013; Sharan et al., 2009). However, progress in learning-based approaches has been hampered by limited data. Existing datasets are small (Gao et al., 2022; Downs et al., 2022; Chen et al., 2025c), contain noisy labels (Lin et al., 2018), use simulator-specific parameters (Mishra, 2024; Xie et al., 2025; Belikov et al., 2015), provide only coarse annotations (Ahmed et al., 2025; Slim et al., 2023; Li et al., 2022) or are biased towards rigid or man-made objects (Cao et al., 2025). Worse, data collection is difficult, relying on rigorous physical experiments (ASTM Committee D20, 2022; ASTM Committee E28, 2024; Pai, 2000), and even then lacking spatial material fields (Loveday et al., 2004) due to digitization and annotation challenges.

As a result, works that infer physical properties from appearance often leverage knowledge from large pre-trained models. NeRF2Physics (Zhai et al., 2024) and PUGS (Shuai et al., 2025) optimize language-embedded feature fields for a NeRF (Mildenhall et al., 2020) or 3D Gaussians (Kerbl et al., 2023), respectively, to predict coarse stiffness categories and density, but require per-object optimization and are limited in their ability to predict values inside objects due to the lack of meaningful features inside NeRFs or splats. Many approaches distill signals from a Video Generation Model and optimize physics parameters by backpropagating through fast, approximate physics simulators, resulting in a slow optimization process, yielding materials deviating from real-world values and overfit to a specific simulation setup (Zhang et al., 2025; Huang et al., 2024b; Liu et al., 2025; Cleac'h et al., 2023; Liu et al., 2024a; Lin et al., 2025b) (§2.1). Many methods are also tailored to a specific 3D representation or real-time simulation implementations, such as Splats (Xie et al., 2024) or explicit Material Point Methods (Sulsky et al., 1994; Le et al., 2025), or work with coarse material categories (Fischer et al., 2024; Hsu et al., 2024; Lin et al., 2025a; Xia et al., 2025) that must be manually mapped to simulation parameters. Instead, we aim to augment objects across 3D representations with fine-grained spatially-varying mechanical properties that are physically accurate and compatible across accurate simulators. Like our method, many techniques leverage vision-language (VLM) models. PhysGen (Liu et al., 2024b) and PhysGen3D (Chen et al., 2025a) use a VLM to infer mass, elasticity, and friction for segmented parts of a single image. Phys4DGen (Lin et al., 2025a) uses a VLM to annotate parts of a 3D model with coarse material labels, which are then mapped to physical parameters, a baseline used in our evaluation. Most works above rely on aggregation of large model outputs for every input shape, which can be brittle and time-consuming at run-time, and can only leverage external segmentation. Instead, our method uses a VLM paired with other data sources to annotate a *training dataset* for a feed-forward model leveraging 3D data to annotate and learn internal material composition.

Like our method, SOPHY (Cao & Kalogerakis, 2025), PhysX-3D (Cao et al., 2025), PhysSplat (Zhao et al., 2024a;b) (a.k.a. SimAnything) and the concurrent Pixie (Le et al., 2025) leverage pre-trained models and 3D data to annotate a *training* dataset with physical materials. PhysSplat trains a network to predict spatially-varying simulator-specific material offset weights for MPM by using outputs from video distillation (Liu et al., 2024a), not focusing on material accuracy. SOPHY and PhysX-3D are 3D generative models, designed to generate new shapes augmented with physical attributes, and cannot augment existing assets, which is our goal. Still, we detail similar aspects of these works. Like these works, our method uses a VLM to annotate 3D objects with Young's Modulus, Poisson's ratio, and density, but we do not rely on the human-in-the-loop and instead leverage multiple data sources, not just VLM knowledge, to ensure more accurate physical properties. As a baseline, SOPHY does implement a material decoder, but it has not been made available, and only considers object surface, while we aim to estimate volumetric properties. Like our method, PhysX-3D adopts the structural latent space of TRELLIS, but trains a joint generative model over these and learned shape-aware physical properties latents in order to generate physics-augmented shapes from

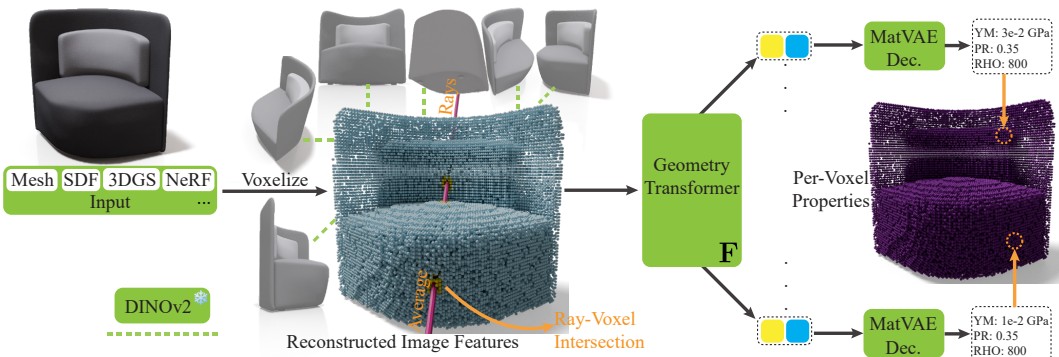

Figure 3: **VoMP Overview.** For any input geometry, we aggregate multi-view DINOv2 features across its volumetric voxelization (§4.1). A trained GeometryTransformer (§4.2) predicts per-voxel material latents, decoded by MatVAE (§3) into mechanical properties ($E$, $\nu$, $\rho$).

scratch. In contrast, we treat material prediction as deterministic inference for simplicity, and further adjust the TRELLIS pipeline to facilitate accurate material prediction inside the object. Pixie (Le et al., 2025), a concurrent work and the only other feed-forward approach, is trained on semantically-segmented objects and uses points from filtering NeRF densities. Thus, Pixie is trained on segments biased toward the surface, as we show in Fig. 15, while we demonstrate being able to estimate volumetric properties with internal structures. Furthermore, unlike Pixie, we specifically focus on estimating physically plausible material properties, such as those measured in the real world.

## 3  MECHANICAL PROPERTIES LATENT SPACE

To learn a latent space of valid Young's modulus, Poisson's ratio, and density triplets ($E$, $\nu$, $\rho$) (§2.1), we propose MatVAE, a variational autoencoder (VAE) trained on a dataset of real-world values $\{m_i := (E_i,\ \nu_i,\ \rho_i)\}$ (§5.1). The model's objective is to map these triplets $m$ into a 2-dimensional latent space, $z \in \mathbb{R}^2$, from which they can be accurately reconstructed. While this offers only minor compression ($\mathbb{R}^3 \to \mathbb{R}^2$), this latent 2D space of material properties is now easy to visualize, sample, and interpolate within, and results in consistent distances between material triplets with disparate units (Fig. 7,§6.4). MatVAE acts like a continuous tokenizer that allows us to always ensure VoMP output properties that fall inside the range of some materials.

We build on VAE (Kingma & Welling, 2022), with the reconstruction component of the loss defined as mean-squared error between the input ($E_i, \nu_i, \rho_i$) and reconstructed material values ($\hat{E}_i, \hat{\nu}_i, \hat{\rho}_i$):

$$\mathcal{L}_{\text{Recon}} = \frac{1}{N} \sum_{i=1}^{N} \left\| ((E_i, \nu_i, \rho_i)^N)^\mathsf{T} - ((\hat{E}_i, \hat{\nu}_i, \hat{\rho}_i)^N)^\mathsf{T} \right\|_2^2, \tag{1}$$

where $\mathsf{T}$ denotes transpose and $N$ per-property normalization, where $E$ and $\rho$ are first log-transformed ($\log_{10}(E), \log_{10}(\rho)$), then normalized to $[0, 1]$, while $\nu$ is directly normalized to $[0, 1]$. We find other normalization schemes without log-transform or standard $z$-score normalization induce a heavy-tailed feature distribution, which is poorly conditioned for learning (§C).

We make several modifications over standard VAE. *First*, to capture a complex posterior beyond a simple Gaussian, the encoder's output is transformed by a (radial) Normalizing Flow (Rezende & Mohamed, 2015), giving us a more flexible variational distribution $q_\phi(z|m)$ since we observe heavy-tailed distribution for Young's Modulus and Density while Poisson's Ratio concentrates near the boundaries after normalization. *Second*, we decompose the KL-divergence term of the ELBO following (Chen et al., 2018). This allows us to directly penalize the total correlation $\text{TC}(z) = \text{KL}(\bar{q}_\phi(z) \| \prod_j \bar{q}_\phi(z_j))$ where $\bar{q}_\phi(z)$ is the aggregated posterior, $z_j$ is the $j \in \{1, 2\}$-th coordinate of the latent vector $z$. Penalizing TC allowed us to reduce the high dependence between latent coordinates which caused MatVAE to encode density in both dimensions. *Third,* we observe imbalanced reconstruction, *i.e.* the latent space collapses to one property, giving us low reconstruction errors for one property and high reconstruction error for others (§C). Thus, to ensure the 2 latent dimensions are actively utilized, we introduce a capacity constraint ($\delta \times z_{\text{dim}}$) based

on (Higgins et al., 2017), resulting in the following final objective:

$$\mathcal{L}_{\text{MatVAE}} = \underbrace{\mathcal{L}_{\text{Recon}}}_{\text{MSE}} + \overbrace{\underbrace{\gamma \cdot \text{MI}(z)}_{\text{Mutual Information}} + \underbrace{\beta \cdot \text{TC}(z)}_{\text{Total Correlation}}}^{\text{Latent Space Regularization}} + \alpha \cdot \sum_{j=1}^{d} \underbrace{\max\big(\delta, \ \text{KL}(q_\phi(z_j) \,\|\, p(z_j))\big)}_{\text{Dimension-wise KL}}, \tag{2}$$

where we set $\gamma, \beta, \alpha = (1.0, 2.0, 1.0)$, with a free nats constraint $\delta = 0.1$. See §F.1 for more details.

## 4 PREDICTING MECHANICAL PROPERTY FIELDS

To predict volumetric mechanical properties across 3D representation, VoMP first aggregates volumetric features for the input geometry (§4.1), which are then processed by a trained feed-forward transformer model (§4.2) that learns in the latent space of MatVAE (§3). See §2.2.

### 4.1 AGGREGATING FEATURES

Our method accepts any 3D representation that can be voxelized and rendered from multiple views. Following recent works (Wang et al., 2023; Dutt et al., 2024; Xiang et al., 2025), we compute rich DINOv2 (Oquab et al., 2024) image features across 3D views and lift them to 3D by projecting each voxel center into every view using the camera parameters to retrieve the corresponding image features. The retrieved image features are then averaged to obtain a feature for every voxel. A critical difference with these prior works is that we also voxelize and process the interior of the objects and not just their surface, which allows us to learn and predict material properties *inside* the objects (See §6.1 for voxelization schemes and see §F.3 for details on voxelization for training). Let's denote all active voxel center positions in a 3D grid of size $N^3$ as $\{\mathbf{p}_i\}_{i=1}^{L}$ where $L$ denotes the number of voxels, $\mathbf{p}_i \in \mathbb{R}^3$ denotes the voxel center, and $\Pi_j : \mathbb{R}^3 \to [-1, 1]^2$ the camera projection for view $j \in J$ where $J$ is the set of rendered views. Let the DINOv2 patch-token map be $T_j \in \mathbb{R}^{1024 \times n \times n}$ which is bilinearly sampled to get a feature map $\mathcal{F}_j : [-1, 1]^2 \to \mathbb{R}^{1024}$. Then for each voxel $i \in \{1, 2, \ldots, L\}$, we obtain a feature $\mathbf{f}_i$:

$$\mathbf{f}_i = \text{Average}(\mathcal{C}_i = \{\mathcal{F}_j\big(\Pi_j(\mathbf{p}_i)\big) \mid j \in J\}) \in \mathbb{R}^{1024} \tag{3}$$

This propagates multi-view information to the voxels in the interior of the object, encoding useful information that our model learns to process to predict internal material composition.

### 4.2 GEOMETRY TRANSFORMER

The main component of VoMP is a Transformer $\mathbf{F}$ that maps voxelized image features to our trained material latent representation. The backbone of our model follows TRELLIS (Xiang et al., 2025) encoder/decoder, and the backbone layers of our model are initialized with TRELLIS weights. The encoder processes a variable-length set of active voxels, represented by their positions and features $\mathbf{X} = \{(\mathbf{p}_i, \mathbf{f}_i)\}_{i=1}^{L}$. To make this data suitable for a Transformer, we first serialize the voxel features into a sequence and then inject spatial awareness by adding sinusoidal positional encodings derived from each voxel's 3D coordinates. Similar to TRELLIS and state-of-the-art 3D Transformers, we adopt a 3D shifted window attention mechanism (Liu et al., 2021; Yang et al., 2025). Contrary to TRELLIS (Xiang et al., 2025), to handle assets of various sizes, we define a maximum sequence length of $L_N$. For assets with fewer voxels $L \leq L_N$, we use the complete set. However, for larger assets where $L > L_N$, we use a stochastic sampling strategy, selecting a random subset of $L_N$ voxels at the start of each training epoch. This dynamic resampling ensures the model is exposed to different parts of the asset over epochs and have a larger number of "effective" max voxels.

For each training asset, we first define $\mathcal{S}$ as the set of voxel indices to be processed in the current iteration. The corresponding sequence of image features $\mathbf{X}_{\mathcal{S}}$ obtained from voxel features (§4.1), is passed to $\mathbf{F}$. The resulting latent representation is then fed into the frozen decoder of pre-trained MatVAE to predict material properties. The MatVAE is run $L$ times *i.e.* once per voxel, which gives us material triplets $(E, \nu, \rho)$ *for each voxel*. We train this transformer with the mean squared error between the predicted materials and the ground truth materials, averaged over all voxels in the set $\mathcal{S}$,

$$\mathcal{L}_{\mathbf{F}} = \frac{1}{|\mathcal{S}|} \sum_{i \in \mathcal{S}} \|\mu_\theta(\mathbf{F}(\mathbf{X}_{\mathcal{S}})_i) - ((E_i, \nu_i, \rho_i)^N)^\mathsf{T}\|_2^2, \tag{4}$$

where $\mu_\theta(\cdot)$ denotes the output of the frozen MatVAE decoder, $((E_i, \nu_i, \rho_i)^N)^\mathsf{T}$ is the ground truth material vector for voxel $i$, and $\mathbf{F}(\mathbf{X}_\mathcal{S})_i$ is the latent representation for voxel $i$.

To transfer voxel materials back to the original representation (*i.e.* splat means, tets for FEM simulation, quadrature points for simulation, etc.), we use nearest neighbour interpolation as outlined in §G.1. The per-voxel latents are passed into the decoder model of MatVAE (§3), which yields per-voxel material triplets, as shown in §2.2.

## 5 Training Data Generation

### 5.1 Material Triplets Dataset (MTD)

To train MatVAE (§3), we collect Material Triplet Dataset(MTD), containing 100,562 triplets $(E, \nu, \rho)$ for real-world materials. We first collect a dataset of measured material properties from multiple online databases (MatWeb, LLC, 2025; Wikipedia contributors, 2024a;b;c; The Engineering Toolbox, 2024; Department of Engineering, University of Cambridge, 2011), containing values obtained experimentally, typically with valid *ranges* for all three properties for all materials. We sample numeric triplets from each material, with the number of samples proportional to the range size. Finally, we filter out duplicates resulting from overlapping ranges for some materials.

### 5.2 Geometry with Volumetric Materials (GVM) Dataset

To train Geometry Transformer (§4), we develop an automatic annotation pipeline to overcome the limited availability of detailed volumetric material datasets (§2.2). Like prior works (Lin et al., 2025a; Cao & Kalogerakis, 2025; Le et al., 2025), we leverage a pre-trained VLM, but overcome its limitations by introducing *additional sources of knowledge* present in our 3D dataset and the MTD(§5.1). We collect high-quality 3D meshes from (NVIDIA Corporation, 2025a;c; NVIDIA Developer, 2025; NVIDIA Corporation, 2025d), containing 1624 part-

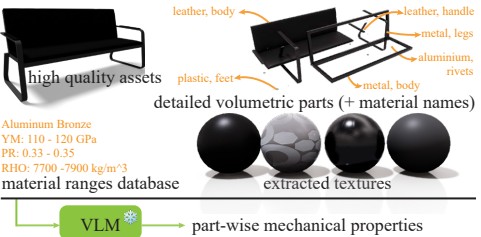

Figure 4: **Training Data** annotation leverages accurate 3D data labels together with a VLM.

segmented 3D models, with a total of 8089 parts, and treat each part as having isotropic material. Each part contains an English material name and its own realistic PBR texture, which can be used as additional cues to the VLM. For each part in each object, we pass the following information to the VLM: rendering of the full object, detail rendering of the part's visual material mapped onto a sphere (showing visual aspects that tend to correlate with material composition), the material names, and the ranges of three closest real-world materials in the MTD (§5.1) based on the material names (See Fig.4, detailed prompt in Fig. 23). The vision-language model then outputs material triplets for each part, and we map to all volumetric voxels within it, resulting in a total of 37M voxels annotated with $(E, \nu, \rho)$. By guiding VLM with real-world material values and extra clues, we avoid inaccuracies and implausible material values. See additional details in §6.1, §E.

## 6 Experiments and Results

We evaluate VoMP end-to-end, showing diverse realistic simulations in §6.2. Quantitative results are presented in §6.3, with MatVAE evaluated separately in §6.4. See ▶ video and §A for many additional results, §B for extra comparisons with concurrent work and §C for ablations.

### 6.1 Implementation Details

**Voxelization:** For voxelizing 3D Gaussian splats (Kerbl et al., 2023), we present *a new voxelizer*, that works in three phases: (1) 3D Gaussians are voxelized over a 3D grid as solid ellipsoids defined by the 99th percentile iso-surface, (2) this set of voxels is rendered from several dozen viewpoints sampled over a sphere to form depth maps, (3) these depthmaps are used to carve away empty space around the exterior of the object, but leaving unseen *interior* voxels to form a solid approximation of the object. We then sample this solid at jittered sample points on a regular grid. We employ

A stack of 3 oranges falling down | 18 dog toys falling through a pachinko

Figure 5: **Simulation-ready physics materials of VoMP** enable realistic simulations for meshes and splats.

octrees as acceleration structures and GPU implementations for both phases. Our test objects can be voxelized in 31 ms (see Tb. 1). To voxelize meshes and SDFs we use standard methods (see §F.3).

**Data and Training:** For material annotation, we experimentally choose Qwen 2.5 VL-72B VLM (Bai et al., 2023; 2025). We partition our MTD and GVM datasets (§5) into 80-10-10 train, validation, and test sets. See §E for data details. For rendering we use Omniverse (NVIDIA Corporation, 2025b) and Blender (Blender Online Community, 2021), and for DINOv2 we use an optimized implementation (NVIDIA, 2025). During training and testing, we set the maximum number of non-empty voxels per object $L_N = 32768$ (sampled stochastically, §4.2), and sparse data structures for efficiency. See §F for more details. All experiments were performed on a machine with four 80GB A100 GPUs, where training took about 12 hours for MatVAE and 5 days for the Transformer.

**Simulation:** We used FEM simulator for meshes and sparse Simplicits (Modi et al., 2024; Fuji Tsang et al.) for our large-scale simulations combining splats and meshes. Details in §G.

## 6.2 END-TO-END QUALITATIVE EVALUATION

We qualitatively evaluate VoMP by using it to annotate volumetric mechanical fields for several meshes and 3D Gaussian Splats, and running physics simulation with these exact spatially varying $(E, \nu, \rho)$ values, resulting in realistic simulations without any hand-tweaks (Fig. 5, Fig. 8, ▶: 0:36). We also show that our approach can work across more representations, including meshes, 3D Gaussian Splats, SDF, and NeRFs ( Fig. 8a, with additional results in §A.2.

## 6.3 QUANTITATIVE EVALUATION

**Datasets and Metrics:** The 10% hold-out test set of GVM (§5) consists of 166 high-quality 3D objects with per-voxel mechanical properties for a total of 4.9 million point annotations, significantly larger than previous works, e.g. 31 points across 11 objects (Zhai et al., 2024). We contribute this as *a new benchmark* and use it for evaluation against baselines. We measure standard metrics, Average Log Displacement Error (ALDE), Average Displacement Error (ADE), Average Log Relative Error (ALRE), and Average Relative Error (ARE) for each mechanical property, further detailed in §D.1. We provide additional intuition for interpreting these errors through targeted simulations in §D.4.

Table 1: **Wall-clock** comparisons and breakdown.

| Method | Time (s) |
|---|---|
| NeRF2Physics | 1454.55 ($\pm$1118) |
| PUGS | 1058.33 ($\pm$6.94) |
| Pixie | 201.63 ($\pm$27.74) |
| Phys4DGen* | 51.65 ($\pm$4.07) |
| Ours | **3.59** ($\pm$1.36) |
| Rendering | 2.11 ($\pm$0.0540) |
| Voxelization | 0.03 ($\pm$0.0016) |
| DINO-v2 Computation | 0.86 ($\pm$0.0020) |
| DINO-v2 Reconstruction | 0.58 ($\pm$0.0053) |
| Geometry Transformer | 0.0082 ($\pm$0.0063) |
| MatVAE | 0.00032 ($\pm$0.00026) |

**Baselines:** We compare against prior art NeRF2Physics (Zhai et al., 2024) and PUGS (Shuai et al., 2025), where we look up material properties at the voxel locations (with proper scaling) using their optimized representations. Note that these techniques do not output Poisson's ratio. Phys4DGen (Lin et al., 2025a) is an important baseline, aggregating VLM prediction directly, but does not provide code. We used our best effort to replicate their method and used prompts provided by the authors, designating this implementation Phys4DGen*. More baseline details in §F.5. We also include early comparisons against concurrent (and as yet unpublished) Pixie (Le et al., 2025), with additional explorations in §B.

**Estimating Mechanical Properties:** Quantitative evaluation of material estimates $(E, \nu, \rho)$ of our method against prior art on our new detailed benchmark shows a *dramatic quality boost across all properties and metrics* (Fig. 6b). According to our explorations (§D.4), ALRE under 0.05 for $E$ and ARE under 0.15 for other properties result in similar simulations, suggesting that our materials will

Table 2: **Mechanical Property Estimates** of our method on the *publicly released dataset* are very close to the full dataset. Per-voxel error rate is first computed per object, then averaged across all objects in the test set to avoid weighing some objects more. Global voxel-level normalization yields similar results, see Supplement Tb. 3.

| Method | Young's Modulus Pa ($E$) | | Poisson's Ratio ($\nu$) | | Density $\frac{kg}{m^3}$ ($\rho$) | |
|---|---|---|---|---|---|---|
| | ALDE ($\downarrow$) | ALRE ($\downarrow$) | ADE ($\downarrow$) | ARE ($\downarrow$) | ADE ($\downarrow$) | ARE ($\downarrow$) |
| NeRF2Physics | 2.8000 ($\pm 1.05$) | 0.1346 ($\pm 0.05$) | - | - | 1432.0343 ($\pm 964.88$) | 1.0365 ($\pm 0.63$) |
| PUGS | 3.3942 ($\pm 1.72$) | 0.1688 ($\pm 0.10$) | - | - | 3568.2150 ($\pm 2839.13$) | 3.2429 ($\pm 3.56$) |
| Phys4DGen* | 4.8967 ($\pm 3.17$) | 0.2227 ($\pm 0.14$) | 0.0407 ($\pm 0.04$) | 0.1467 ($\pm 0.18$) | 1865.5673 ($\pm 2176.90$) | 1.4394 ($\pm 2.35$) |
| Ours | **0.3794** ($\pm 0.29$) | **0.0409** ($\pm 0.04$) | **0.0241** ($\pm 0.01$) | **0.0818** ($\pm 0.03$) | **142.7017** ($\pm 166.92$) | **0.0921** ($\pm 0.07$) |

lead to more faithful simulations than competitors when using an accurate simulator. Qualitatively (Fig. 6a), we observe that this performance difference may be due to baselines occasionally mislabeling segments (e.g. by Phys4DGen), due to noisy estimates (e.g. NeRF2Physics and PUGS), and less accurate values in the objects' interior due to the baselines' design.

We are unable to make the vegetation subset of our dataset publicly available. Thus, we compute the mechanical property estimations on the public version of the dataset in Tb. 2 and 3. We find that our results averaged over the public dataset are highly similar to the full dataset.

**Run-Time:** To show approximate speed difference, we report average material estimate speeds across 100 runs on objects with an average of 53.9K Gaussians for our method and the baselines in Tb. 1. To ensure fair compute between CPU and GPU heavy methods, we ran this experiment on a machine with only one A100 GPU and 64 CPUs. While we do not provide timing breakdown of the other methods, this result suggests a speed up of 5-100x achieved by our method, which is not surprising given that it is the only feed-forward model among previous work. Concurrent Pixie, which is also feed-forward, involves a heavier pre-processing step, including per-object optimization, affecting its end-to-end time. In the timing breakdown of our method, rendering and pre-processing take the most time, and could be further optimized.

**Mass Estimation:** Following NeRF2Physics (Zhai et al., 2024) and PUGS (Shuai et al., 2025), we also evaluate our dataset on the ABO-500 (Collins et al., 2022) object mass estimation benchmark, following the evaluation protocol of PUGS. We run our model to estimate density $\rho$ for upto 32678 voxels per object, then average these values and multiply by the known object volume to obtain mass. While this is only an imperfect proxy for measuring the accuracy of volumetric density $\rho$, it is a benchmark used by prior works, and we include it for completeness. We achieve better or on-par performance across most metrics (Fig. 6c), with qualitative results in §A.3.

**Validity:** To gauge how well different methods are at predicting realistic materials, such as those measured in the real world, which is our goal, we leverage our MTD dataset of real materials. First, we run all methods on GVM test set objects, and for each test voxel compute relative errors to the nearest possible material range from MTD (error is 0 for estimates within an existing material range). These errors are averaged across all the voxels and reported in Fig. 6d. We observe that our method, on average, outputs much more realistic materials, as it was explicitly designed to do so.

## 6.4 Reconstructing and Generating Materials with MatVAE

Given no prior works exploring a latent space of material triplets ($E$, $\nu$, $\rho$), we evaluate MatVAE on the MTD test set (§6.1), achieving low reconstruction errors in Fig. 7a (See Appendix D.1, Appendix D.4) for metrics). Further, in Fig. 7 we show the desirable properties of this learned latent space. In (a), samples throughout the 2D latent space map to real-world material ranges in MTD. In (b), we show that ($E$, $\nu$, $\rho$) values of real materials encoded to the latent space vary smoothly. Further, the latent space ensures valid interpolation points between materials (c), facilitating valid assignment from predicted voxel materials back to the original geometry. We include detailed ablations of MatVAE design (§C), and additional latent space explorations (§A.4).

## 7 Discussion

We introduce a representation-agnostic method that maps any 3D asset (mesh, SDF, Gaussian splat, or voxel grid) to a volumetric field of physically valid mechanical properties ($E, \nu, \rho$). We show

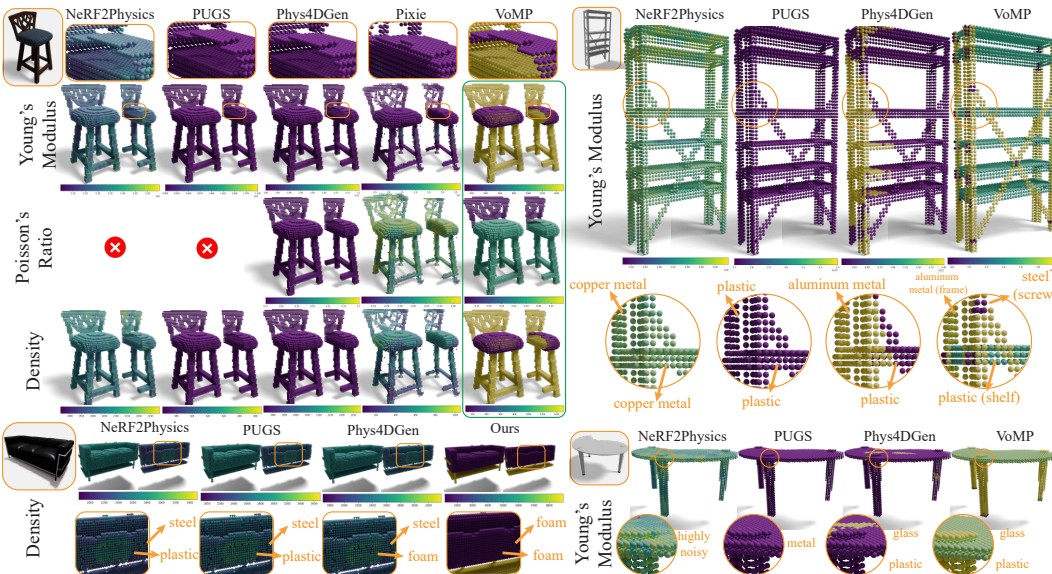

(a) **Qualitative Comparison:** We show that qualitiative VoMP tends to provide less noisy volumetric? values compared to the baselines. We show the color coded fields and slice planes through the fields.

| Method | Young's Modulus Pa ($E$) | | Poisson's Ratio ($\nu$) | | Density $\frac{kg}{m^3}$ ($\rho$) | |
|---|---|---|---|---|---|---|
| | ALDE ($\downarrow$) | ALRE ($\downarrow$) | ADE ($\downarrow$) | ARE ($\downarrow$) | ADE ($\downarrow$) | ARE ($\downarrow$) |
| NeRF2Physics | 2.8000 (±1.05) | 0.1346 (±0.05) | - | - | 1432.0343 (±964.88) | 1.0365 (±0.63) |
| PUGS | 3.3942 (±1.72) | 0.1688 (±0.10) | - | - | 3568.2150 (±2839.13) | 3.2429 (±3.56) |
| Phys4DGen⋆ | 4.8967 (±3.17) | 0.2227 (±0.14) | 0.0407 (±0.04) | 0.1467 (±0.18) | 1865.5673 (±2176.90) | 1.4394 (±2.35) |
| Ours | **0.3793** (±0.29) | **0.0409** (±0.04) | **0.0241** (±0.01) | **0.0818** (±0.03) | **142.6949** (±166.90) | **0.0921** (±0.07) |

(b) **Mechanical Property Estimates** of our method significantly outperform the baselines on all metrics. Per-voxel error rate is first computed per object, then averaged across all objects in the test set to avoid weighing some objects more. Global voxel-level normalization yields similar results, see Supplement Tb. 4.

| Method | ALDE ($\downarrow$) | ADE ($\downarrow$) | ARE ($\downarrow$) | MnRE ($\uparrow$) |
|---|---|---|---|---|
| NeRF2Physics | 0.736 | 12.725 | 1.040 | 0.564 |
| PUGS | 0.661 | 9.461 | **0.767** | **0.576** |
| Phys4DGen⋆ | 0.664 | 9.961 | 0.825 | 0.566 |
| Ours | **0.631** | **8.433** | 0.887 | **0.576** |

(c) **Mass Estimate:** We show the errors for estimating mass of objects on the ABO-500 (Collins et al., 2022) dataset, the only existing benchmark, approximating the accuracy of our $\rho$ estimates.

| Method | $\log(E)(\downarrow)$ | $\nu(\downarrow)$ | $\rho(\downarrow)$ |
|---|---|---|---|
| NeRF2Physics | 1.62 (±4.96) | – | 19.75 (±46.60) |
| PUGS | 1.87 (±4.50) | – | 13.24 (±12.63) |
| Phys4DGen⋆ | 1.77 (±8.53) | 0.85 (±3.01) | 39.49 (±35.47) |
| Pixie | 11.90 (±17.41) | 3.46 (±4.42) | 46.58 (±36.35) |
| Ours | **0.29** (±1.23) | **0.00** (±0.00) | **11.75** (±4.02) |

(d) **Material Validity:** We report mean values and relative errors (in %) with the closest physically measured material range in MTD (§5.1).

Figure 6: **Quantitative Results and Comparisons:** We compare our method against prior art NeRF2Physics (Zhai et al., 2024), PUGS (Shuai et al., 2025) and Phys4DGen (Lin et al., 2025a), and include limited early results comparing with concurrent method Pixie (Le et al., 2025).

that our method significantly outperforms prior art in accuracy and speed, lowering the barrier for integrating accurate physics into digital workflows across 3D representations, with potential impact across digital twins, robotics, and beyond.

While we show important advances over existing works, our method is not without limitations, which we hope will open exciting avenues of future research. Due to fixed-grid voxelization, our output resolution is limited, causing oversmoothing in highly heterogeneous regions, and may result in approximation errors when transferring results to more detailed input geometry. During annotation, we assume part-level materials are isotropic, which is not a true assumption for some common materials like wood. Further, future work could extend our method to predict additional properties like yield strength, shear modulus and thermal expansion, or to adapt true material properties output by our method to simulator-specific scales required for faster algorithms or implementations. We hope to support future directions in this area by releasing our material estimation benchmark, and trained models.

| $\log(E)$ ($\downarrow$) | $\nu$ ($\downarrow$) | $\rho$ ($\downarrow$) | $\log(E/\rho)$ ($\downarrow$) | $\log(G)$ ($\downarrow$) | $\log(K)$ ($\downarrow$) | L.S. ($\downarrow$) | E.A. ($\downarrow$) | Bray–Curtis ($\downarrow$) |
|---|---|---|---|---|---|---|---|---|
| 0.0034 | 0.0426 | 0.0330 | 0.0054 | 0.0036 | 0.0036 | 0.0131 | 0.4439 | 0.0411 |

(a) **MatVAE shows excellent reconstruction errors** on the MTD test set across all metrics.

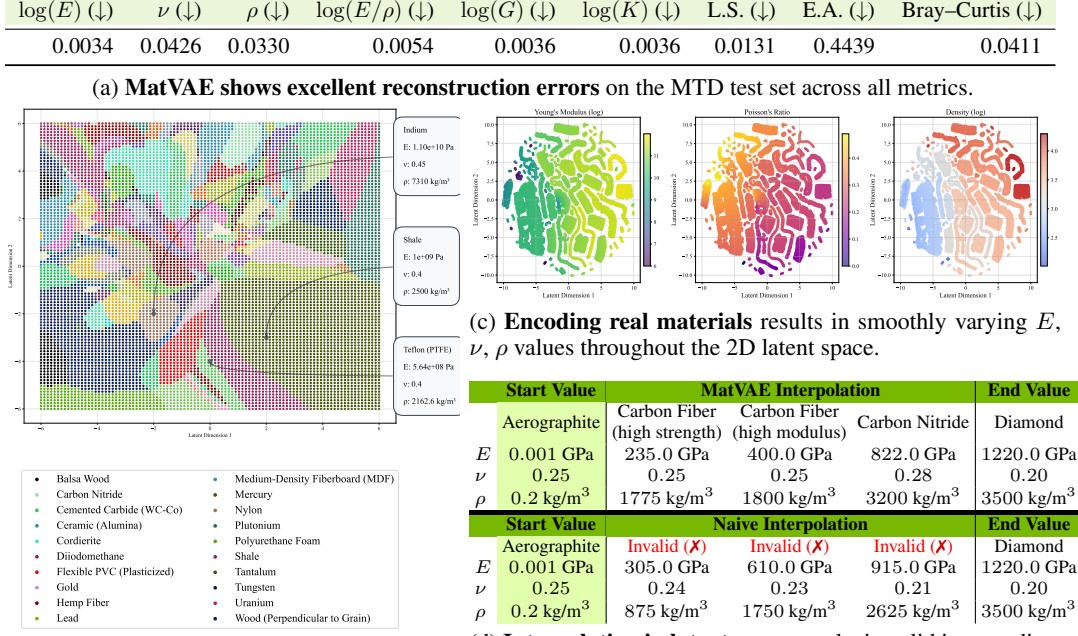

(c) **Encoding real materials** results in smoothly varying $E$, $\nu$, $\rho$ values throughout the 2D latent space.

| | **Start Value** | **MatVAE Interpolation** | | | **End Value** |
|---|---|---|---|---|---|
| | Aerographite | Carbon Fiber (high strength) | Carbon Fiber (high modulus) | Carbon Nitride | Diamond |
| $E$ | 0.001 GPa | 235.0 GPa | 400.0 GPa | 822.0 GPa | 1220.0 GPa |
| $\nu$ | 0.25 | 0.25 | 0.25 | 0.28 | 0.20 |
| $\rho$ | 0.2 kg/m$^3$ | 1775 kg/m$^3$ | 1800 kg/m$^3$ | 3200 kg/m$^3$ | 3500 kg/m$^3$ |
| | **Start Value** | **Naive Interpolation** | | | **End Value** |
| | Aerographite | Invalid (✗) | Invalid (✗) | Invalid (✗) | Diamond |
| $E$ | 0.001 GPa | 305.0 GPa | 610.0 GPa | 915.0 GPa | 1220.0 GPa |
| $\nu$ | 0.25 | 0.24 | 0.23 | 0.21 | 0.20 |
| $\rho$ | 0.2 kg/m$^3$ | 875 kg/m$^3$ | 1750 kg/m$^3$ | 2625 kg/m$^3$ | 3500 kg/m$^3$ |

Balsa Wood
Carbon Nitride
Cemented Carbide (WC-Co)
Ceramic (Alumina)
Cordierite
Diiodomethane
Flexible PVC (Plasticized)
Gold
Hemp Fiber
Lead
Medium-Density Fiberboard (MDF)
Mercury
Nylon
Plutonium
Polyurethane Foam
Shale
Tantalum
Tungsten
Uranium
Wood (Perpendicular to Grain)

(b) **Decoding latent samples** leads to plausible $(E, \nu, \rho)$ values within real-world materials.

(d) **Interpolating in latent space** results in valid intermediate materials, unlike naive $(E, \nu, \rho)$ interpolation.

Figure 7: **Material Latent Space** learned by MatVAE (§3) ensures faithful (a), valid (b), smoothly varying (c), and interpolatable (d) materials. "Invalid" values (c) fall outside all material ranges in MTD (§5.1).

ACKNOWLEDGMENTS

We thank Gilles Daviet for help in setting up some of the simulations. We thank Jean-Francois Lafleche for help with rendering. We thank Beau Perschall for help in using the datasets.

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

# Supplementary Material for VoMP: Predicting Volumetric Mechanical Property Fields

SUPPLEMENTARY CONTENTS

Table 3: **Voxel Mechanical Property Estimation.** Errors for predicting mechanical properties on the publicly released dataset, which does not include the vegetation subset, are very close to the full dataset. These metrics are computed by averaging across all voxels across all 3D objects in the public test set.

| Method | Young's Modulus Pa ($E$) | | Poisson's Ratio ($\nu$) | | Density $\frac{kg}{m^3}$ ($\rho$) | |
|---|---|---|---|---|---|---|
| | ALDE ($\downarrow$) | ALRE ($\downarrow$) | ADE ($\downarrow$) | ARE ($\downarrow$) | ADE ($\downarrow$) | ARE ($\downarrow$) |
| NeRF2Physics | 2.8000 (±1.05) | 0.1346 (±0.05) | - | - | 1432.0343 (±964.88) | 1.0365 (±0.63) |
| PUGS | 3.3942 (±1.72) | 0.1688 (±0.10) | - | - | 3568.2150 (±2839.13) | 3.2429 (±3.56) |
| Phys4DGen* | 4.8967 (±3.17) | 0.2227 (±0.14) | 0.0407 (±0.04) | 0.1467 (±0.18) | 1865.5673 (±2176.90) | 1.4394 (±2.35) |
| Ours | **0.3766** (±0.39) | **0.0421** (±0.05) | **0.0250** (±0.01) | **0.0837** (±0.03) | **113.4683** (±302.19) | **0.0909** (±0.14) |

# A    ADDITIONAL RESULTS

## A.1    END-TO-END EXAMPLES WITH SIMULATION

We demonstrate the process that enables creating simulation-ready, realistic assets from Gaussian Splats and meshes in Fig. 8, demonstrating convincing simulations without any fine-tuning. For example, in Fig. 8c, we first capture a video of an object, and then train a 3D Gaussian Splatting model. Then, we pass it to VoMP which estimates the mechanical properties in a couple of seconds. We then use these properties in a simulator to produce a realistic (see ▶: 0:36), greatly reducing the barrier toward constructing realistic interactive digital worlds directly from our physical reality. We demonstrate a Gaussian Splat scene with multiple simulated objects each of which has properties estimated with VoMP, we then place a robot in the scene interacting with the splats in Fig. 14. Through our experiments on Gaussian Splats we find the Gaussian Splat voxelization scheme (§6.1) we introduce empirically qualitatively accurately voxelizes complex, noisy real-world splat objects.

## A.2    MORE MECHANICAL PROPERTY PREDICTION RESULTS

We show qualitative results for inferring mechanical property fields in Fig. 9 and 10. We notice from Fig. 9 row 1, column 2 that our model can pick up small details like the stem of the orange at the top of the object, which is given a different Young's modulus, though it only spans a few voxels (see ▶: 1:38). We notice from Fig. 9 row 2, column 2 that our model finds that the space inside the pot should be made up of properties that fall in the range of dirt, even though the inside of the pot was not observed through external renders (see ▶: 1:50). We notice from Fig. 9 row 3, column 2 that our model can tolerate some noise in assets such as the Gaussian splat of a bowl with fruits segmented from a larger Gaussian splat (see ▶: 2:01). We notice from Fig. 10 row 1, column 1 that our model can accurately predict thin segments and thin boundaries, like for the seat of the chair (see ▶: 2:10). We notice from Fig. 9 row 3, column 2 that our models can handle complex assets and complex materials like trees and accurately handle fine details, such as understanding where all the leaves lie and giving them different material properties than wood (see ▶: 2:16). We notice from Fig. 9 row 4, column 1 that our models can handle complex volumetric materials, such as annotating the properties of wood under the flowers (see ▶: 2:28). We notice from Fig. 9 row 4, column 2 that our models can handle thin materials and identify the vein of the leaves (see ▶: 2:35). For completeness, we also include our metrics normalized by total test set voxels in Tb. 4 and observe the same performance boost compared to prior art as with the per-object normalization as presented in the main paper Fig. 6b.

We present dataset voxel-averaged results on the publicly-released dataset in Tb. 3. The publicly-released dataset does not include the vegetation subset.

## A.3    MASS ESTIMATION EXAMPLE RESULTS

We show qualitative results from our model on mass estimation on the ABO-500 (Collins et al., 2022) dataset in Fig. 11.

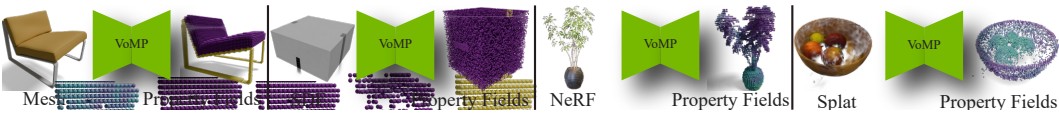

(a) **Inference across representations:** We show that VoMP can operate on any geometry that can be voxelized and rendered, including meshes, SDF, NeRFs, and Gaussian Splats (▶: 1:30).

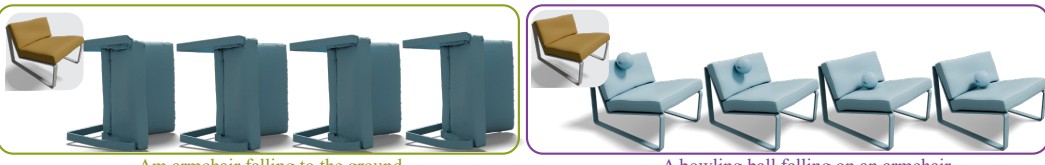

(b) **Simulating meshes:** We show realistic simulations for meshes using predicted material values (▶: 3:40).

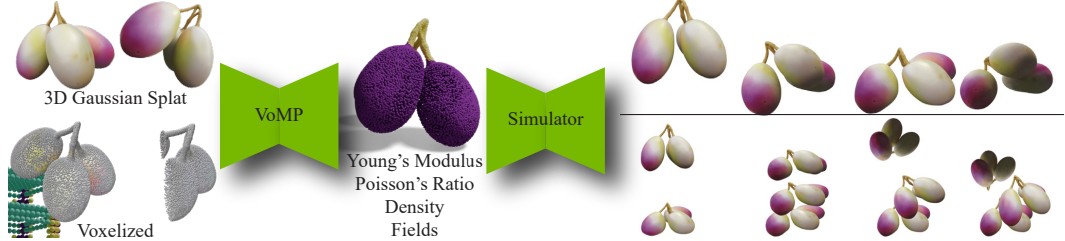

(c) **Simulating captured Gaussian Splat model**: In this example, we apply VoMP to this Gaussian Splat model that we captured using a commercial app. Our method converts this model into a simulation-ready asset (▶: 0:36).

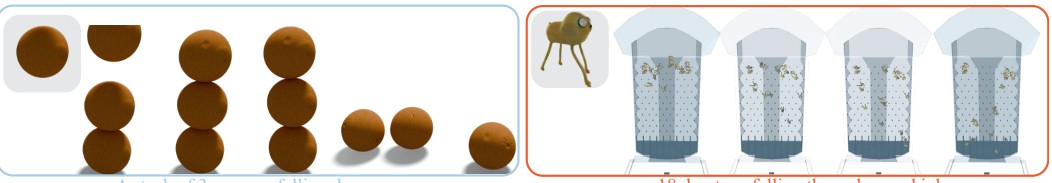

(d) **Simulating meshes and splats:** We show realistic simulations for meshes (left) and splats (right) using predicted material values (▶: 3:20).

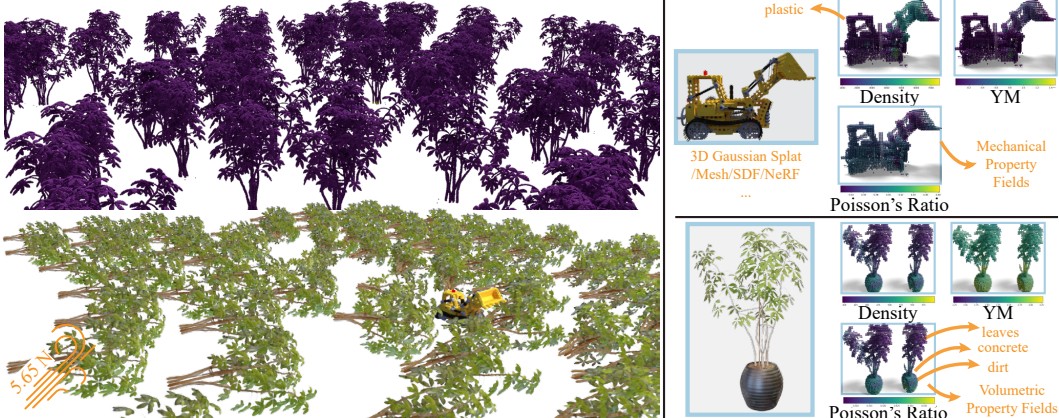

(e) **Simulating Gaussian Splats at scale:** an elastodynamic simulation of a Gaussian Splat bulldozer going through a forest of 100 Gaussian splat ficuses in the presence of wind; all materials predicted by VoMP(▶: 3:00).

Figure 8: **End To End Results:** We test VoMP material estimates on a variety of input representations (a), and show realistic simulations without any hand-tuning for meshes and splats across diverse scenarios. No hand-tuning of our predicted material parameters was performed, showing that VoMP directly predicts simulation-ready parameters. See our ▶ video for the simulation of these examples.

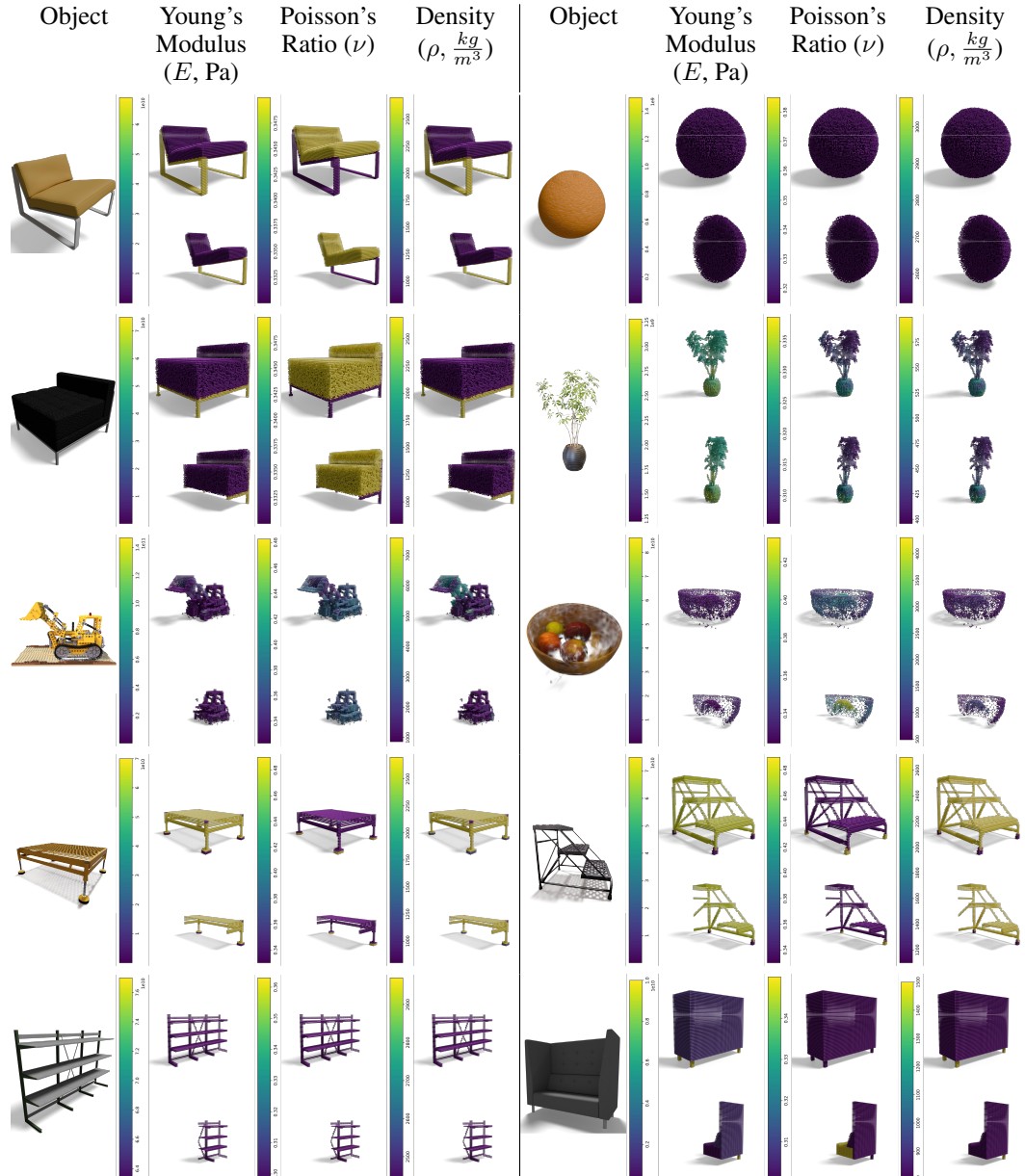

Figure 9: **Inferred Mechanical Property Fields.** We show additional mechnical property fields and slice planes through mechanical property fields estimated by VoMP (▶: 1:40).

Table 4: **Voxel Mechanical Property Estimation.** Errors for predicting mechanical properties from 3D objects averaged across all voxels in the test set.

| Method | Young's Modulus Pa ($E$) | | Poisson's Ratio ($\nu$) | | Density $\frac{kg}{m^3}$ ($\rho$) | |
|---|---|---|---|---|---|---|
| | ALDE (↓) | ALRE (↓) | ADE (↓) | ARE (↓) | ADE (↓) | ARE (↓) |
| NeRF2Physics (Zhai et al., 2024) | 2.5719 (±1.15) | 0.4122 (±0.08) | - | - | 1354.9458 (±1315.71) | 1.1496 (±0.67) |
| PUGS (Shuai et al., 2025) | 3.8619 (±2.01) | 0.4512 (±0.11) | - | - | 3641.0715 (±3320.78) | 4.0413 (±4.16) |
| Phys4DGen* (Lin et al., 2025a) | 5.2977 (±3.36) | 0.4825 (±0.14) | 0.0394 (±0.05) | 0.1425 (±0.21) | 1285.9489 (±1981.11) | 1.0445 (±2.53) |
| Ours | **0.3765** (±0.39) | **0.0421** (±0.05) | **0.0250** (±0.01) | **0.0837** (±0.03) | **113.3807** (±301.90) | **0.0908** (±0.14) |

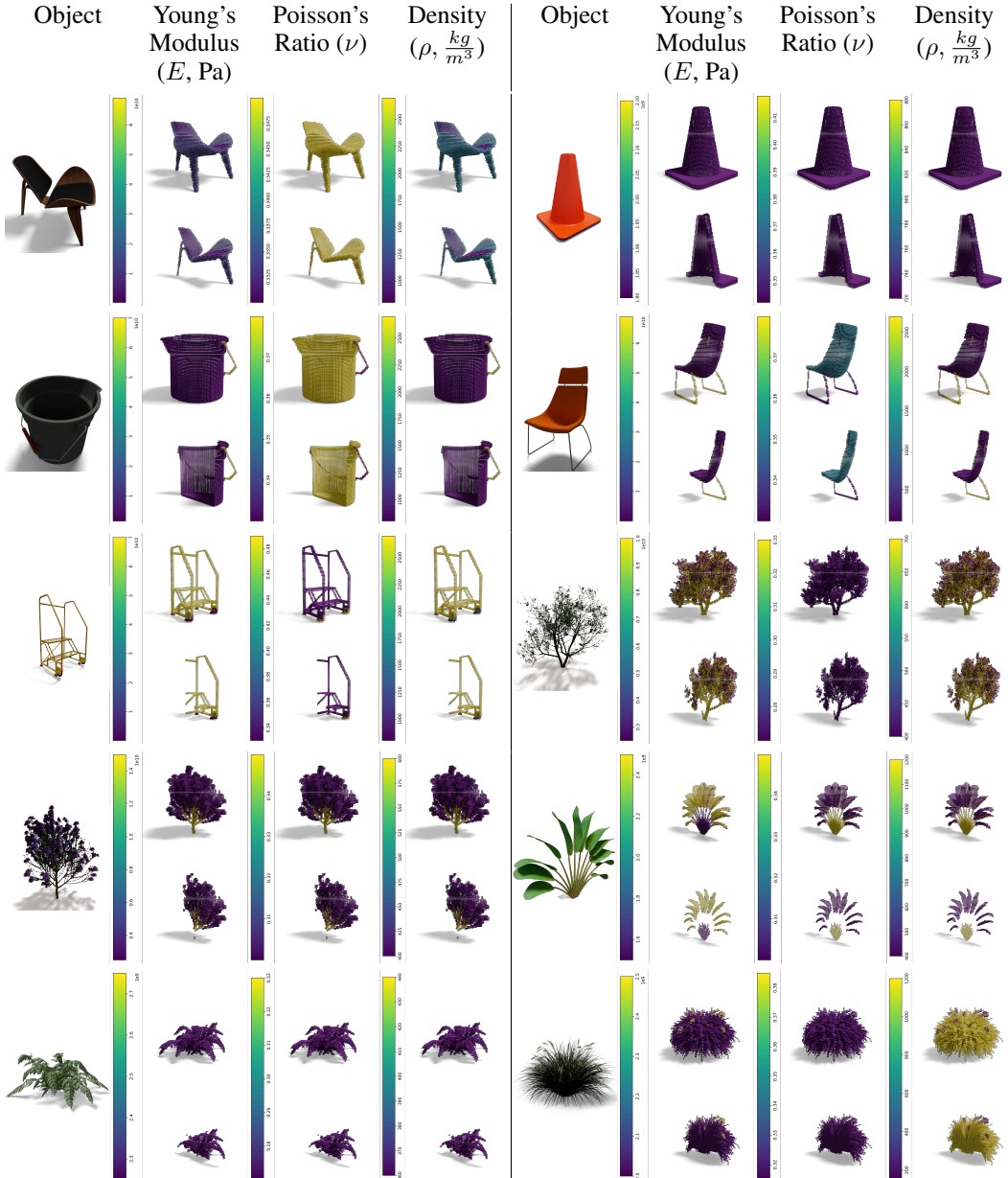

Figure 10: **Inferred Mechanical Property Fields.** We show additional mechnical property fields and slice planes through mechanical property fields estimated by VoMP (▶: 2:10).

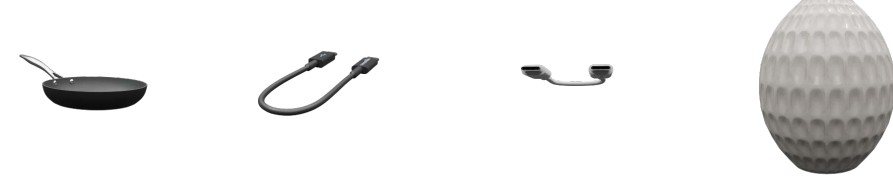

Predicted: 1.58 kg   Predicted: 0.01 kg   Predicted: 0.02 kg   Predicted: 0.71 kg
Ground Truth: 1.58 kg Ground Truth: 0.01 kg Ground Truth: 0.01 kg Ground Truth: 0.70 kg

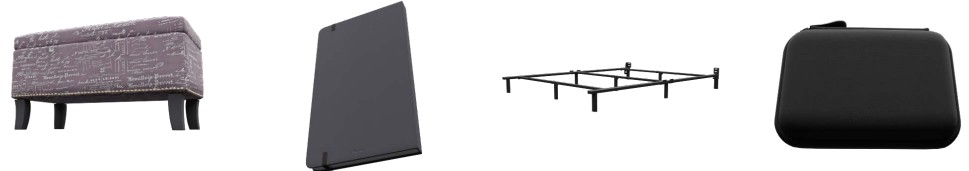

Predicted: 9.09 kg   Predicted: 0.37 kg   Predicted: 7.52 kg   Predicted: 0.14 kg
Ground Truth: 9.07 kg Ground Truth: 0.34 kg Ground Truth: 7.54 kg Ground Truth: 0.10 kg

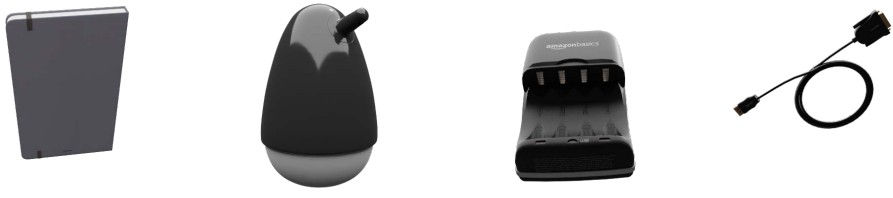

Predicted: 0.38 kg   Predicted: 0.25 kg   Predicted: 0.05 kg   Predicted: 0.19 kg
Ground Truth: 0.32 kg Ground Truth: 0.19 kg Ground Truth: 0.11 kg Ground Truth: 0.12 kg

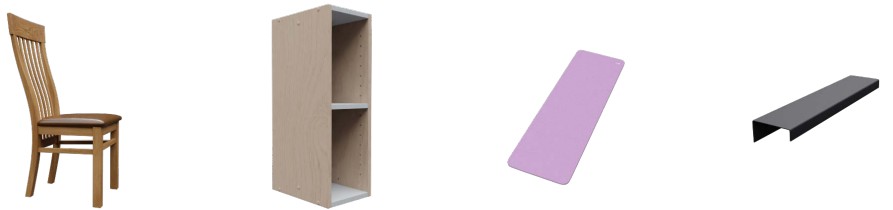

Predicted: 7.68 kg   Predicted: 4.56 kg   Predicted: 1.23 kg   Predicted: 0.80 kg
Ground Truth: 7.60 kg Ground Truth: 4.65 kg Ground Truth: 1.12 kg Ground Truth: 0.91 kg

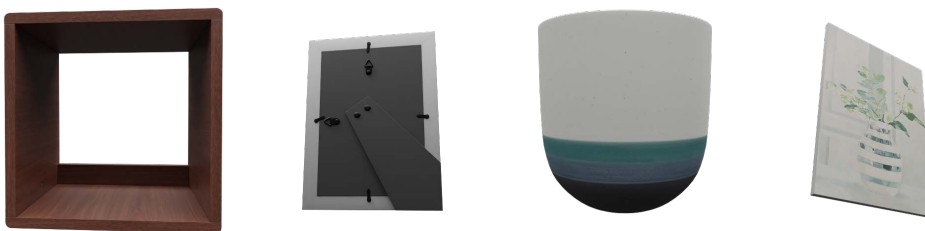

Predicted: 4.47 kg   Predicted: 0.42 kg   Predicted: 2.40 kg   Predicted: 2.85 kg
Ground Truth: 4.36 kg Ground Truth: 0.29 kg Ground Truth: 2.27 kg Ground Truth: 2.72 kg

Figure 11: **Mass Estimation.** We show qualitative results of estimating mass from the ABO-500 (Collins et al., 2022) dataset.

Table 5: **Distribution learned by MatVAE** compared to the distribution of MTD test set.

| Young's Modulus ($E$) | | | Poisson's Ratio ($\nu$) | | | Density ($\rho$) | | |
|---|---|---|---|---|---|---|---|---|
| $W_1$ ($\downarrow$) | $W_2$ ($\downarrow$) | $D_{KL}$ ($\downarrow$) | $W_1$ ($\downarrow$) | $W_2$ ($\downarrow$) | $D_{KL}$ ($\downarrow$) | $W_1$ ($\downarrow$) | $W_2$ ($\downarrow$) | $D_{KL}$ ($\downarrow$) |
| 0.0405 | 0.0798 | 0.1379 | 0.0317 | 0.0437 | 0.0342 | 0.0132 | 0.0172 | 0.0260 |

## A.4 ADDITIONAL MATVAE RESULTS

### A.4.1 DISTRIBUTION LEARNED BY MATVAE

We measure standard metrics used for measuring the difference in the distribution learned by MatVAE and the distribution of the test set in Tb. 5, observing small errors which suggest that MatVAE learned a good approximation of the true material distribution.

### A.4.2 MOVING ACROSS MATVAE LATENT SPACE

We show an example of moving across the latent space in Fig. 13. For each setting, we take a point in our latent space and move across both of its dimensions to obtain multiple smoothly varying material properties. We apply these properties to a bunny and simulate dropping it to the ground with a FEM simulator and these various material properties. The color plots show the average displacements of the mesh from its rest state across simulation steps (e.g. Fig. 13b), demonstrating that even the actual physical behavior correlates with the dimensions of the latent space. Thus, we find that MatVAE learns a rich, meaningful latent space with smooth interpolation and ensures generating physically valid material triplets.

### A.4.3 INTERPOLATION WITH MATVAE

We show additional examples of interpolating in the MatVAE latent space (§6.4) in Fig. 12.

| | Start Value | MatVAE Interpolation | | | End Value |
|---|---|---|---|---|---|
| | Styrofoam | Vanadium | Rhodium | Tungsten | Osmium |
| $E$ | 0.002 GPa | 128.0 GPa | 275.0 GPa | 411.0 GPa | 550.0 GPa |
| $\nu$ | 0.33 | 0.37 | 0.30 | 0.28 | 0.25 |
| $\rho$ | 25 kg/m$^3$ | 6110 kg/m$^3$ | 12450 kg/m$^3$ | 19250 kg/m$^3$ | 22570 kg/m$^3$ |
| | Start Value | Naive Interpolation | | | End Value |
| | Styrofoam | Invalid (✗) | Invalid (✗) | Invalid (✗) | Osmium |
| $E$ | 0.002 GPa | 137.5 GPa | 275.0 GPa | 412.5 GPa | 550.0 GPa |
| $\nu$ | 0.33 | 0.31 | 0.29 | 0.27 | 0.25 |
| $\rho$ | 25 kg/m$^3$ | 5661 kg/m$^3$ | 11298 kg/m$^3$ | 16934 kg/m$^3$ | 22570 kg/m$^3$ |

Figure 12: **Interpolating in MatVAE latent space:** an additional example of interpolation, complementary to Fig. 7d.

| Setting | Position | Material | Young's Modulus (Pa) | | Poisson's Ratio | | Density (kg/m³) | |
| --- | --- | --- | --- | --- | --- | --- | --- | --- |
| | | | Interpolated | True Range | Interpolated | True Range | Interpolated | True Range |
| Fig. 13b | Top-left (↖) | Aerographite | $4.4 \times 10^5$ | $1.0 \times 10^5 - 1.0 \times 10^6$ | 0.241 | 0.2–0.3 | 0.2 | 0.2–0.2 |
| | Top-right (↗) | Polyurethane Foam | $4.8 \times 10^6$ | $1.0 \times 10^5 - 5.0 \times 10^6$ | 0.304 | 0.30–0.30 | 298.2.0 | 50–300 |
| | Bottom-left (↙) | Rubber (soft) | $3.1 \times 10^6$ | $3.0 \times 10^6 - 5.0 \times 10^6$ | 0.488 | 0.48–0.50 | 952.0 | 950–950 |
| | Bottom-right (↘) | Styrofoam | $1.6 \times 10^6$ | $1.0 \times 10^6 - 3.0 \times 10^6$ | 0.322 | 0.3–0.35 | 22.6 | 15–35 |
| Fig. 13c | Top-left (↖) | Aerogel | $4.4 \times 10^6$ | $1.0 \times 10^6 - 1.0 \times 10^7$ | 0.257 | 0.2–0.3 | 1.0 | 1.0–1.0 |
| | Top-right (↗) | Neoprene | $1.0 \times 10^7$ | $1.0 \times 10^6 - 1.0 \times 10^7$ | 0.494 | 0.45–0.5 | 1232.0 | 1230–1250 |
| | Bottom-left (↙) | EPDM Rubber | $6.6 \times 10^6$ | $5.0 \times 10^6 - 1.0 \times 10^7$ | 0.488 | 0.49–0.49 | 1100.9 | 1100–1100 |
| | Bottom-right (↘) | Flexible PVC (Plasticized) | $4.8 \times 10^7$ | $2.0 \times 10^7 - 1.0 \times 10^8$ | 0.450 | 0.45–0.45 | 1209.5 | 1200–1400 |
| Fig. 13d | Top-left (↖) | Polystyrene Foam (EPS) | $2.6 \times 10^6$ | $1.0 \times 10^6 - 5.0 \times 10^6$ | 0.104 | 0.10–0.10 | 59.1 | 30–100 |
| | Top-right (↗) | Chloroprene Rubber (Neoprene) | $5.0 \times 10^6$ | $5.0 \times 10^6 - 5.0 \times 10^6$ | 0.490 | 0.49–0.49 | 1200.8 | 1200–1200 |
| | Bottom-left (↙) | Polystyrene (Foam) | $5.8 \times 10^6$ | $2.5 \times 10^6 - 7.0 \times 10^6$ | 0.371 | 0.34–0.4 | 34.8 | 15–35 |
| | Bottom-right (↘) | Polybutylene (PB) | $2.5 \times 10^8$ | $2.5 \times 10^8 - 3.0 \times 10^8$ | 0.400 | 0.4–0.42 | 932.0 | 930–950 |

(a) Corner Materials for our experiments on moving across the latent space (Fig. 13b to 13d).

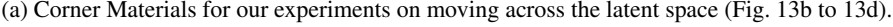
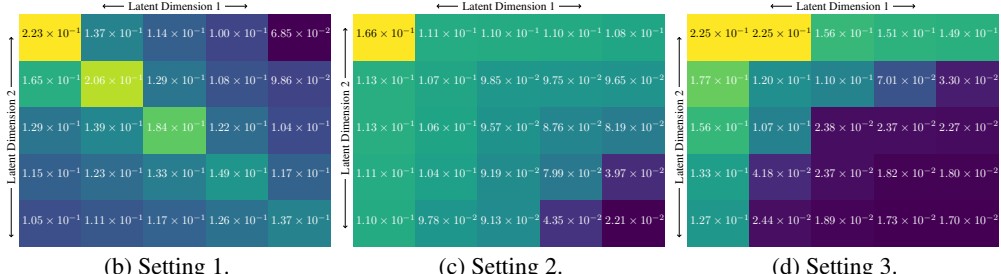

(b) Setting 1.      (c) Setting 2.      (d) Setting 3.

Figure 13: **Moving Across MatVAE latent space.** We sample 3 different valid mechanical property triplets ($E$, $\nu$, $\rho$) (Setting 1,2,3), corresponding to the middle square in the three color diagrams. We encode each of these triplets with MatVAE, and then traverse the 2D latent space to build a $5 \times 5$ grid of latents around the starting value, which are each decoded to actual mechanical properties. To visualize if latent space dimensions correlate with actual simulation performance, we apply each meachnical property triplet to a dropping bunny simulation and measure its mean displacement from rest, which is color coded in the graphs below. Each diagram (b, c, d) thus corresponds to 25 simulation runs with different parameters. We observe a clear correlation between latent dimensions and simulation behavior. (▶: 4:40)

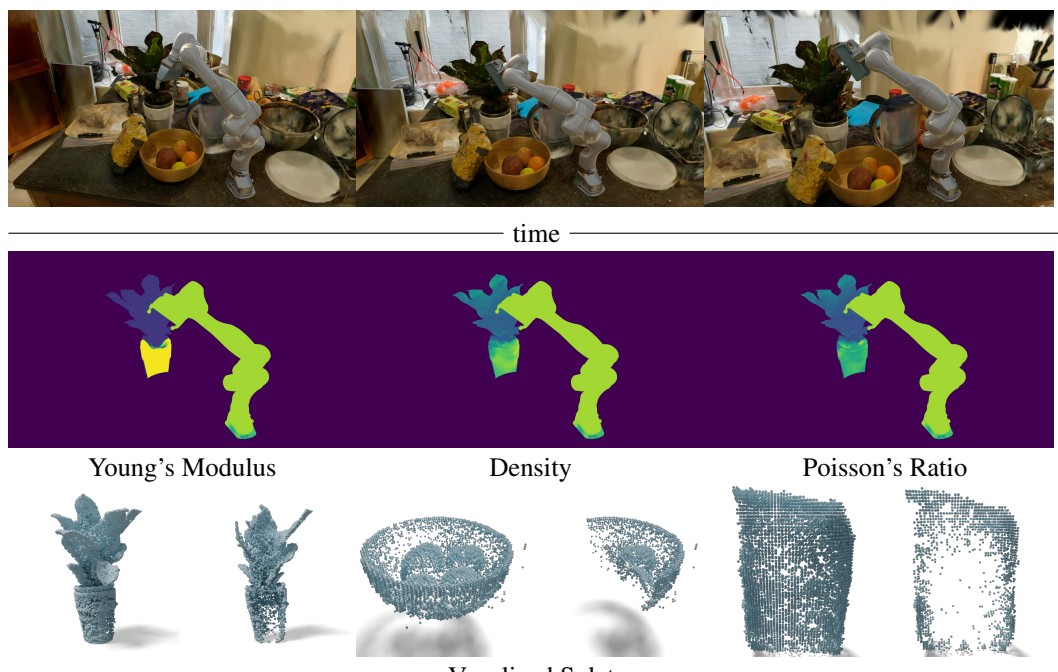

Figure 14: **Simulating a Large Gaussian Splat Scene.** We demonstrate an elastodynamic simulation of a large Gaussian Splat scene with multiple objects segmented out and being assigned properties with VoMP.

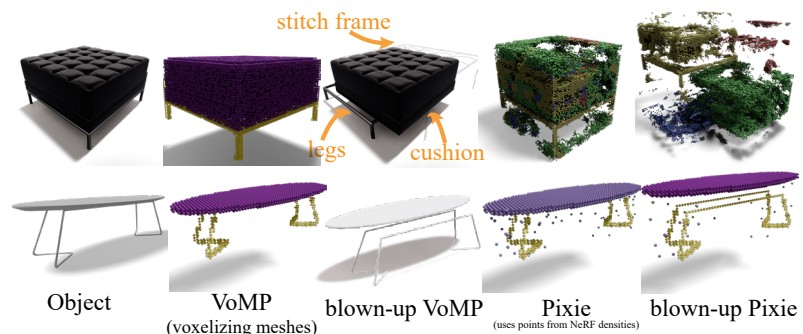

Figure 15: **Data Annotation.** We compare data annotation process of VoMP and Pixie.

Table 6: **Comparison of Mapped Material Properties** between Pixie's in-context physics examples (Le et al., 2025) and the datasets of known material properties.

| Item | Mapped Materials | Pixie $\rho$ | Dataset $\rho$ | Pixie $E$ | Dataset $E$ | Pixie $\nu$ | Dataset $\nu$ |
|---|---|---|---|---|---|---|---|
| tree/pot | Clay Brick
Porcelain (Ceramic)
Glass Ceramic | [400, 400] | [1900, 1900]
[2400, 2400]
[2400, 2600] | [2.000e+08, 2.000e+08] | [2.000e+09, 6.000e+09]
[7.000e+10, 7.000e+10]
[9.000e+10, 1.100e+11] | [0.40, 0.40] | [0.20, 0.20]
[0.20, 0.20]
[0.24, 0.25] |
| tree/trunk | Wood
Oak (White)
Maple Wood (Sugar) | [400, 400] | [700, 700]
[770, 800]
[630, 690] | [2.000e+06, 2.000e+06] | [8.000e+09, 1.100e+10]
[1.200e+10, 1.500e+10]
[1.000e+10, 1.300e+10] | [0.40, 0.40] | [0.30, 0.50]
[0.30, 0.40]
[0.30, 0.40] |
| tree/leaves | — | — | [200, 200] | [2.000e+04, 2.000e+04] | — | [0.40, 0.40] | — |
| flowers/vase | Glass (Soda-Lime)
Glass (Borosilicate) | [500, 500] | [2500, 2500]
[2300, 2300] | [1.000e+06, 1.000e+06] | [7.200e+10, 7.400e+10]
[6.200e+10, 8.100e+10] | [0.30, 0.30] | [0.23, 0.23]
[0.20, 0.20] |
| flowers/flowers | — | — | [100, 100] | [1.000e+04, 1.000e+04] | — | [0.40, 0.40] | — |
| shrub/stems | Wood | [300, 300] | [700, 700] | [1.000e+05, 1.000e+05] | [8.000e+09, 1.100e+10] | [0.35, 0.35] | [0.30, 0.50] |
| shrub/twigs | Wood | [250, 250] | [700, 700] | [6.000e+04, 6.000e+04] | [8.000e+09, 1.100e+10] | [0.38, 0.38] | [0.30, 0.50] |
| shrub/foliage | — | — | [150, 150] | [2.000e+04, 2.000e+04] | — | [0.40, 0.40] | — |
| grass/blades | Rubber (soft)
EPDM Rubber
Neoprene | [80, 80] | [950, 950]
[1100, 1100]
[1230, 1250] | [1.000e+04, 1.000e+04] | [3.000e+06, 5.000e+06]
[1.000e+07, 1.000e+07]
[1.000e+06, 1.000e+07] | [0.45, 0.45] | [0.48, 0.50]
[0.49, 0.49]
[0.45, 0.50] |
| soil (if visible) | Sandy Loam | [1200, 1200] | [1600, 1800] | [5.000e+05, 5.000e+05] | [1.000e+08, 5.000e+08] | [0.30, 0.30] | [0.31, 0.31] |
| rubber_ducks_and_toys/toy | Rubber (soft)
EPDM Rubber
Neoprene
Flexible PVC (Plasticized) | [80, 150] | [950, 950]
[1100, 1100]
[1230, 1250]
[1200, 1400] | [3.000e+04, 5.000e+04] | [3.000e+06, 5.000e+06]
[1.000e+07, 1.000e+07]
[1.000e+06, 1.000e+07]
[2.000e+07, 1.000e+08] | [0.40, 0.45] | [0.48, 0.50]
[0.49, 0.49]
[0.45, 0.50]
[0.45, 0.45] |
| sport_balls/ball | Rubber (soft)
EPDM Rubber
Neoprene | [80, 150] | [950, 950]
[1100, 1100]
[1230, 1250] | [3.000e+04, 5.000e+04] | [3.000e+06, 5.000e+06]
[1.000e+07, 1.000e+07]
[1.000e+06, 1.000e+07] | [0.40, 0.45] | [0.48, 0.50]
[0.49, 0.49]
[0.45, 0.50] |
| soda_cans/can | Aluminium
Aluminum 2024-T3
Aluminum 7075-T6 | [2600, 2800] | [2700, 2700]
[2780, 2780]
[2810, 2810] | [5.000e+10, 8.000e+10] | [7.000e+10, 7.000e+10]
[7.240e+10, 7.240e+10]
[7.100e+10, 7.100e+10] | [0.25, 0.35] | [0.35, 0.35]
[0.33, 0.33]
[0.33, 0.33] |
| metal_crates/crate | Steel
Stainless Steel 17-7PH
Stainless Steel 440A | [2500, 2900] | [7700, 7700]
[7800, 7800]
[7800, 7800] | [8.000e+07, 1.200e+08] | [2.000e+11, 2.000e+11]
[2.040e+11, 2.040e+11]
[2.000e+11, 2.000e+11] | [0.25, 0.35] | [0.31, 0.31]
[0.30, 0.30]
[0.30, 0.30] |
| sand/sand | Sandy Loam | [1800, 2200] | [1600, 1800] | [4.000e+07, 6.000e+07] | [1.000e+08, 5.000e+08] | [0.25, 0.35] | [0.31, 0.31] |
| jello_block/jello | — | — | [40, 60] | — | [8.000e+02, 1.200e+03] | — | [0.25, 0.35] | — |
| snow_and_mud/snow_and_mud | Sandy Loam | [2000, 3000] | [1600, 1800] | [8.000e+04, 1.200e+05] | [1.000e+08, 5.000e+08] | [0.15, 0.25] | [0.31, 0.31] |

## B    COMPARISON WITH CONCURRENT WORK

Although Pixie (Le et al., 2025) is concurrent with us, we still compare aspects of our approach with Pixie. We discuss differences with Pixie in §2.2.

**Data Annotation Process.**    We compare our data annotation process with the annotation process of Pixie (Le et al., 2025) in Fig. 15. Our method performs annotation from meshes while Pixie (Le et al., 2025) gets points from training a NeRF, which often produces noisy points, and the segmentation is performed based on CLIP features, which often produces noisy segmentation for difficult objects.

**Validity of Materials for Data Annotation.**    Although we do not have access to the Pixie (Le et al., 2025) dataset, Pixie uses in-context physics examples, which include material names and ranges of mechanical property triplets in the annotation process. We analyze these in-context physics examples and compare them with real material ranges from MTD (§5.1) in Tb. 6. We find that some of these in-context properties might create pleasing simulations with a particular simulator but can fall outside the range of real materials.

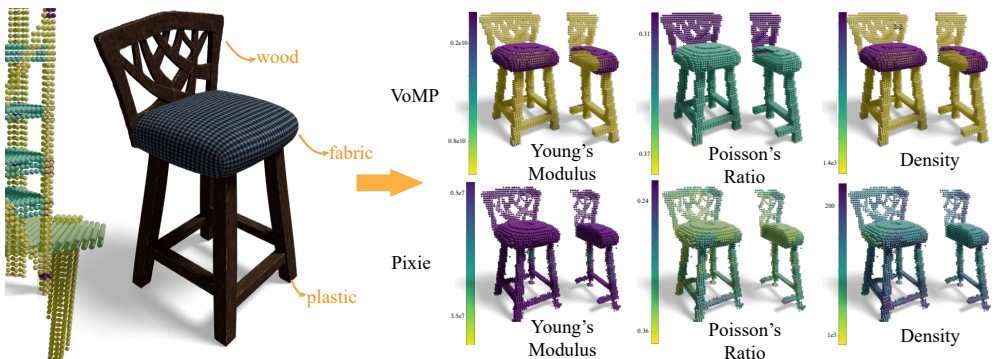

Figure 16: **Comparing Estimations.** We compare property estimates between VoMP and Pixie for a mesh.

**Comparison of Outputs.** Although Pixie (Le et al., 2025) is concurrent with us, we still compare the outputs of Pixie with VoMP on one object in Fig. 16.

## C ABLATIONS

We provide an in-depth analysis motivating our MatVAE and Geometry Transformer training scheme by ablating each component in Tb. 7 and 8. Our ablations require changing the hyperparameters for fair comparisons; thus, for each ablation, we tune our hyperparameters within an identical compute budget.

**MatVAE vs Vanilla VAE.** Technically, MatVAE (§3) is built on top of the vanilla VAE (Kingma & Welling, 2022) and we can use a vanilla VAE (Kingma & Welling, 2022) in its place. In Tb. 7, we show material property reconstruction and distributional metrics between a vanilla VAE and our MatVAE. MatVAE outperforms the vanilla VAE in almost all metrics. We find that Vanilla VAE collapses to the Young's Modulus property, giving us a low reconstruction error for Young's Modulus but significantly higher errors for other properties.

**Image Features.** For aggregating image features (§4.1), we experiment with using DINOv2 (Oquab et al., 2024), CLIP (Radford et al., 2021), and RGB colors by average pooling in the voxel. The results are shown in Tb. 8. Our model had many layers in the Geometry Transformer (§4.2 and appendix F.2) initialized from a generation model (Xiang et al., 2025). This set of ablations was trained starting from random weights, due to the absence of generation weights for these settings. We find that using DINOv2 (Oquab et al., 2024) and CLIP (Radford et al., 2021) without initializing the weights from TRELLIS (Xiang et al., 2025) performs slightly worse, whereas simply using RGB colors performs significantly worse.

**MatVAE.** While MatVAE acts as a continuous tokenizer, it is possible to have the Geometry Transformer directly predict a $\mathbb{R}^3$ vector *i.e.* directly predict the material triplets. We find this produces significantly worse results for Young's Modulus estimation and Poisson's Ratio estimation.

**Normalization Scheme.** We experiment with different normalization schemes (§3) like $Z$-score, and not using $\log$-space transform for either Young's Modulus or Density. All of these normalization schemes lead to a significant degradation in performance. Most notably, removing the logarithmic scaling for Young's Modulus (w/o $\log(E)$) or using a simple $Z$-score severely harms prediction accuracy.

**Loss.** Our Geometry Transformer (§4.2) is trained with $\ell_2$ loss for reconstruction (Equation (4)). We test the effect of replacing this with an $\ell_1$ loss. This change results in a substantial drop in performance across all metrics, with errors increasing by a factor of 2-3x for most properties. This indicates that the squared error penalty of the $\ell_2$ loss is more effective for this material property regression task.

# D  METRICS

We present an explanation of the metrics we use, and experiments on interpreting these metrics.

## D.1  METRICS FOR MASS AND FIELD ESTIMATION

To evaluate the accuracy of predicted scalar quantities such as object mass, as well as continuous scalar fields like density or stiffness, we use several commonly adopted metrics. Let $y$ denote a ground-truth scalar value or voxel-wise field (e.g., density), and $\hat{y}$ its predicted counterpart.

**Absolute Difference Error (ADE).**  The average absolute error between predicted and ground-truth values:

$$\text{ADE} = \frac{1}{N} \sum_{i=1}^{N} |y_i - \hat{y}_i|. \tag{5}$$

This metric is scale-sensitive and reports the error in physical units (e.g., $\text{kg}/\text{m}^3$ for density, $\text{kg}$ for mass).

**Absolute Log Difference Error (ALDE).**  The average absolute error in logarithmic space:

$$\text{ALDE} = \frac{1}{N} \sum_{i=1}^{N} |\log y_i - \log \hat{y}_i|. \tag{6}$$

This metric captures multiplicative error and is particularly useful for quantities that vary over several orders of magnitude.

**Average Relative Error (ARE).**  The mean relative deviation between predictions and ground truth:

$$\text{ARE} = \frac{1}{N} \sum_{i=1}^{N} \left| \frac{y_i - \hat{y}_i}{y_i} \right|. \tag{7}$$

This dimensionless metric penalizes over- and under-estimates proportionally, making it appropriate for comparing across varying scales.

**Minimum Ratio Error (MnRE).**  A symmetric and bounded measure of relative accuracy:

$$\text{MnRE} = \frac{1}{N} \sum_{i=1}^{N} \min \left( \frac{y_i}{\hat{y}_i}, \frac{\hat{y}_i}{y_i} \right). \tag{8}$$

This metric ranges from 0 to 1 and is maximized when predictions are perfectly accurate. As suggested in prior work (Standley et al., 2017), MnRE avoids bias toward systematic over- or under-estimation and reduces sensitivity to outliers, making it particularly effective for evaluating physical quantity predictions across heterogeneous samples.

Table 7: **MatVAE Ablation.** We ablate MatVAE by comparing it against a vanilla VAE and present reconstruction and distirbutional metrics.

| Model | Young's Modulus ($E$) | | | Poisson's Ratio ($\nu$) | | | Density ($\rho$) | | |
|---|---|---|---|---|---|---|---|---|---|
| | $W_1$ ($\downarrow$) | $W_2$ ($\downarrow$) | $D_{\text{KL}}$ ($\downarrow$) | $W_1$ ($\downarrow$) | $W_2$ ($\downarrow$) | $D_{\text{KL}}$ ($\downarrow$) | $W_1$ ($\downarrow$) | $W_2$ ($\downarrow$) | $D_{\text{KL}}$ ($\downarrow$) |
| Vanilla VAE (Kingma & Welling, 2022) | 0.0653 | 0.0868 | **0.0547** | 0.0849 | 0.1057 | 0.0689 | 0.0547 | 0.0744 | **0.0175** |
| MatVAE | **0.0405** | **0.0798** | 0.1379 | **0.0317** | **0.0437** | **0.0342** | **0.0132** | **0.0172** | 0.0260 |
| | (-0.025) | (-0.007) | (+0.083) | (-0.053) | (-0.062) | (-0.035) | (-0.042) | (-0.057) | (+0.009) |
| w/o NF | 0.0339 | 0.0447 | 0.0441 | 0.0417 | 0.0504 | 0.0848 | 0.0599 | 0.0819 | 0.0529 |
| w/o TC penalty | 0.0633 | 0.0855 | 0.0500 | 0.0844 | 0.1052 | 0.0672 | 0.0525 | 0.0715 | 0.0109 |
| w/o free nats | 0.0749 | 0.1168 | 0.1311 | 0.2014 | 0.2064 | 0.6376 | 0.0421 | 0.0507 | 0.0223 |
| | $\log(E)$ ($\downarrow$) | $\nu$ ($\downarrow$) | $\rho$ ($\downarrow$) | $\log(E/\rho)$ ($\downarrow$) | $\log(G)$ ($\downarrow$) | $\log(K)$ ($\downarrow$) | L.S. ($\downarrow$) | E.A. ($\downarrow$) | Bray–Curtis ($\downarrow$) |
| Vanilla VAE (Kingma & Welling, 2022) | 0.0512 | 15366.8750 | 0.8306 | 0.0542 | 0.0544 | 0.0447 | 0.2384 | 2.1893 | 0.4690 |
| MatVAE | **0.0034** | **0.0426** | **0.0330** | **0.0054** | **0.0036** | **0.0036** | **0.0131** | **0.4439** | **0.0411** |
| | (-0.048) | (-15366.8) | (-0.798) | (-0.049) | (-0.051) | (-0.041) | (-0.225) | (-1.745) | (-0.428) |
| w/o NF | 0.0020 | nan | 0.0160 | 0.0033 | 0.0021 | 0.0021 | 0.0086 | 0.1729 | 0.0234 |
| w/o TC penalty | 0.0499 | 15567.0986 | 0.8298 | 0.0537 | 0.0530 | 0.0437 | 0.2382 | 2.1514 | 0.4562 |
| w/o free nats | 0.0036 | 16332.7829 | 0.0276 | 0.0053 | 0.0037 | 0.0041 | 0.0116 | 0.2966 | 0.0436 |

Table 8: **Ablations.** We ablate VoMP with choice of image features, using MatVAE, normalization scheme, and choice of a loss function. We report the voxel-level mechanical property difference errors.

| Method | Young's Modulus Pa ($E$) | | Poisson's Ratio ($\nu$) | | Density $\frac{kg}{m^3}$ ($\rho$) | |
|---|---|---|---|---|---|---|
| | ALDE ($\downarrow$) | ALRE ($\downarrow$) | ADE ($\downarrow$) | ARE ($\downarrow$) | ADE ($\downarrow$) | ARE ($\downarrow$) |
| Image Features (Geometry Transformer start from random weights, §4.1) | | | | | | |
| w/ DINOv2 (Oquab et al., 2024) | **0.2888** ($\pm$0.41) | 0.0536 ($\pm$0.06) | 0.0259 ($\pm$0.02) | 0.0803 ($\pm$0.08) | 373.5183 ($\pm$675.90) | 0.3126 ($\pm$0.79) |
| w/ CLIP (Radford et al., 2021) | 0.2695 ($\pm$0.42) | 0.0508 ($\pm$0.06) | 0.0250 ($\pm$0.02) | 0.0771 ($\pm$0.07) | 383.5844 ($\pm$766.41) | 0.3110 ($\pm$0.85) |
| w/ RGB colors | 1.2176 ($\pm$0.88) | 0.6593 ($\pm$0.49) | 0.1379 ($\pm$0.06) | 1.1642 ($\pm$0.78) | 3678.4451 ($\pm$8421.17) | 4.7430 ($\pm$2.85) |
| MatVAE (§3) | | | | | | |
| w/o MatVAE | 1.1284 ($\pm$0.52) | 0.1289 ($\pm$0.08) | 0.0480 ($\pm$0.02) | 0.1638 ($\pm$0.08) | 917.5879 ($\pm$428.50) | 0.8637 ($\pm$0.61) |
| Normalization Scheme (§3) | | | | | | |
| w/ $Z$-score | 0.8838 ($\pm$0.61) | 0.0996 ($\pm$0.08) | 0.0814 ($\pm$0.04) | 0.2938 ($\pm$0.20) | 5269.2900 ($\pm$946.03) | 6.4656 ($\pm$4.00) |
| w/o $\log(\rho)$ | 0.6654 ($\pm$0.54) | 0.0748 ($\pm$0.07) | 0.0542 ($\pm$0.03) | 0.1806 ($\pm$0.11) | 549.9512 ($\pm$513.57) | 0.5976 ($\pm$0.39) |
| w/o $\log(E)$ | 0.9033 ($\pm$0.60) | 0.1024 ($\pm$0.09) | 0.1182 ($\pm$0.05) | 0.4189 ($\pm$0.25) | 4051.9121 ($\pm$838.98) | 5.0428 ($\pm$3.22) |
| Loss | | | | | | |
| w/ $\ell_1$ | 0.8947 ($\pm$0.62) | 0.1038 ($\pm$0.09) | 0.0468 ($\pm$0.04) | 0.1666 ($\pm$0.16) | 568.7543 ($\pm$734.10) | 0.6337 ($\pm$0.88) |
| Ours | 0.3765 ($\pm$0.39) | **0.0421** ($\pm$0.05) | **0.0250** ($\pm$0.01) | **0.0837** ($\pm$0.03) | **113.3807** ($\pm$301.90) | **0.0908** ($\pm$0.14) |

## D.2 METRICS TO MEASURE DIFFERENCES IN MECHANICAL PROPERTIES

We use multiple commonly used metrics for measuring differences between mechanical properties.

**Relative Error in** $\log(E)$**.** Relative error between predicted and true values of the logarithm of Young's modulus $E$ reported in units of Pa. This captures relative error in material stiffness across several orders of magnitude.

**Relative Error in** $\nu$**.** Relative Error in linear space for Poisson's ratio $\nu$, a dimensionless measure of lateral contraction under uniaxial loading.

**Relative Error in** $\rho$**.** Relative Error between predicted and true values of material density $\rho$, reported in units of $\text{kg}/\text{m}^3$.

**Relative Error in** $\log(E/\rho)$**.** Relative Error in the logarithm of specific modulus, where $E$ is Young's modulus and $\rho$ is density. Reflects relative deviation in stiffness-to-weight efficiency.

**Relative Error in** $\log(G)$**.** Relative Error in the logarithm of shear modulus $G = \frac{E}{2(1+\nu)}$, representing resistance to shear deformation.

**Relative Error in** $\log(K)$**.** Relative Error in the logarithm of bulk modulus $K = \frac{E}{3(1-2\nu)}$, characterizing resistance to uniform volumetric compression.

**Lightweight Stiffness Ashby Index** ($P = E^{1/2}/\rho$)**.** The Relative Error in $\log(P)$, where $P = E^{1/2}/\rho$, reflecting relative error in predicting material efficiency for maximizing stiffness per unit weight (Ashby & Cebon, 1993).

**Energy Absorption Ashby Index** ($P = E^{1/3}/\rho$)**.** The Relative Error in $\log(P)$, where $P = E^{1/3}/\rho$, quantifying relative deviation in predicted energy absorption efficiency (Ashby & Cebon, 1993).

**Bray–Curtis dissimilarity.** Bray–Curtis dissimilarity (Bray & Curtis, 1957) between predicted and ground-truth property vectors $\mathbf{x}$ and $\mathbf{y}$:

$$\text{BC}(\mathbf{x}, \mathbf{y}) = \frac{\sum_i |x_i - y_i|}{\sum_i (x_i + y_i)}. \tag{9}$$

A normalized, dimensionless measure in $[0, 1]$ capturing overall distributional divergence across multiple material properties.

## D.3 METRICS TO MEASURE DIFFERENCES IN DISTRIBUTIONS

We use multiple commonly-used metrics for measuring differences between distributions.

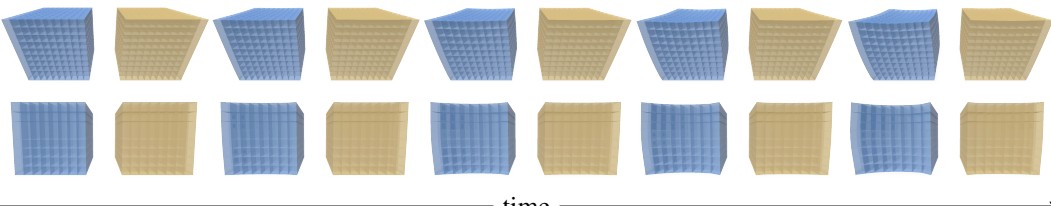

time

Figure 17: **Simulations to Interpret Errors.** We demonstrate simulations performed to show the relation between relative error and simulation error.

**Wasserstein–1 Distance ($W_1$).** For probability measures $\mu, \nu$ on a space $\mathcal{X}$,

$$W_1(\mu, \nu) = \inf_{\gamma \in \Pi(\mu,\nu)} \int_{\mathcal{X} \times \mathcal{X}} \|x - y\| \, d\gamma(x, y), \tag{10}$$

where $\Pi(\mu, \nu)$ is the set of all couplings of $\mu$ and $\nu$. $W_1$ equals the minimum average "work" to move mass from $\mu$ to $\nu$.

**Wasserstein–2 Distance ($W_2$).** For probability measures $\mu, \nu$ on a space $\mathcal{X}$,

$$W_2(\mu, \nu) = \left( \inf_{\gamma \in \Pi(\mu,\nu)} \int_{\mathcal{X} \times \mathcal{X}} \|x - y\|^2 \, d\gamma(x, y) \right)^{1/2}, \tag{11}$$

where $\Pi(\mu, \nu)$ is the set of all couplings of $\mu$ and $\nu$. The root-mean-square transport cost between $\mu$ and $\nu$.

**Kullback–Leibler Divergence ($D_{\mathrm{KL}}$).** For densities $p, q$ on $\mathcal{X}$,

$$D_{\mathrm{KL}}(p\|q) = \int_{\mathcal{X}} p(x) \, \log \frac{p(x)}{q(x)} \, dx. \tag{12}$$

$D_{\mathrm{KL}}$ measures the expected extra log-likelihood of data drawn from $p$ when it is coded using $q$ instead of $p$.

### D.4 INTERPRETING ERRORS FOR MATERIAL PROPERTY ESTIMATION

We experimentally demonstrate an interpretation of how relative changes in material properties affect simulations of the finite element method (FEM) solver. We do so by simulating the deformation of unit cubes under many different material properties and scenarios with an FEM solver.

For each baseline triplet $(E_0, \nu_0, \rho_0)$ representing Young's modulus, Poisson's ratio, and density, we introduce variations following the scaling laws: density variations follow linear scaling $\rho_{\mathrm{new}} = \rho_0(1 + \Delta)$, Poisson's ratio variations use the same linear relationship $\nu_{\mathrm{new}} = \nu_0(1 + \Delta)$, while Young's modulus variations use exponential scaling $E_{\mathrm{new}} = E_0 e^{\Delta}$ to accommodate the wide range of stiffness values. We then apply every such unique material triplet to a unit cube and perform a simulation under some external forces.

During each of these simulations, we measure the final volume and potential energy of the cube after the Newton iterations have converged.

**Measuring Volume.** For a body undergoing deformation, the deformation gradient $\mathbf{F} = \nabla \mathbf{u} + \mathbb{I}$ maps material points from the reference to the current configuration, where $\mathbf{u}$ represents the displacement field and $\mathbb{I}$ is the $3 \times 3$ identity tensor. The local volume change is quantified by the Jacobian $J = \det(\mathbf{F})$, which represents the ratio of deformed to reference volume at each material point. The total deformed volume is: $V_{\mathrm{def}} = \int_{\Omega_0} J \, dV$, where $\Omega_0$ denotes the reference configuration. The relative volume change, defined as $\Delta V / V = (V_{\mathrm{def}} - V_0)/V_0$, provides a dimensionless measure of volumetric deformation.

**Measuring Potential Energy.** We compute the total potential energy by combining elastic strain energy and kinetic-potential contributions. We use corotated linear elasticity, where we calculate the deformation gradient $\mathbf{F} = \nabla \mathbf{u} + \mathbb{I}$ and symmetric strain tensor $\mathbf{S} = \frac{1}{2}(\mathbf{F} + \mathbf{F}^T) - \mathbb{I}$ to obtain the energy density $W = \mu \operatorname{tr}(\mathbf{S}^2) + \frac{\lambda}{2}(\operatorname{tr}(\mathbf{S}))^2$, with Lamé parameters $\mu$ and $\lambda$ derived from the Young's modulus and Poisson's ratio. We use three distinct contributions in the kinetic-potential term: an

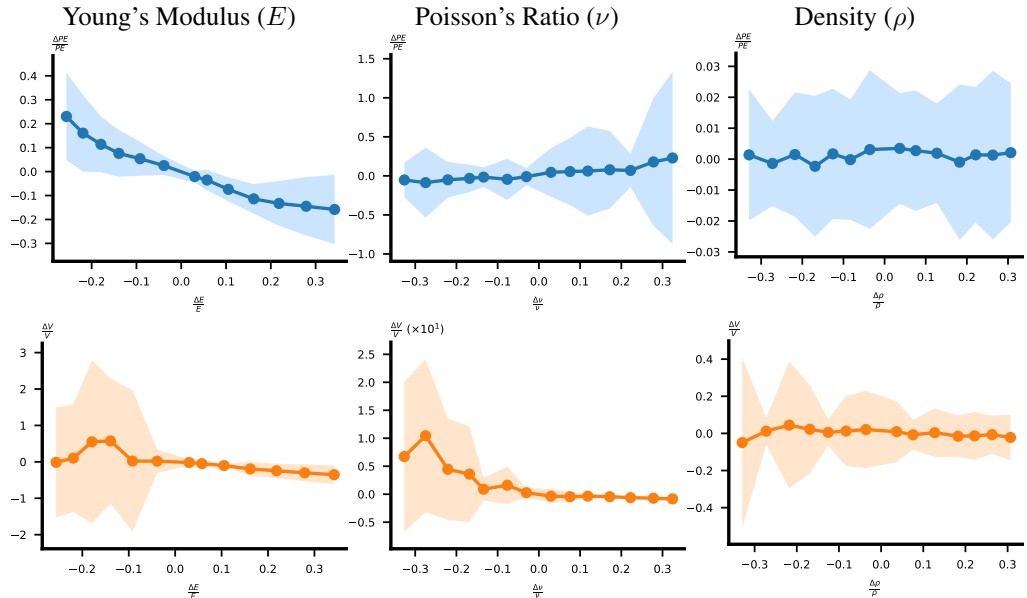

Figure 18: **Gripping Force by Robots.** We demonstrate the relation between relative errors in materials and relative change in P.E. (top) and volume (bottom). We then show the confidence bounds in light shaded regions.

inertial component $E_{\text{inertia}} = \int_\Omega \frac{\rho}{2\Delta t^2}|\mathbf{u}^{n+1} - \mathbf{u}^n|^2\, dV$ that captures displacement changes between iterations in our quasi-static solver, a gravitational potential $E_{\text{gravity}} = -\int_\Omega \rho\, \mathbf{u} \cdot \mathbf{g}\, dV$ accounting for body forces, and an external work term $E_{\text{ext}} = -\int_\Omega \mathbf{u} \cdot \mathbf{f}_{\text{ext}}\, dV$ representing the applied loads. We thus compute the total potential energy as $E_{\text{total}} = \int_\Omega W\, dV + E_{\text{inertia}} + E_{\text{gravity}} + E_{\text{ext}}$, evaluated at the converged displacement field.

We perform the simulations in the following scenarios,

**Gripping force by robots.** We simulate a 140 N compressive force, which is common in robotic gripping applications, for example, the Franka Emika (frankaemika, 2025) "Hand" end effector applies a maximum of 70 N per finger with a maximum clamping force of 140 N. We demonstrate the results from 486 simulations in this setting, all of which were run to convergence in Fig. 18.

**Impact Force on Dropping Objects.** We simulate a 120 N impact force that simulates package drop scenarios, calculated from the impact dynamics of a 1 kg package dropped from 0.6 m height with a 5 cm deformation distance. We demonstrate the results from 486 simulations in this setting, all of which were run to convergence in Fig. 19.

**Tensile Testing Machines.** We simulate a 330 N force corresponding to standard tensile testing conditions employed in bench-top universal testing machines (ASTM Committee D20, 2022; ASTM Committee E28, 2024). We demonstrate the results from 486 simulations in this setting, all of which were run to convergence in Fig. 20.

**Tension.** We simulate a 200 N force, which represents typical pretension in tendon-driven robotic systems, where continuum arms and wearable assistive devices maintain structural stiffness through cable tensions (Schäffer et al., 2024). We demonstrate the results from 486 simulations in this setting all of which were run to convergence in Fig. 21.

# E    DATASET DETAILS

We present additional details about our datasets for training MatVAE (§3) and Geometry Transformer (§4.2).

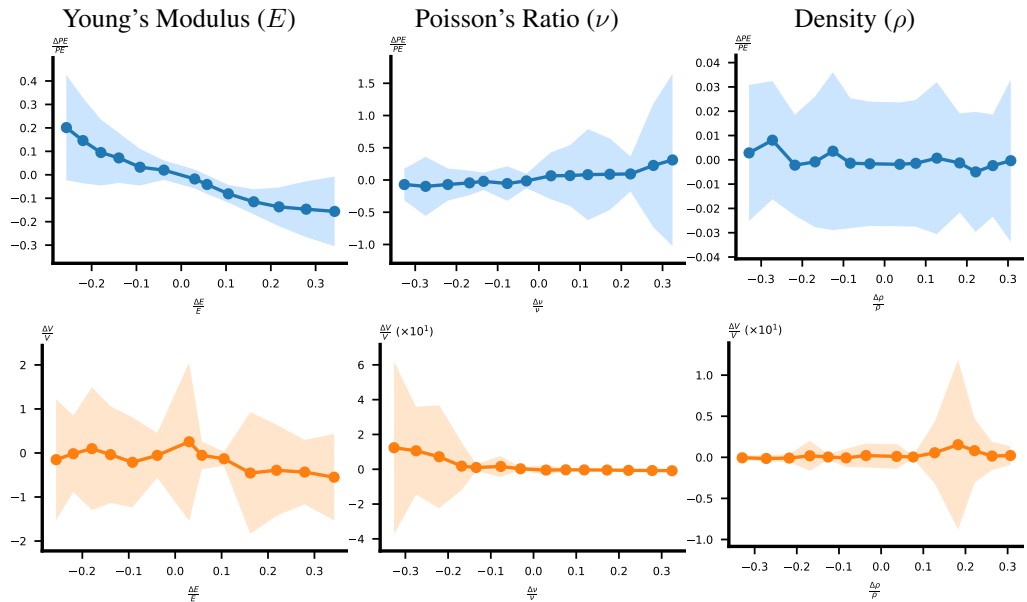

Figure 19: **Impact Force on Dropping Objects.** We demonstrate the relation between relative errors in materials and relative change in P.E. (top) and volume (bottom). We show the confidence bounds in light shaded regions.

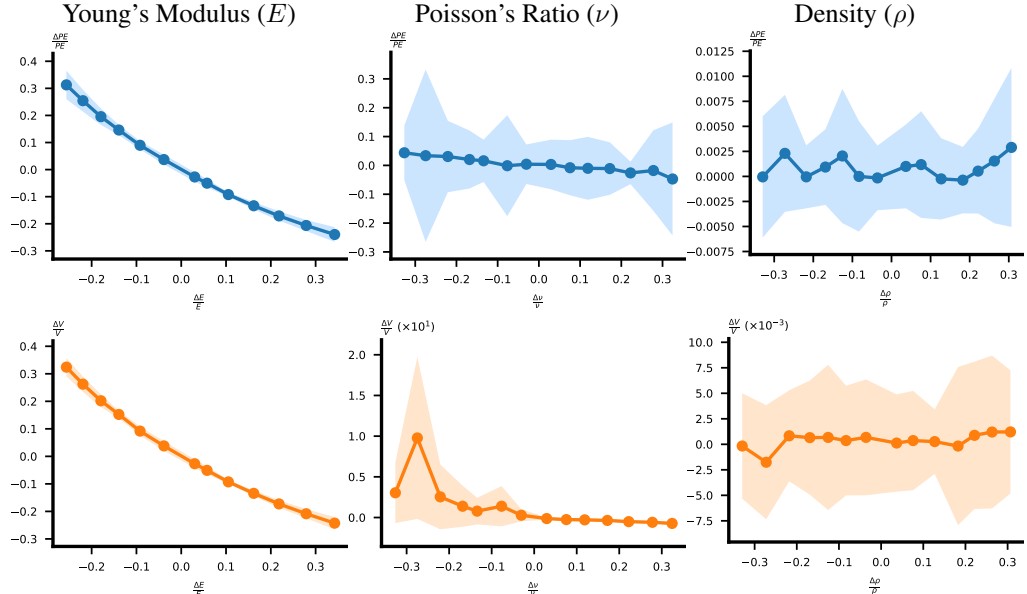

Figure 20: **Tensile Testing Machine.** We demonstrate the relation between relative errors in materials and relative change in P.E. (top) and volume (bottom). We show the confidence bounds in light shaded regions.

### E.1 ANNOTATION WITH VISION-LANGUAGE MODEL

To create our training dataset (§5) for the Geometry Transformer, we use a Vision-Language Model (VLM) coupled with multiple other data sources like 3D assets, component-wise part segmentations, material databases (§5.1), visual textures, and material names to annotate our dataset. We run the VLM on every segment of every object individually. We experiment with Qwen2.5-VL 7B, Qwen2.5-VL 32B, Qwen2.5-VL 72B (Bai et al., 2023; 2025), VL-Rethinker (Wang et al., 2023), SpatialRGPT (Cheng et al., 2024), and Cosmos Nemotron (Lin et al., 2024a). We experimentally choose Qwen2.5-VL 72B for the data annotation. We show the system prompts and the user prompts that we use in Fig. 22 to 25. We find the best performing system prompts with TextGrad (Yuksek-

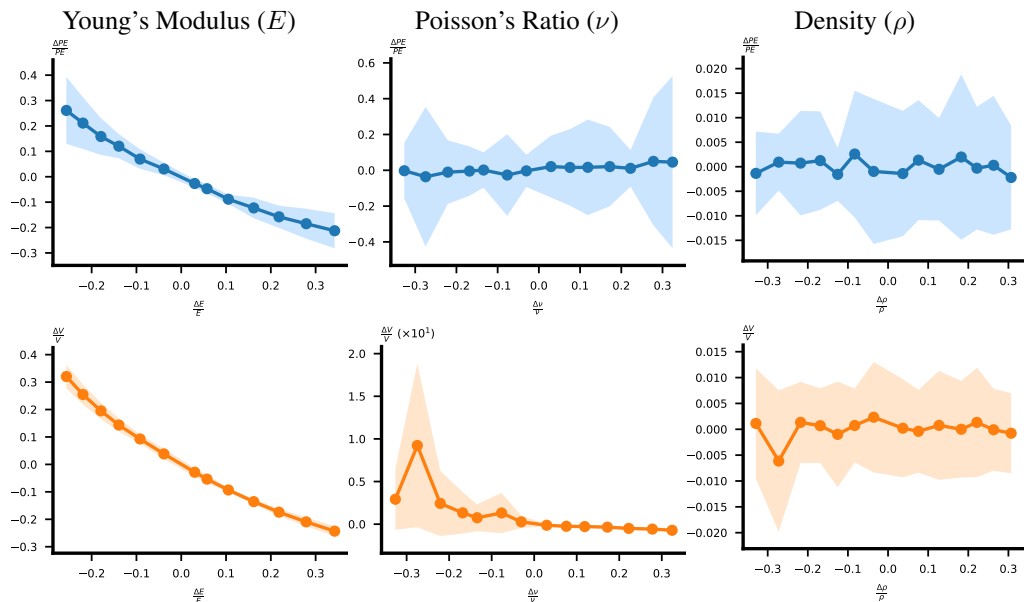

Figure 21: **Tension.** We demonstrate the relation between relative errors in materials and relative change in P.E. (top) and volume (bottom). We show the confidence bounds in light shaded regions.

Table 9: **VLM Annotation Errors.** Errors for the VLM annotation for mechanical property annotation.

| $\log(E)$ (↓) | $\nu$ (↓) | $\rho$ (↓) | $\log(E/\rho)$ (↓) | $\log(G)$ (↓) | $\log(K)$ (↓) | L.S. (↓) | E.A. (↓) | Bray–Curtis (↓) |
|---|---|---|---|---|---|---|---|---|
| 0.0295 | 0.0426 | 0.1348 | 0.1961 | 0.0303 | 0.0330 | 0.2022 | 0.2162 | 0.2342 |

gonul et al., 2025). We show a response from the model for one of the segments from an object in our dataset in Fig. 26 and 27.

We construct a tiny dataset consisting of complex objects that are manually annotated and compare these properties with the annotations from the VLM (Qwen2.5-VL 72B ) in Tb. 9. We observe that the VLM, given significant additional information as we provide, performs close to human annotator performance. We report the commonly used metrics we list in Appendix D.2 for measuring differences in mechanical properties.

### E.2 DATASET STATISTICS

Our dataset comprises a diverse collection of 1,692 objects, sourced from four datasets: simready (NVIDIA Developer, 2025), residential (NVIDIA Corporation, 2025c), vegetation (NVIDIA Corporation, 2025d), and commercial (NVIDIA Corporation, 2025a). As shown in Tb. 10, the majority of objects belong to the simready and residential categories, with vegetation and commercial objects providing additional variety. Each object is decomposed into multiple segments, with a total of 8,128 segments across the dataset. Most parts are labeled with English material names, and for few parts that do not have these material names, we infer these from the PBR texture names that were applied to these parts. To characterize the physical realism and diversity of the dataset, we analyze the distribution of key material properties for all segments in Tb. 11. The wide range of material properties highlights the heterogeneity of the dataset, which is essential for robust learning and evaluation of material-aware models. We summarize the most frequent material categories (e.g., metal, plastic, wood, cardboard) and object classes (e.g., residential, shelf, container), along with their respective counts and proportions in Tb. 12.

We report the statistics of our Material Triplet dataset in Tb. 13. To train MatVAE, we use the "Filtered Dataset".

**System Prompt.**

You are a materials science expert specializing in analyzing material properties from visual appearances and physical data. Your task is to provide precise numerical estimates for Young's modulus, Poisson's ratio, and density based on the images and context provided.

Important Context: The material segment you are analyzing may be an internal component or structure that is not visible from the outside of the object. For example:
- Internal support structures, frames, or reinforcements
- Hidden layers or core materials
- Components enclosed within the outer shell
- Structural elements that are only visible when the object is disassembled

When analyzing:
- Consider that the material might be completely hidden from external view
- Use the semantic usage and material type hints to infer properties of internal components
- Internal structural components often have different properties than visible surfaces
- For example, a soft exterior might hide a rigid internal frame

Critical Instruction: You MUST provide numerical estimates for ALL materials, even organic, biological, or unusual materials like leaves, feathers, or paper.
- For organic materials, estimate properties based on similar natural materials with known values
- For leaves, consider them as thin plant fiber composites with values similar to paper or dried plant fibers
- Never respond with "N/A" or any non-numeric value in your property estimates

When analyzing materials, use step-by-step reasoning:
1. First identify the likely material class and subtype based on visual appearance (if visible) or contextual clues (if internal)
2. Consider how texture, color, and reflectivity inform your understanding of the material (when visible)
3. Incorporate the provided physical properties and contextual usage information
4. For each mechanical property, reason through how the visual and physical attributes lead to your estimate
5. Consider how the material compares to reference materials with known properties
6. If the material appears to be internal/hidden, use the object type and usage context to make informed estimates

Important Formatting Requirements:
- Young's modulus must be provided in scientific notation followed by "Pa" (e.g., 2.0e11 Pa)
- Poisson's ratio must be a simple decimal between 0.0 and 0.5 with no units (e.g., 0.34)
- Density must be provided in kg/m^3 (e.g., 7800 kg/m^3)
- Each property must be on its own line with exactly the label shown in the examples
- Do not include explanatory text or parenthetical notes after the values
- ALWAYS provide numerical values, never text like "N/A" or "unknown"

Figure 22: **System Prompt.** The System Prompt we use for every segment of every object.

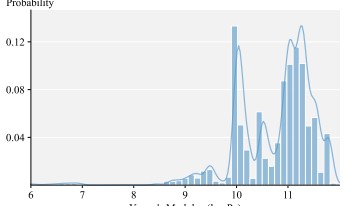
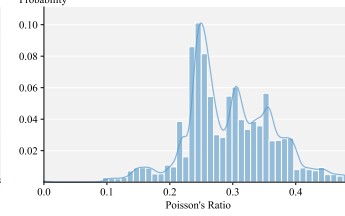
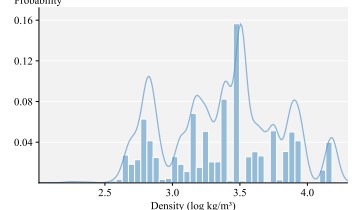

Figure 28: **Young's Modulus Pa ($E$).** Histogram of Young's Modulus in our Geometry with Volumetric Materials Dataset (§5).

Figure 29: **Poisson's Ratio ($\nu$).** Histogram of Poisson's Ratio in our Geometry with Volumetric Materials Dataset (§5).

Figure 30: **Density** $\frac{kg}{m^3}$**.** Histogram of Density in our Geometry with Volumetric Materials Dataset (§5).

Table 10: **Dataset Statistics.** Number of objects, total segments, total points, average segments per object (std. dev.), and average points per object (std. dev.) for each dataset.

| Dataset | Total Objects | Segments (%) | Voxels (%) | Avg. Segments/Object | Avg. Voxels/Object |
|---|---|---|---|---|---|
| commercial | 82 | 650 (8.0) | 1,812,064 (4.9) | 7.93 ($\pm$7.19) | 22,098 ($\pm$22,774) |
| residential | 449 | 4225 (52.2) | 9,109,380 (24.4) | 9.41 ($\pm$21.82) | 20,288 ($\pm$21,714) |
| simready | 1029 | 2544 (31.5) | 24,148,660 (64.7) | 2.47 ($\pm$1.33) | 23,468 ($\pm$25,032) |
| vegetation | 104 | 670 (8.3) | 2,267,848 (6.1) | 6.44 ($\pm$4.53) | 21,806 ($\pm$19,428) |
| train | 1333 | 6477 (80.1) | 28,709,190 (76.9) | 4.86 ($\pm$12.69) | 21,537 ($\pm$23,431) |
| validation | 165 | 552 (6.8) | 3,719,996 (10.0) | 3.35 ($\pm$3.19) | 22,545 ($\pm$23,095) |
| test | 166 | 1060 (13.1) | 4,908,766 (13.1) | 6.39 ($\pm$11.33) | 29,571 ($\pm$25,987) |
| **Total** | 1664 | 8089 (100.0) | 37,337,952 (100.0) | 4.86 ($\pm$11.97) | 22,439 ($\pm$23,786) |

Table 11: **Material property statistics for all segments in the dataset.** We report the minimum, maximum, mean, median, standard deviation, and outlier count (% of values outside $\pm 3\sigma$) for Young's modulus, Poisson's ratio, and Density.

| Property | Min | Max | Mean | Median | Std Dev | Outliers (%) |
|---|---|---|---|---|---|---|
| Density (kg/m$^3$) | $5.0 \times 10^1$ | $1.93 \times 10^4$ | $2.28 \times 10^3$ | $1.20 \times 10^3$ | $2.44 \times 10^3$ | 25 (0.3) |
| Young's Modulus (Pa) | $1.0 \times 10^5$ | $2.8 \times 10^{11}$ | $4.19 \times 10^{10}$ | $1.0 \times 10^{10}$ | $6.53 \times 10^{10}$ | 165 (2.0) |
| Poisson's Ratio | $1.6 \times 10^{-1}$ | $4.9 \times 10^{-1}$ | $3.36 \times 10^{-1}$ | $3.5 \times 10^{-1}$ | $4.36 \times 10^{-2}$ | 88 (1.1) |

# F  ADDITIONAL IMPLEMENTATION DETAILS

We present additional implementation details.

## F.1  DESIGN OF MATVAE

We explain the motivation behind the design of MatVAE.

**Normalizing Flow.** Material triplets remain statistically non-Gaussian even after normalization (heavy-tailed/multi-modal $\log_{10} E$, $\log_{10} \rho$; boundary-concentrated $\nu \in [0, 0.5)$), so a diagonal-Gaussian $q_\phi(z \mid m)$ tends to mode-average and miscalibrate tails. We therefore parameterize the posterior with a bijective normalizing flow (Rezende & Mohamed, 2015) $f_\psi$ where $\psi$ is the parameter of the flow network f, and $\Psi$ is the space of all parameters $\psi$. This $f_\psi$ applied to a Gaussian base $q_0$: sample $u \sim q_0(u \mid m) = \mathcal{N}(u; \mu_\phi(m), \text{diag}\, \sigma_\phi^2(m))$, set $z = f_\psi(u)$, and compute the density by change of variables

$$\log q_\phi(z \mid m) = \underbrace{\log q_0\big(f_\psi^{-1}(z) \mid m\big)}_{\text{base density}} + \underbrace{\log\big|\det J_{f_\psi^{-1}}(z)\big|}_{\text{log-Jacobian}}$$

$$= \underbrace{\log q_0(u \mid m)}_{\text{base density}} - \underbrace{\log\big|\det J_{f_\psi}(u)\big|}_{\text{log-Jacobian}}\bigg|_{u=f_\psi^{-1}(z)}. \tag{13}$$

with a standard normal prior $p(z) = \mathcal{N}(0, I)$ and decoder likelihood $p_\theta(m \mid z)$.

Table 12: **Most frequent high-level material categories and object classes.** We report the top high-level material categories (aggregated and deduplicated) and the most common object classes in the dataset, with their respective counts and percentages.

| Mat. Category | Count (%) | Object Class | Count (%) |
|---|---|---|---|
| Metal | 1434 (17.7) | residential | 477 (28.2) |
| Plastic | 553 (6.8) | shelf | 250 (14.8) |
| Wood | 707 (8.7) | container | 193 (11.4) |
| Cardboard | 315 (3.9) | cardboard box | 106 (6.3) |
| Leather | 140 (1.7) | vegetation | 105 (6.2) |
| Chrome | 171 (2.1) | crate | 101 (6.0) |
| Glass | 85 (1.0) | commercial | 82 (4.8) |
| Fabric | 60 (0.7) | pallet | 67 (4.0) |
| Rubber | 55 (0.7) | barricade tape | 59 (3.5) |
| Stone | 40 (0.5) | inclined plane | 22 (1.3) |

Table 13: **Dataset Details.** We present statistics of our dataset for training MatVAE (§3).

| Dataset | Size |
|---|---|
| Material Ranges | 249 |
| Extracted Materials | 105456 |
| Filtered Materials | 101517 |

This keeps the ELBO form unchanged while strictly enlarging the variational family (the identity map recovers the Gaussian case), allowing $q_\phi$ to match the true posterior and avoid mode-averaging on $(E, \nu, \rho)$. We instantiate $f_\psi$ as a single radial flow,

$$f_\psi(u) = u + \underbrace{\beta\,h(u)}_{\text{radial scale}}\,\overbrace{(u - z_0)}^{\text{displacement}},$$

$$h(u) = \frac{1}{\left(\alpha + \|u - z_0\|_2\right)},$$

(14)

whose log-determinant has the closed form (substitute $h' = -h^2$ in the closed form from (Rezende & Mohamed, 2015)),

$$\log \det J_{f_\psi}(u) = \underbrace{(D - 1)\,\log\!\big(1 + \beta h(u)\big)}_{\text{angular}}$$

$$+ \underbrace{\log\!\big(1 + \beta h(u) - \beta\,h(u)^2\,r(u)\big)}_{\text{radial}},$$

(15)

$$r(u) = \|u - z_0\|_2,$$

where $D$ is the dimensionality of the latent space and $z_0$ is a *trainable* $D$-dimensional vector (one per flow layer) representing the centre of the deformation.

Radial flows are invertible iff $\alpha > 0$ and $\beta > -\alpha$, we satisfy these by $\alpha = \mathrm{softplus}(\tilde{\alpha})$, $\beta = -\alpha + \mathrm{softplus}(\tilde{\beta})$ (Rezende & Mohamed, 2015) with unconstrained trainable parameters $\tilde{\alpha}, \tilde{\beta} \in \mathbb{R}$. We show our implementation in Algorithm 1.

**Penalizing TC.** We observed high dependence between latent coordinates in the aggregated posterior $\bar{q}_\phi(z)$ (both dimensions tended to encode $\rho$). Thus, we decompose the KL-divergence term of the ELBO following (Chen et al., 2018). For MatVAE, this allows us to directly penalize the total correlation $\mathrm{TC}(z) = \mathrm{KL}(\bar{q}_\phi(z) \| \prod_j \bar{q}_\phi(z_j))$ where $\bar{q}_\phi(z)$ is the aggregated posterior, $z_j$ is the $j \in \{1, 2\}$-th coordinate of the latent vector $z$. This allowed us to reduce the high dependence between latent coordinates, causing both dimensions to encode density. During training, we follow (Chen et al., 2018) and approximate the aggregated posterior $\bar{q}_\phi(z) = \mathbb{E}_{m \sim p_{data}}[q_\phi(z \mid m)]$ using samples from a mini-batch where $p_{data}$ is the empirical data distribution.

**Preventing Posterior Collapse.** For a fixed $m$, define per-dimension marginals $q_\phi(z_j \mid m)$ and
$$\mathrm{KL}_j(z \mid m) = \mathrm{KL}\left(q_\phi(z_j \mid m) \,\|\, p(z_j)\right).$$

(16)

Then, for each $m$,

$$\mathrm{KL}\left(q_\phi(z \mid m) \,\|\, p(z)\right) = \sum_j \mathrm{KL}_j(m)$$

$$+ \underbrace{\mathrm{TC}\left(q_\phi(z \mid m)\right)}_{\mathrm{KL}\left(q_\phi(z|m)\,\|\,\prod_j q_\phi(z_j|m)\right)}.$$

(17)

So the total KL contains a per-dimension rate term plus a per-sample total correlation. In the Gaussian (no-flow) case, $\mathrm{KL}_j(z \mid m) = \frac{1}{2}\left(\mu_j(m)^2 + \sigma_j(m)^2 - \log \sigma_j(m)^2 - 1\right)$, whose gradients drive $\mu_j \to 0$, $\sigma_j \to 1$ under posterior collapse. We therefore impose a capacity constraint ("free nats") on the dim-wise term to prevent collapse:

$$\sum_{j=1}^{d} \underbrace{\max\!\big(\delta,\ \mathbb{E}_{p_{\mathrm{data}}} \mathrm{KL}_j(z)\big)}_{\text{free-nats}},$$

(18)

---

**Algorithm 1** MatVAE posterior update with a radial normalizing flow.

**Require:**
      Batch $x \in \mathbb{R}^{B \times 3}$
      Encoder outputs $\mu(x) \in \mathbb{R}^{B \times D}$
      Encoder outputs $\log \sigma^2(x) \in \mathbb{R}^{B \times D}$
      Flow param $z_0 \in \mathbb{R}^{1 \times D}$
      Flow param $\log \alpha \in \mathbb{R}$
      Flow param $\beta_{\text{raw}} \in \mathbb{R}$
      Prior $p(z) = \mathcal{N}(0, I)$
**Ensure:**
      Flowed latent $z \in \mathbb{R}^{B \times D}$
      Post-flow log-density $\log q(z \mid x) \in \mathbb{R}^B$
      KL term denoted as KL
      Reconstruction loss $\mathcal{L}_{\text{recon}}$

1: ▷   Encode and reparameterize
2: $\mu \leftarrow \text{Encoder}_\mu(x)$
3: $\log \sigma^2 \leftarrow \text{Encoder}_{\log \sigma^2}(x)$
4: $\varepsilon \sim \mathcal{N}(0, I)$
5: $z_{\text{base}} \leftarrow \mu + \exp(\frac{1}{2} \log \sigma^2) \odot \varepsilon$
6:
7: ▷   Base posterior log-density
8: $\log q_0 \leftarrow \sum_{d=1}^{D} \log \mathcal{N}\big(z_{\text{base},d}; \mu_d, \exp(\log \sigma_d^2)\big)$
9:
10: ▷   Radial flow parameters ($\alpha > 0$, $\beta > -\alpha$)
11: $\alpha \leftarrow \text{softplus}(\log \alpha) + \varepsilon_\alpha$
12: $\beta \leftarrow -\alpha + \text{softplus}(\beta_{\text{raw}})$
13:
14: ▷   Radial flow transform
15: $\text{diff} \leftarrow z_{\text{base}} - z_0$
16: $r \leftarrow \|\text{diff}\|_2 + \varepsilon_r$
17: $h \leftarrow \frac{1}{(\alpha + r)}$
18: $z \leftarrow z_{\text{base}} + \beta \times h \times \text{diff}$
19:
20: ▷   Log-determinant of Jacobian
21: $bh \leftarrow \beta \times h$
22: $bh_{\text{stab}} \leftarrow \text{clamp}(bh, -c, c)$
23: $\text{term}_1 \leftarrow (D - 1) \cdot \log(1 + bh_{\text{stab}})$
24: $\text{term}_2 \leftarrow \log(1 + bh_{\text{stab}} - \beta h^2 r)$
25: $\Delta \log |J| \leftarrow \text{term}_1 + \text{term}_2$
26:
27: ▷   Change of variables and KL pieces
28: $\log q \leftarrow \log q_0 - \Delta \log |J|$
29: $\log p_z \leftarrow \sum_{d=1}^{D} \log \mathcal{N}(z_d; 0, 1)$
30:
31: ▷   Losses (non-relevant details shown as . . .)
32: $(\hat{E}, \hat{\nu}, \hat{\rho}, \ldots) \leftarrow \text{Decoder}(z)$
33: $\mathcal{L}_{\text{recon}} \leftarrow \text{MSE/NLL in transformed space } (\ldots)$
34: $\text{KL} \leftarrow \log q - \log p_z$
35: $\mathcal{L} \leftarrow \mathcal{L}_{\text{recon}} + \ldots$
36: **return** ($z$, $\log q$, KL, $\mathcal{L}_{\text{recon}}$)

---

which enforces a minimum information budget $\delta = 0.1$ per coordinate (zero subgradient below $\delta$). This allows us to fix the empirically observed imbalance where one latent carried most information and the other collapsed. An aggregated alternative consistent with the KL decomposition is $\max\big(\phi \cdot d, \sum_j \text{KL}(q_\phi(z_j) \| p(z_j))\big)$.

Table 14: **Training Hyperparameters.** We show the hyperparameters for the MatVAE and Geometry Transformer.

| MatVAE | | Geometry Transformer | |
|---|---|---|---|
| Training Precision | FP-32 | Training Precision | FP-16 |
| Hidden Width | 256 | Voxel Grid Resolution | $64^3$ |
| Network Depth | 3 ($\times 2$) | Input Channels | 1024 |
| Latent Dimensions | 2 | Model Channels | 768 |
| Dropout Rate | 0.05 | Latent Channels | 2 |
| Epochs | 850 | Transformer Blocks | 12 |
| Batch Size | 256 | Attention Heads | 12 |
| Optimizer | AdamW | MLP Ratio | 4 |
| Learning Rate | $10^{-4}$ | Attention Mode | Swin |
| Weight Decay | $10^{-4}$ | Window Size | 8 |
| LR Scheduler | Cosine Annealing | Max Training Steps | 200,000 |
| Final Learning Rate | $10^{-5}$ | Batch Size per GPU | 4 |
| Gradient Clipping | 5.0 | Total Batch Size | 16 |
| | $\alpha = 1.0$ (KL) | Optimizer | AdamW |
| $\beta$-TC Loss Weights | $\beta = 2.0$ (TC) | Learning Rate | $10^{-4}$ |
| | $\gamma = 1.0$ (MI) | Weight Decay | $5 \times 10^{-2}$ |
| Free Nats | 0.1 | Gradient Clipping | 1.0 |
| KL Annealing Epochs | 200 | Loss Function | $\ell_2$ |
| Data Normalization | Log Min-Max | EMA Rate | 0.9999 |

## F.2 NETWORK DESIGN

We now present our network architecture.

**MatVAE.** The *encoder* architecture begins by projecting the 3-dimensional material triplet through a linear transformation into a 256-dimensional hidden space, followed by SiLU activation. The resulting representation then passes through three "ResidualBlocks", each using a bottleneck design that compresses the 256-dimensional vector to 128 dimensions via LayerNorm and SiLU activation, applies another linear transformation, and restores the original dimensionality through a second LayerNorm-SiLU sequence. Each "ResidualBlock" maintains a skip connection that adds the input directly to the final output. The encoder finally has separate linear heads that project the processed representation into the latent space parameters: one head predicts the posterior mean $\mu_\phi(m)$ and another predicts the log-variance $\log \sigma_\phi^2(m)$ for the 2-dimensional latent code $z$.

The *decoder* mirrors this architecture in reverse, beginning with a linear projection from the 2-dimensional latent space back to the 256-dimensional hidden representation, followed by SiLU activation. The latent encoding then goes through three "ResidualBlocks" with an identical bottleneck structure and skip connections as the encoder. Finally, three separate linear heads decode the processed representation into the reconstructed material properties: Young's modulus, Poisson's ratio, and density, each predicted as scalar values in the normalized space.

**Geometry Transformer.** Our model is based on TRELLIS (Xiang et al., 2025). We use a transformer-based architecture specifically designed for processing sparse voxel representations with associated material properties. The model operates on a $64^3$ resolution voxel grid, accepting 1024-dimensional DINOv2 visual features as input and compressing them to a compact 2-dimensional latent representation through a 12-layer transformer backbone. Each transformer block utilizes 12 attention heads with a 4:1 MLP expansion ratio, using Swin attention with $8 \times 8$ local windows. During training, the Geometry Transformer operates in conjunction with a frozen MatVAE that decodes the latent into material properties.

## F.3 TRAINING

We present our voxelization scheme for training on meshes in Algorithms 2 and 3. We present the hyperparameters used for training MatVAE and Geometry Transformer in Tb. 14.

---

**Algorithm 2** Segment-aware volumetric voxelization for meshes.

---

**Require:**

Full-mesh vertices $V_{\text{all}} \in \mathbb{R}^{N \times 3}$, faces $F_{\text{all}}$

Segments $\mathcal{S} = \{(V_i \in \mathbb{R}^{N_i \times 3}, F_i, \text{sid}_i)\}_{i=1}^M$

Grid resolution $r \in \mathbb{N}$ (voxel pitch $h = 1/r$)

Per-segment cap $K_{\text{seg}} \in \mathbb{N}$

Global cap $K_{\text{all}} \in \mathbb{N}$

**Ensure:**

Combined voxel centers $C_{\text{all}} \in \mathbb{R}^{L \times 3}$ within $[-0.5, 0.5]^3$

Segment identifiers $\text{sid}_{\text{all}} \in \{\text{str}\}^L$

Discretized centers $\hat{C}_{\text{all}} \in \mathbb{R}^{L \times 3}$ on an $r^3$ grid

1: ▷   Global normalization from the full mesh
2: $v_{\min} \leftarrow \min(V_{\text{all}})$
3: $v_{\max} \leftarrow \max(V_{\text{all}})$
4: $c \leftarrow (v_{\min} + v_{\max})/2$
5: $s \leftarrow \max(v_{\max} - v_{\min})$
6: $\varepsilon \leftarrow 10^{-6}$
7:
8: $C_{\text{acc}} \leftarrow [\,]$
9: $\text{sid}_{\text{acc}} \leftarrow [\,]$
10: **for** $i = 1$ to $M$ **do**
11:    ▷   Normalize segment to $[-0.5, 0.5]^3$ and ensure triangles
12:    $V_i' \leftarrow \text{clip}((V_i - c)/s, \ -0.5 + \varepsilon, \ 0.5 - \varepsilon)$
13:    $F_i' \leftarrow \text{triangulate}(F_i)$
14:
15:    ▷   Voxelize segment and solid-fill (Algorithm 3)
16:    $(C_i, Y_i) \leftarrow \text{VOXELIZESOLID}(V_i', F_i', r)$
17:    **if** $K_{\text{seg}}$ is given and $|C_i| > K_{\text{seg}}$ **then**
18:       $I \leftarrow \text{choice}(|C_i|, K_{\text{seg}}, \text{without replacement})$
19:       $C_i \leftarrow C_i[I]$
20:    **if** $|C_i| = 0$ **then**
21:       **continue**
22:    $C_{\text{acc}}.\text{append}(C_i)$
23:    $\text{sid}_{\text{acc}}.\text{append}([\text{sid}_i]^{|C_i|})$
24:
25: **if** $|C_{\text{acc}}| = 0$ **then**
26:    **return** $(\varnothing, \varnothing, \varnothing)$
27: $C_{\text{all}} \leftarrow \text{concat}(C_{\text{acc}})$
28: $\text{sid}_{\text{all}} \leftarrow \text{concat}(\text{sid}_{\text{acc}})$
29:
30: ▷   Optional global subsampling
31: **if** $K_{\text{all}}$ is given and $|C_{\text{all}}| > K_{\text{all}}$ **then**
32:    $I \leftarrow \text{choice}(|C_{\text{all}}|, K_{\text{all}}, \text{without replacement})$
33:    $C_{\text{all}} \leftarrow C_{\text{all}}[I]$
34:    $\text{sid}_{\text{all}} \leftarrow \text{sid}_{\text{all}}[I]$
35:
36: ▷   Discretize to an $r^3$ grid aligned with $[-0.5, 0.5]^3$
37: $J \leftarrow \text{clip}(\lfloor(C_{\text{all}} + 0.5) \cdot r\rfloor, \ 0, \ r - 1)$
38: $\hat{C}_{\text{all}} \leftarrow J/r - 0.5$
39: **return** $(C_{\text{all}}, \text{sid}_{\text{all}}, \hat{C}_{\text{all}})$

---

**Voxelization For Training.**   Our training dataset contains Universal Scene Description (USD) files with multi-segment meshes. Each mesh is normalized to the range $[-0.5, 0.5]$ using a global bounding box computed across all segments to preserve relative spatial relationships. We use volumetric voxelization using a regular 3D grid with a resolution of $\frac{1}{64}$, where each voxel center is tested for interior containment within the mesh volume through point-in-polyhedron testing, followed by vol-

---

**Algorithm 3** Voxelization and flood fill primitives for meshes.

---

1: **procedure** VOXELIZESOLID($V$, $F$, $r$)
2:     ▷ Grid setup over a padded mesh AABB
3:     $h \leftarrow 1/r$
4:     $a_{\min} \leftarrow \min(V)$; $a_{\max} \leftarrow \max(V)$
5:     $b_{\min} \leftarrow a_{\min} - h$
6:     $b_{\max} \leftarrow a_{\max} + h$
7:     $n_x \leftarrow \lceil (b_{\max,x} - b_{\min,x})/h \rceil$
8:     $n_y \leftarrow \lceil (b_{\max,y} - b_{\min,y})/h \rceil$
9:     $n_z \leftarrow \lceil (b_{\max,z} - b_{\min,z})/h \rceil$
10:     $S[n_x, n_y, n_z] \leftarrow$ false
11:     $X[n_x, n_y, n_z] \leftarrow$ false
12:
13:     ▷ Triangle rasterization: mark surface cells
14:     **for** each triangle $t = (v_0, v_1, v_2) \in F$ **do**
15:        Compute triangle AABB $[t_{\min}, t_{\max}]$ in world coordinates
16:        Convert to grid index ranges $(i_{\min} : i_{\max}, j_{\min} : j_{\max}, k_{\min} : k_{\max})$
17:        **for** $i = i_{\min}$ to $i_{\max}$ **do**
18:           **for** $j = j_{\min}$ to $j_{\max}$ **do**
19:              **for** $k = k_{\min}$ to $k_{\max}$ **do**
20:                 Cell box $B = [b_{\min} + (i, j, k)h, \; b_{\min} + (i + 1, j + 1, k + 1)h]$
21:                 **if** TRIANGLEBOXINTERSECT($t$, $B$) **then**
22:                    $S[i, j, k] \leftarrow$ true
23:
24:     ▷ Exterior marking by flood fill on non-surface cells
25:     Initialize queue $Q$ with boundary indices $(i, j, k)$ where $S[i, j, k] =$ false
26:     **while** $Q$ not empty **do**
27:        $u \leftarrow Q.\text{pop}()$
28:        **if** $X[u]$ **then**
29:           **continue**
30:        $X[u] \leftarrow$ true
31:        **for** each 6-neighbor $v$ of $u$ within bounds **do**
32:           **if** $S[v] =$ false and $X[v] =$ false **then**
33:              $Q.\text{push}(v)$
34:
35:     ▷ Solid fill (interior) and center extraction
36:     $Y \leftarrow \neg X \wedge \neg S$
37:     $C \leftarrow [\,]$
38:     **for** all indices $(i, j, k)$ where $Y[i, j, k] =$ true **do**
39:        $c \leftarrow b_{\min} + (i + 0.5, \; j + 0.5, \; k + 0.5) \cdot h$
40:        $C.\text{append}(c)$
41:     **return** $(C, Y)$

---

umetric filling to generate solid voxel representations rather than surface-only discretizations. All the voxels inside a given segment receive the material properties of the segment they lie in.

**Rendering for Training.** For multi-view image rendering of meshes, we use a path-tracing renderer to produce photorealistic renderings of 3D objects. Camera viewpoints are sampled using a quasi-random Hammersley sequence distributed uniformly across a sphere. For training and testing, we render 150 views, though our method can work by rendering as many views as needed, with cameras positioned at a fixed radius of 2 units from the object center and configured with a 40-degree field of view. Images are rendered at $512 \times 512$ pixel resolution. The rendering pipeline outputs both the RGB images and the corresponding camera extrinsics and intrinsics. For rendering splats, we simply replace the renderer with the 3D Gaussian Splat renderer (Kerbl et al., 2023) in our workflow. For rendering SDFs, we render meshes using many points collected from the SDF. For rendering NeRFs (Mildenhall et al., 2020), we simply replace the renderer with nerfstudio (Tancik et al., 2023) in our workflow.

**Feature Aggregation.** For visual feature extraction, we employ DINOv2-ViT-L/14 (Oquab et al., 2024) with registers. We use a patch size of 14×14 pixels and process input images resized to 518×518 pixels, resulting in a 37×37 patch. We use the `nv-dinov2`[2] (NVIDIA, 2025) implementation.

### F.4 SIMULATION AND RENDERING

For our mesh simulations (Fig. 5), we simulate with the finite-element method (FEM) using the `libuipc` (Huang et al., 2025; 2024a) implementation, and we render the simulations in a path-tracing-based renderer. While comparing with other simulators (Fig. 2) we use MPM (Sulsky et al., 1994) using `taichi-mpm` (Hu & contributors, 2018), XPBD (Macklin et al., 2016) using `PositionBasedDynamics` (Bender & contributors, 2015), and FEM using Warp (Macklin, 2022).

For our large-scale splat simulations or splat + mesh simulations, we use Simplicits (Modi et al., 2024) using the sparse simplicits implementation using Kaolin (Modi et al., 2024; Fuji Tsang et al.). For rendering our large-scale splat simulations or splat + mesh simulations, we use Polyscope (Sharp et al., 2019) and composite splat renders from `gsplat` (Ye et al., 2024). For these simulations, we apply material property tolerances to reduce numerical noise: voxels with Young's modulus differing by less than $10^1$ Pa, Poisson's ratio by less than $10^{-3}$, or density by less than $10^1$ kg/m³ are assigned identical values for the respective property. We present additional details for deforming and rendering deformed Gaussian Splats in Appendix G.5.

### F.5 BASELINES

**Converting Hardness to Young's Modulus.** NeRF2Physics (Zhai et al., 2024) does not estimate a numerical value of Young's Modulus, but instead predicts Shore A-Shore D hardness. Thus, to compare our method with NeRF2Physics (Zhai et al., 2024) we convert these Shore hardness values to average Young's Modulus values.

**Shore A.** For Shore A hardness, we follow (ASTM International, 2015) and use:
$$E_{\text{MPa}} = e^{(S_A \times 0.0235) - 0.6403} \tag{19}$$
where $S_A$ is the Shore A hardness value and $E_{\text{MPa}}$ is Young's modulus in megapascals.

**Shore D.** For Shore D hardness, we follow (ASTM International, 2015) and use:
$$E_{\text{MPa}} = e^{((S_D + 50) \times 0.0235) - 0.6403} \tag{20}$$
where $S_D$ is the Shore D hardness value and $E_{\text{MPa}}$ is Young's modulus in megapascals.

**Point or Voxel Sampling.** The baselines NeRF2Physics (Zhai et al., 2024) and PUGS (Shuai et al., 2025) in their methods sample points from the NeRF or Gaussian splat, respectively, and predict mechanical properties at those points. To ensure fair comparisons in Tb. 4 and Fig. 6b, we explicitly make these methods work on the same set of points in the object on which our method is evaluated.

**Implementation details of Baselines.** The baseline NeRF2Physics (Zhai et al., 2024) uses `gpt-3.5-turbo` for certain parts of their pipeline. We replace `gpt-3.5-turbo` in their pipeline with a better performing model, GPT-4o (OpenAI & et al., 2024). The baseline Phys4DGen (Liu et al., 2024b) does not have code available. Thus, we faithfully reproduce the parts, "Material Grouping and Internal Discovery" and "MLLMs-Guided Material Identification". We reproduce these parts of their pipeline using GPT-4o (OpenAI & et al., 2024) for the MLLMs-Guided Material Identification. Furthermore, we obtained the prompts from the authors of Phys4DGen (Liu et al., 2024b) and use the same prompts.

## G ADDITIONAL DETAILS ON THE SIMULATIONS

We experiment with Simplicits (Modi et al., 2024), a reduced-order simulator (Fig. 1, 5, 8c and 8e) and an accurate finite-element method (FEM) simulator (Fig. 1, 5 and 8b) with our material prop-

---

[2] https://build.nvidia.com/nvidia/nv-dinov2

Table 15: Hyperparameters for FEM simulation.

| Hyperparameter | Value | Hyperparameter | Value |
|---|---|---|---|
| Time Integrator | Backward Euler | Linear Solver | pre-conditioned CG |
| Nonlinear Solver | Newton's w/ line search | Linear tolerance | $10^{-3}$ |
| Newton max iters. | 1024 | Line search | |
| Velocity tol. | $0.05\ ms^{-1}$ | max iters | 8 |
| CCD tol. | 1.0 | Collision | |
| Transform rate tol. | 0.1/s | Friction | 0.5 |
| $dt$ | 0.02 | Contact Resistance | 1.0 |
| Gravity | $[0.0, -9.8, 0.0]$ | $\hat{d}$ | 0.01 |

erties. We also use a FEM simulator for our experiments on interpreting errors in properties (Appendix D.4). We use a material point method (MPM) (Sulsky et al., 1994), and an Extended Position Based Dynamics (XPBD) (Macklin et al., 2016) simulator for our experiments to compare between simulators (Fig. 2). We share details on these simulations. We also share details on the interpolation we use across all our simulations. We share the hyperparameters used for all the FEM simulations in Tb. 15.

### G.1 INTERPOLATION SCHEME

Our simulations receive a material field sampled on a voxel grid predicted by VoMP, i.e., values $m(\mathbf{X}_i)$ given at lattice points $\{\mathbf{X}_i\} \subset \Omega$. When the simulator needs material values at arbitrary query locations $\mathbf{X}$ (e.g., element centroids or vertices), we evaluate a nearest-neighbour interpolation of the voxel field:

$$i^*(\mathbf{X}) = \arg\min_i \|\mathbf{X} - \mathbf{X}_i\|_2,$$

$$m^*(\mathbf{X}) = m\big(\mathbf{X}_{i^*(\mathbf{X})}\big). \tag{21}$$

We intentionally avoid higher-order interpolation of material fields since real objects are piecewise-constant across label regions, and convex blending across parts of the objects invents intermediate materials. These intermediate materials might not be physically present or admissible, while our outputs fall into a valid material due to the MatVAE (§3). Nearest-neighbour preserves sharp interfaces and is usually robust for arbitrary query locations.

### G.2 PREPARING SCENES AND ASSIGNING MATERIALS FOR THE FEM SOLVER

Mechanical properties are set either uniformly (like in Appendix D.4) or heterogeneously from a voxel field. For uniform assignment, given $E$ and $\nu$ we compute Lamé parameters

$$\lambda = \frac{E\,\nu}{(1+\nu)(1-2\nu)}, \qquad \mu = \frac{E}{2(1+\nu)}, \tag{22}$$

which are used elementwise together with a constant mass density $\rho$.

For heterogeneous assignment, a voxel lattice provides $E(\mathbf{X})$, $\nu(\mathbf{X})$, and $\rho(\mathbf{X})$ at voxel centers. After applying the same rigid/scale transform as the mesh, each tetrahedron takes $\lambda, \mu$ from the nearest voxel to its centroid, and each vertex takes $\rho$ from the nearest voxel to its position. This produces per-tetrahedron $\lambda, \mu$ and per-vertex $\rho$ fields that are directly used in the elastic strain energy density per unit reference volume ($W$), first variation of the incremental potential ($R$), and Newton-Jacobian ($\mathcal{K}$).

During simulation, a visual mesh is embedded into the physics mesh by assigning each visual vertex $\mathbf{x}_v$ to a containing (or nearest) tetrahedron with vertices $\{\mathbf{X}_a\}_{a=1}^4$ and barycentric weights $\{w_a\}_{a=1}^4$ satisfying $\sum_a w_a = 1$ and $\sum_a w_a \mathbf{X}_a = \mathbf{x}_v$; its deformed position is then the barycentric interpolation of current nodal positions $\{\mathbf{x}_a\}_{a=1}^4$:

$$\mathbf{x}_v^{\mathrm{def}} = \sum_{a=1}^4 w_a\, \mathbf{x}_a. \tag{23}$$

The state update in our simulation experiments is computed time-step by time-step, and we also deform and move the visual mesh according to the physics mesh at each time step.

### G.3 DETAILS OF THE FEM SOLVER

For FEM simulations, we use a simulator based on the `libuipc` (Huang et al., 2025; 2024a) implementation and the Warp (`warp.fem`) (Macklin, 2022) implementation. We first explain the details for our simulations in §D.4.

We consider a deformable continuum body with reference configuration $\Omega \subset \mathbb{R}^3$ and boundary $\partial\Omega = \Gamma_D \cup \Gamma_N$, where $\Gamma_D$ denotes boundary points with Dirichlet boundary conditions, and $\Gamma_N$ denotes boundary points with Neumann boundary conditions. The unknown to solve for is the displacement field $\mathbf{u} : \Omega \to \mathbb{R}^3$. Time is discretized into frames with a fixed step $\Delta t$. At each frame we compute an increment $\Delta\mathbf{u}$ that advances the configuration $\mathbf{u} \leftarrow \mathbf{u} + \Delta\mathbf{u}$ while enforcing Dirichlet constraints on $\Gamma_D$. The deformation map is $\varphi(\mathbf{X}) = \mathbf{X} + \mathbf{u}(\mathbf{X})$, with deformation gradient $\mathbf{F}(\mathbf{u}) = \mathbf{I} + \nabla\mathbf{u}$, Jacobian $J = \det \mathbf{F}$, and isochoric invariant $I_c = \mathrm{tr}(\mathbf{F}^\top\mathbf{F})$. For corotational modeling, we use the stretch tensor $\mathbf{S}$ from the polar/SVD decomposition: if $\mathbf{F} = \mathbf{U}\,\mathrm{diag}(\boldsymbol{\sigma})\,\mathbf{V}^\top$ then $\mathbf{S} = \mathbf{V}\,\mathrm{diag}(\boldsymbol{\sigma})\,\mathbf{V}^\top$. Given Young's modulus $E$ and Poisson ratio $\nu$, the Lamé parameters are $\lambda = E\nu/((1+\nu)(1-2\nu))$ and $\mu = E/(2(1+\nu))$. Here $\nabla$ denotes the gradient with respect to reference coordinates, $\mathbf{A}{:}\mathbf{B} = \mathrm{tr}(\mathbf{A}^\top\mathbf{B})$ is the Frobenius inner product, and $\|\cdot\|$ is the Euclidean norm.

The elastic response we use is the corotational Hookean model. Define the small strain $\boldsymbol{\varepsilon} = \mathbf{S} - \mathbf{I}$. The strain energy density and Kirchhoff stress are

$$W_{\mathrm{CR}}(\mathbf{S}) = \underbrace{\mu\,\boldsymbol{\varepsilon}{:}\boldsymbol{\varepsilon}}_{\text{shear (deviatoric)}} + \underbrace{\tfrac{\lambda}{2}\,\mathrm{tr}(\boldsymbol{\varepsilon})^2}_{\text{volumetric}},$$
$$\boldsymbol{\tau}(\mathbf{S}) = \underbrace{2\mu\,\boldsymbol{\varepsilon}}_{\text{shear}} + \underbrace{\lambda\,\mathrm{tr}(\boldsymbol{\varepsilon})\,\mathbf{I}}_{\text{volumetric}}, \tag{24}$$

with a consistent linearization obtained via the variation of $\mathbf{S}$ with respect to $\mathbf{F}$ and projected to maintain symmetry and positive semidefiniteness.

Each frame solves an incremental variational problem. Given the previous increment $\Delta\mathbf{u}^{n-1}$, we seek $\Delta\mathbf{u}^n$ that approximately minimizes the incremental potential

$$\Pi(\Delta\mathbf{u}) = \underbrace{\int_\Omega \rho\left(\tfrac{1}{2}\,\frac{\|\Delta\mathbf{u} - \Delta\mathbf{u}^{n-1}\|^2}{\Delta t^2}\right)\mathrm{d}V}_{\text{inertial regularization}}$$

$$+ \underbrace{\int_\Omega \left(-\rho\,\mathbf{g}\cdot\Delta\mathbf{u} - \mathbf{f}_{\mathrm{ext}}\cdot\Delta\mathbf{u}\right)\mathrm{d}V}_{\text{body and external work}}$$

$$+ \underbrace{\int_\Omega W_{\mathrm{CR}}\big(\mathbf{S}(\mathbf{u}^{n-1} + \Delta\mathbf{u})\big)\,\mathrm{d}V}_{\text{elastic energy}}$$

$$+ \underbrace{\Pi_{\mathrm{int}}(\Delta\mathbf{u})}_{\text{interior/boundary regularization}}, \tag{25}$$

where $\rho$ is the mass density, $\mathbf{g}$ is the gravitational acceleration vector, $\mathbf{f}_{\mathrm{ext}}$ denotes prescribed volumetric loads, and $\Pi_{\mathrm{int}}$ denotes any interior/boundary regularization term (e.g., a jump penalty in discontinuous settings). The admissible test function $\mathbf{v}$ is any sufficiently smooth virtual displacement that vanishes on $\Gamma_D$. The first variation $\delta\Pi(\Delta\mathbf{u}; \mathbf{v}) = 0$ for all such $\mathbf{v}$ yields the residual

functional

$$
R(\Delta\mathbf{u})[\mathbf{v}] = \underbrace{\int_{\Omega} \rho\, \frac{\Delta\mathbf{u} - \Delta\mathbf{u}^{n-1}}{\Delta t^2} \cdot \mathbf{v}\, \mathrm{d}V}_{\text{inertia}}
$$

$$
+ \underbrace{\int_{\Omega} \big( -\rho\, \mathbf{g} \cdot \mathbf{v} - \mathbf{f}_{\text{ext}} \cdot \mathbf{v} \big)\, \mathrm{d}V}_{\text{body/external}} \tag{26}
$$

$$
- \underbrace{\int_{\Omega} \boldsymbol{\tau}\big(\mathbf{S}(\mathbf{u}^{n-1} + \Delta\mathbf{u})\big) : \nabla\mathbf{v}\, \mathrm{d}V}_{\text{elastic (internal) virtual work}}
$$

$$
+ \underbrace{R_{\text{int}}(\Delta\mathbf{u})[\mathbf{v}]}_{\text{regularization}},
$$

which is set to zero for all $\mathbf{v}$. Newton's method is applied to $R(\Delta\mathbf{u}) = 0$. At iterate $\Delta\mathbf{u}^{(k)}$ we assemble the consistent tangent operator $\mathcal{K} = \mathrm{D}R\big[\Delta\mathbf{u}^{(k)}\big]$ (the Gâteaux derivative of $R$) and solve the linear system

$$
\mathcal{K}\, \delta\mathbf{u} = -R\big(\Delta\mathbf{u}^{(k)}\big),
$$
$$
\Delta\mathbf{u}^{(k+1)} = \Delta\mathbf{u}^{(k)} + \alpha\, \delta\mathbf{u} \tag{27}
$$

where $\alpha \in (0, 1]$ is chosen by a backtracking Armijo rule to guarantee sufficient decrease of $\Pi$. The operator $\mathcal{K}$ contains an inertial mass-like term $\int_{\Omega} \rho\, \Delta t^{-2}\, \delta\mathbf{u} \cdot \mathbf{v}\, \mathrm{d}V$, the consistent elastic tangent from the linearization of $\boldsymbol{\tau}(\mathbf{S}(\cdot))$, and any interior/boundary penalty contributions. This procedure is repeated until the update norm or residual falls below a prescribed tolerance.

For all our other simulations (*i.e.* except the simulations in §D.4) we use a closely related variant whose differences are in the constitutive law, material assignment, mesh preparation/interpolation, and contact handling. First, the stored energy and stress are taken to be compressible Neo-Hookean with volumetric regularization. Writing $\mathbf{C} = \mathbf{F}^{\top}\mathbf{F}$ and $\mathbf{B} = \mathbf{F}\,\mathbf{F}^{\top}$, the energy and Kirchhoff stress are

$$
W_{\text{NH}}(\mathbf{F}) = \tfrac{\mu}{2}\big(\operatorname{tr}\mathbf{C} - 3 - 2\ln J\big) + \tfrac{\lambda}{2}(\ln J)^2,
$$
$$
\boldsymbol{\tau}_{\text{NH}}(\mathbf{F}) = \mu\,(\mathbf{B} - \mathbf{I}) + \lambda\,\ln J\,\mathbf{I}. \tag{28}
$$

This change only affects the elastic terms in $\Pi$, $R$, and $\mathcal{K}$; the kinematics and inertial terms remain the same. For the simulation experiments, we also use IPC (Li et al., 2020a) for collision handling.

### G.4 PREPARING SCENES AND ASSIGNING MATERIALS FOR THE SIMPLICITS SOLVER

Each object is specified by a set of quadrature points $\mathcal{Q} = \{\mathbf{X}_q\}$ that sample its volume (used for elasticity and inertia), a set of collision particles $\mathcal{C} = \{\mathbf{X}_c\}$ for contact, and a set of visual vertices for rendering. We position objects with a rigid transform (origin and rotation) and an object scale; these transforms are applied consistently when evaluating kinematics, gravity, and material fields.

We embed all objects into a regular grid domain and attach to this grid a low-dimensional Simplicits subspace. The displacement basis is the product of a trilinear grid shape and a per-object handle shape, with multiple duplicated handles per grid vertex. At each quadrature point, we evaluate and cache per-node subspace weights and their spatial gradients. These weights modulate the duplicated handle functions during assembly, instantiating the Simplicits subspace on the grid.

Material parameters are assigned per quadrature point from a voxel lattice providing $E(\mathbf{X})$, $\nu(\mathbf{X})$, and $\rho(\mathbf{X})$. After applying the same rigid/scale transform as the object, each quadrature location $\mathbf{X}_q$ takes its material from the nearest voxel to $\mathbf{X}_q$. We compute Lamé parameters per point as

$$
\lambda_q = \frac{E(\mathbf{X}_q)\,\nu(\mathbf{X}_q)}{(1 + \nu(\mathbf{X}_q))(1 - 2\nu(\mathbf{X}_q))},
$$
$$
\mu_q = \frac{E(\mathbf{X}_q)}{2(1 + \nu(\mathbf{X}_q))}, \tag{29}
$$

and use $\rho_q = \rho(\mathbf{X}_q)$ in the inertial terms.

The quadrature weights are set uniformly as

$$w_q = \frac{v}{|\mathcal{Q}|},$$
(30)

where $v$ is the object volume estimate and $|\mathcal{Q}|$ is the number of quadrature points. This makes elastic and inertial energies invariant to the sampling density.

Collision particles $\mathcal{C}$ are used for detecting and resolving contact against other particles and registered kinematic triangle meshes (containers and obstacles). We use an IPC-style barrier with Coulomb friction, and scale the contact stiffness by object volume and the number of collision particles to obtain comparable penalties across scenes.

### G.5  Deforming Splats and Rendering Deformed Splats

We render each object as a set of anisotropic Gaussian splats. At rest, a splat is parameterized by its mean $\boldsymbol{\mu}_0 \in \mathbb{R}^3$, a unit quaternion $\mathbf{q}_0$ (with rotation $\mathbf{R}_0 \in \mathrm{SO}(3)$), and axis scales $\mathbf{s}_0 \in \mathbb{R}^3_{>0}$. We define the rest-frame shape operator

$$\underbrace{\mathbf{L}}_{\text{rest anisotropy}} = \mathbf{R}_0 \operatorname{diag}(\mathbf{s}_0).$$
(31)

During simulation, the displacement field yields a world-space deformation gradient $\underbrace{\mathbf{F}}_{\text{local deformation}}$ at the splat center and a world-space position $\boldsymbol{\mu}$ (obtained by evaluating the embedded deformation at the visual vertex). We map the rest anisotropy through the local deformation to obtain the world-space covariance of the splat as

$$\underbrace{\Sigma}_{\text{world covariance}} = \underbrace{(\mathbf{F}\,\mathbf{L})}_{\text{deformed axes}} \overbrace{(\mathbf{F}\,\mathbf{L})^\top}^{\text{deformed axes}^\top} + \underbrace{\varepsilon\,\mathbf{I}}_{\text{SPD padding}}.$$
(32)

Here $\varepsilon > 0$ is a small scalar that guarantees positive-definiteness under extreme compression.

For rasterization we pass $\boldsymbol{\mu}$ and $\Sigma$ to the Gaussian renderer. The renderer (`gsplat` (Ye et al., 2024)) expects a symmetric 6-vector parameterization; we therefore pack the lower-triangular entries as

$$\underbrace{\mathbf{c}}_{\text{packed covariance}} = \begin{bmatrix} \Sigma_{11}, \ \Sigma_{12}, \ \Sigma_{13}, \ \Sigma_{22}, \ \Sigma_{23}, \ \Sigma_{33} \end{bmatrix}^\top.$$
(33)

Color appearance (spherical-harmonic coefficients) and opacity are carried from the rest representation; only the mean $\boldsymbol{\mu}$ and covariance $\Sigma$ change over time. We also support a scalar scale multiplier applied to $\mathbf{s}_0$ for interactive control in qualitative visualizations.

Given the view (extrinsic) matrix $\mathbf{V}$ and vertical field-of-view, we synthesize camera intrinsics

$$\mathbf{K} = \begin{bmatrix} f_x & 0 & c_x \\ 0 & f_y & c_y \\ 0 & 0 & 1 \end{bmatrix},$$
$$f_x = \frac{W}{2\tan(\frac{\mathrm{fov}_x}{2})}, \ f_y = \frac{H}{2\tan(\frac{\mathrm{fov}_y}{2})},$$
$$c_x = \frac{W}{2}, \ c_y = \frac{H}{2},$$
(34)

with image size $(W, H)$. We then render the set of splats $\{\boldsymbol{\mu}, \mathbf{c}\}$ under $(\mathbf{V}, \mathbf{K})$ to produce RGB (and depth) frames. At every frame, we interpolate $\boldsymbol{\mu}$ and $\mathbf{F}$ at the visual vertices, form $\Sigma = \mathbf{F}\,\mathbf{L}\,\mathbf{L}^\top\mathbf{F}^\top + \varepsilon\,\mathbf{I}$, pack it as $\mathbf{c}$, and feed the Gaussian rasterizer together with the stored colors and opacities.

### G.6  Details of the Simplicits Solver

We use the Simplicits (Modi et al., 2024) simulator based on the implementation in Kaolin (Fuji Tsang et al.). Simplicits solves for a displacement field represented in a low-dimensional subspace attached to a regular grid. This subspace is a product basis between trilinear grid polynomials and duplicated per-vertex "handle" functions whose influence is modulated by per-point weights and weight gradients evaluated at quadrature points. We assemble inertia and compressible Neo-Hookean elasticity on this subspace, using the per-quadrature Lamé parameters

$(\lambda_q, \mu_q)$ and measures $w_q$. Each frame performs Newton steps on the incremental potential with backtracking line search. Linear systems are solved by preconditioned conjugate gradients.

*In the Kaolin implementation*, splat–splat contact uses particle pairs: for a pair $(a, b)$ with current positions $\mathbf{x}_a, \mathbf{x}_b$ and contact radius $r$, we set $\mathbf{n}_c = (\mathbf{x}_a - \mathbf{x}_b)/\|\mathbf{x}_a - \mathbf{x}_b\|$, $r_c = r$, and use the relative offset $\mathbf{o}_c(\Delta \mathbf{u}) = \Delta \mathbf{u}_a - \Delta \mathbf{u}_b$, so that $d_c = \mathbf{n}_c \cdot (\mathbf{x}_a - \mathbf{x}_b)$ and $\mathbf{v}_{t,c}$ measures tangential slip between the two.

*However, in our implementation*, splat–mesh contact is handled differently than splat-splat contact. For splat-mesh contact, instead of using the collision points, we use triangle meshes as kinematic colliders: for a particle at $\mathbf{x}$ and its closest point $\mathbf{p}$ on a nearby triangle (with interpolated mesh normal/velocity $\mathbf{n}_c, \mathbf{v}_m$), we take $r_c = 2r$ and

$$\mathbf{o}_c(\Delta \mathbf{u}) = \Delta \mathbf{u} + (\|\mathbf{x} - \mathbf{p}\| - \mathbf{n}_c \cdot \Delta \mathbf{u}) \mathbf{n}_c - \mathbf{v}_m, \qquad d_c = \|\mathbf{x} - \mathbf{p}\| - \mathbf{n}_c \cdot \mathbf{v}_m. \tag{35}$$

Only the simulated-object DOFs enter $\mathbf{o}_c$; mesh or splat motion appears through $\mathbf{v}_m$.

These terms are used in the Newton system as

$$\mathbf{K} \leftarrow \mathbf{K} + \underbrace{\alpha \mathbf{H}^\top \mathbf{C} \mathbf{H}}_{\text{contact stiffness}}, \qquad \mathbf{r} \leftarrow \mathbf{r} - \underbrace{\alpha \mathbf{H}^\top \mathbf{g}}_{\text{contact force}}, \tag{36}$$

where $\mathbf{H}$ is the Jacobian of contact offsets, and $\mathbf{g}, \mathbf{C}$ are the per-contact gradient/Hessian with respect to $\mathbf{o}_c$. For splat–mesh contacts only the simulated-object block of $\mathbf{H}$ is present; for splat–splat contacts the two object blocks appear with opposite signs.

### G.7 PREPARING SCENES AND ASSIGNING MATERIALS FOR THE XPBD SOLVER

We also use an Extended Position-Based Dynamics (XPBD) solver (Macklin et al., 2016) based on `PositionBasedDynamics` (Bender & contributors, 2015).

**Particles and initialization.** Objects are represented by particles positioned at rest locations $\{\mathbf{X}_i\}$. For each particle we initialize position $\mathbf{x}_i \leftarrow \mathbf{X}_i$ and a previous position equal to $\mathbf{X}_i$ (Verlet), set mass $m_i$ and inverse mass $w_i = 1/m_i$ (pinned points use $w_i = 0$). A soft sphere uses one center particle and a set of surface particles sampled on a UV sphere; the center forms simple tetrahedra with nearby surface points for volume preservation.

**Material parameters and compliance.** To target an elastic behavior with Young's modulus $E$ and Poisson ratio $\nu$ (bulk modulus $K = E/(3(1 - 2\nu))$), we choose distance- and volume-constraint compliances $\alpha_{\text{dist}}, \alpha_{\text{vol}}$ inversely proportional to $E$ and $K$. XPBD uses the scaled compliance $\alpha_\Delta = \alpha/\Delta t^2$ so that smaller $\alpha$ yields stiffer response.

**Prediction, projections, and updates.** Each frame predicts positions with Verlet integration, then iteratively projects distance, volume, and collision constraints. For a pairwise distance constraint, the XPBD update displaces endpoints along the edge with a factor

$$\gamma = \frac{\alpha_\Delta}{\alpha_\Delta + w_i + w_j}, \quad \Delta \mathbf{x}_i \propto -\gamma \, C_{ij} \frac{\mathbf{x}_i - \mathbf{x}_j}{\|\mathbf{x}_i - \mathbf{x}_j\|}, \quad \Delta \mathbf{x}_j = -\frac{w_i}{w_j} \Delta \mathbf{x}_i, \tag{37}$$

with the analogous formulation for volume constraints using their gradients. After projections, we update positions and apply ground-plane contact with Coulomb friction.

**Visualization.** We track the evolving surface by updating a triangular mesh whose vertices coincide with the surface particles.

### G.8 PREPARING SCENES AND ASSIGNING MATERIALS FOR THE MPM SOLVER

We use a material point method (MPM) simulator (Sulsky et al., 1994) based on `taichi-mpm` (Hu & contributors, 2018). Scenes are specified by a uniform Cartesian grid, a set of particles sampling each object's volume, and per-object mechanical properties.

**Domain, grid, and timestep.** We embed all objects in a unit cube domain discretized by an $n_{\text{grid}} \times n_{\text{grid}} \times n_{\text{grid}}$ grid (cell size $\Delta x = 1/n_{\text{grid}}$). We use a fixed time step $\Delta t$ small enough for stability (e.g., $\Delta t = 5 \times 10^{-5}$ in our experiments).

**Particles and initialization.** Each object is sampled with material points (particles) positioned at rest locations $\{\mathbf{X}_p\}$. For each particle we initialize position $\mathbf{x}_p \leftarrow \mathbf{X}_p$, velocity $\mathbf{v}_p \leftarrow \mathbf{0}$, affine velocity field $\mathbf{C}_p \leftarrow \mathbf{0}$, and deformation gradient $\mathbf{F}_p \leftarrow \mathbf{I}$. Mass is set as $m_p = \rho V_p$ with density $\rho$ and particle volume $V_p$ consistent with the grid resolution. In the simple drop test, we sample a sphere at a height and let gravity act.

**Material parameters and constitutive law.** We assign per-object Young's modulus $E$ and Poisson ratio $\nu$, compute Lamé parameters $\mu = E/(2(1 + \nu))$ and $\lambda = E\nu/((1 + \nu)(1 - 2\nu))$, and use a fast corotated (FCR) elastic model. With polar/SVD decomposition $\mathbf{F} = \mathbf{U}\operatorname{diag}(\boldsymbol{\sigma})\mathbf{V}^\top$ and rotation $\mathbf{R} = \mathbf{U}\mathbf{V}^\top$, the Kirchhoff stress is

$$\boldsymbol{\tau}_{\mathrm{FCR}}(\mathbf{F}) \;=\; 2\mu\,(\mathbf{F} - \mathbf{R})\,\mathbf{F}^\top \;+\; \lambda\,J\,(J - 1)\,\mathbf{I}, \qquad J = \det \mathbf{F}. \tag{38}$$

**Transfers and updates.** Each step performs Particle-to-Grid (P2G) transfers using quadratic B-spline weights: we scatter particle mass, momentum, and internal forces $-V_p\,\boldsymbol{\tau}\,\nabla w$ to grid nodes. On the grid, we (i) convert momentum to velocity, (ii) add gravity, and (iii) enforce box boundary conditions by clamping outward normal velocities to zero. We then perform Grid-to-Particle (G2P) to interpolate grid velocities back to particles, update the affine field, and integrate

$$\mathbf{x}_p \leftarrow \mathbf{x}_p + \Delta t\,\mathbf{v}_p, \qquad \mathbf{F}_p \leftarrow (\mathbf{I} + \Delta t\,\nabla\mathbf{v})\,\mathbf{F}_p. \tag{39}$$

**Visualization.** We embed a coarse surface at rest, align it to the particle center of mass, and update its vertices via interpolation of nearby particle displacements.

## H    OTHER RELATED WORKS

For completeness, we include other tangentially related works here. A different setting from ours is inferring physical properties given additional observations, such as video (Davis et al., 2015; Mottaghi et al., 2016; Bhat et al., 2002; Chen et al., 2025b; Liu et al., 2024a; Xue et al., 2023; Li et al., 2025; Brubaker et al., 2009; Yildirim et al., 2016; Li et al., 2023; 2020b; Wu et al., 2016; 2015; 2017; Xia et al., 2024; Xu et al., 2019; Feng et al., 2024; Lin et al., 2024b) or physical manipulation of real objects (Yu et al., 2024; Pai et al., 2001; Lang et al., 2003; Lloyd & Pai, 2001; Pai et al., 2008; Pai, 2000; Yao & Hauser, 2023; Pinto et al., 2016). Other related works focus on generating new physically plausible shapes, e.g. stable under gravity or other interactions, but cannot augment existing 3D assets with mechanical properties, which is our goal (Lin et al., 2025b; Guo et al., 2024; Chen et al., 2024; Ni et al., 2024; Yang et al., 2024; Mezghanni et al., 2022; Chen et al., 2025a; Cao et al., 2025; Cao & Kalogerakis, 2025). Other methods predict displacements (Zhang et al., 2024; Shi et al., 2023), bypassing mechanical properties, or focus on other aspects such as articulation (Xia et al., 2025).

**User Prompt.**

You are a materials science expert analyzing two images:
1. A photo of the full object (showing how the material appears in context).
2. A sphere with the material's texture (showing color/roughness/reflectivity in isolation).

Using both images and the information below, identify the real-world material and estimate its mechanical properties.

Material context:
∗Material type: `fill in from dataset`
∗Opacity: `fill in from dataset`
∗Density: `fill in from dataset` kg/m^3
∗Dynamic friction: `fill in from dataset`
∗Static friction: `fill in from dataset`
∗Restitution: `fill in from dataset`
∗Usage: `fill in from dataset`

Your task is to provide three specific properties:
1. Young's modulus (in Pa using scientific notation)
2. Poisson's ratio (a value between 0.0 and 0.5)
3. Density (in kg/m^3 using scientific notation)

Additional reference material property ranges to help you make accurate estimations:
- `closest match from material database` : Young's modulus range `fill in` , Poisson's ratio range `fill in` , Density range `fill in` kg/m^3
- `closest match from material database` : Young's modulus range `fill in` , Poisson's ratio range `fill in` , Density range `fill in` kg/m^3
- `closest match from material database` : Young's modulus range `fill in` GPa, Poisson's ratio range `fill in` , Density range `fill in` kg/m^3

Example 1:
Material: metal
Opacity: opaque
Density: 7800 kg/m^3
Dynamic friction: 0.3
Static friction: 0.4
Restitution: 0.3
Usage: structural component

Analysis:
Step 1: Based on the images, this appears to be a standard structural steel with a matte gray finish.
Step 2: The surface has medium roughness with some subtle texture visible in the reflection pattern.
Step 3: The physical properties (density, friction values, restitution) are consistent with carbon steel.
Step 4: Considering the usage and measured properties:
- High stiffness (Young's modulus 200 GPa) based on typical steel values
- Medium Poisson's ratio typical of metals
- High density matching the measured 7800 kg/m^3

Young's modulus: 2.0e11 Pa
Poisson's ratio: 0.29
Density: 7800 kg/m^3

Figure 23: **User Prompt I.** The User Prompt we use for every segment of every object.

**User Prompt Continued.**

Example 2:
Material: plastic
Opacity: opaque
Density: 950 kg/m^3
Dynamic friction: 0.25
Static friction: 0.35
Restitution: 0.6
Usage: household container

Analysis:
Step 1: The material shows the characteristic smooth, uniform appearance of a consumer plastic.
Step 2: It has moderate gloss with some translucency and a slight texture.
Step 3: The physical properties (medium-low density, moderate friction, higher restitution) match polypropylene.
Step 4: Based on these observations and measurements:
- Medium-low stiffness typical of polyolefin plastics
- Higher Poisson's ratio indicating good lateral deformation
- Density matching the measured 950 kg/m^3

Young's modulus: 1.3e9 Pa
Poisson's ratio: 0.42
Density: 950 kg/m^3

Example 3:
Material: fabric
Opacity: opaque
Density: 300 kg/m^3
Dynamic friction: 0.55
Static friction: 0.75
Restitution: 0.2
Usage: furniture covering

Analysis:
Step 1: The material shows a woven textile structure with visible fibers.
Step 2: The surface has significant texture with a matte appearance and no specular highlights.
Step 3: The physical properties (low density, high friction, low restitution) match a woven textile.
Step 4: Based on these observations and measurements:
- Low stiffness as expected for flexible textiles
- Medium-high Poisson's ratio from the woven structure
- Density matching the measured 300 kg/m^3

Young's modulus: 1.2e8 Pa
Poisson's ratio: 0.38
Density: 300 kg/m^3

Figure 24: **User Prompt II.** The User Prompt we use for every segment of every object.

**User Prompt Continued.**

Example 4:
Material: organic
Opacity: opaque
Density: 400 kg/m^3
Dynamic friction: 0.45
Static friction: 0.65
Restitution: 0.15
Usage: decorative element

Analysis:
Step 1: This is an organic material with the characteristic structure of natural fibers.
Step 2: The surface shows a natural pattern, matte finish, and relatively brittle structure.
Step 3: The physical properties (low density, moderate-high friction, low restitution) align with plant-based materials.
Step 4: Considering similar organic materials and the measured properties:
- Low-medium stiffness in the fiber direction
- Medium Poisson's ratio reflecting the fibrous structure
- Density matching the measured 400 kg/m^3

Young's modulus: 2.5e9 Pa
Poisson's ratio: 0.30
Density: 400 kg/m^3

Based on the provided images and context information, analyze the material properties. Note: The material segment might be internal to the object and not visible from the outside.

Respond using EXACTLY the following format (do not deviate from this structure):

Analysis:
Step 1: Identify the material class/type based on visual appearance
Step 2: Describe the surface characteristics (texture, reflectivity, color)
Step 3: Determine the specific material subtype considering its physical properties
Step 4: Reason through each property estimate based on visual and measured data

Young's modulus: <value in scientific notation> Pa
Poisson's ratio: <single decimal value between 0.0 and 0.5>
Density: <value in scientific notation> kg/m^3

Critical Instructions:
1. You MUST provide numerical estimates for ALL materials, including organic or unusual materials
2. For natural materials like leaves, wood, or paper, provide estimates based on similar materials with known properties
3. Never use "N/A", "unknown", or any non-numeric responses for the material properties
4. For Poisson's ratio, provide a simple decimal number (like 0.3 or 0.42)
5. Each property should be on its own line with exact formatting shown above

Figure 25: **User Prompt III.** The User Prompt we use for every segment of every object.

**Object.**

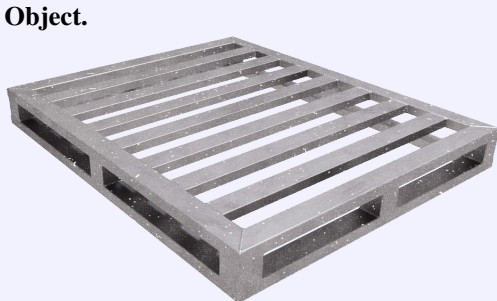

**VLM Response.**

Step 1: Identify the material class/type based on visual appearance
The material appears to be metallic, specifically aluminum, given the context of it being used in a pallet and the provided density value which matches aluminum.

Step 2: Describe the surface characteristics (texture, reflectivity, color)
The surface is smooth with a matte finish, showing minimal reflectivity. The color is a uniform light gray, typical of aluminum.

Step 3: Determine the specific material subtype considering its physical properties
Considering the density of 2700 kg/m³, the material is most likely aluminum, a common choice for pallets due to its lightweight yet durable properties.

Step 4: Reason through each property estimate based on visual and measured data
- **Young's modulus**: Aluminum typically has a Young's modulus around 70 GPa. Given the usage as a pallet, it needs to be stiff but not overly so, supporting the estimate of 70 GPa.
- **Poisson's ratio**: Aluminum generally has a Poisson's ratio around 0.33, which is typical for many metals.
- **Density**: The provided density is 2700 kg/m^3, which is consistent with aluminum.

Young's modulus: 7.0e10 Pa
Poisson's ratio: 0.33
Density: 2.7e3 kg/m^3

Figure 26: **Example Response I.** We demonstrate an example response for a segment from one of the objects from our dataset. The given object has one part.

**Object.**

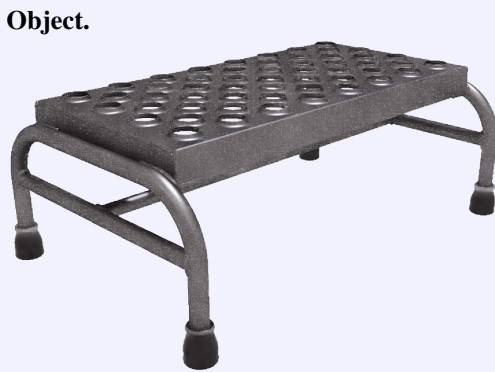

**VLM Response.**

Step 1: Identify the material class/type based on visual appearance
The material appears to be a type of rubber, specifically used for antiskid caps based on the context provided. The surface of the sphere suggests a matte finish, which is common for rubber materials designed for grip and friction.

Step 2: Describe the surface characteristics (texture, reflectivity, color)
The surface of the sphere is matte with a uniform gray color, indicating a lack of significant reflectivity. There is a slight texture visible, suggesting it is not perfectly smooth, which is consistent with rubber materials intended for anti-slip applications.

Step 3: Determine the specific material subtype considering its physical properties
Given the usage as an antiskid cap and the provided physical properties such as density (1100.0 kg/m^3), dynamic and static friction (both 0.7), and restitution (0.8), the material is likely a type of rubber with good elasticity and frictional properties. The density matches well with EPDM rubber, which is commonly used for such applications due to its durability and resistance to environmental factors.

Step 4: Reason through each property estimate based on visual and measured data
- **Young's modulus**: Considering the material is a rubber used for antiskid purposes, it needs to be flexible yet provide sufficient stiffness to maintain shape under load. The given density and usage suggest a medium-hard rubber, so an estimate within the upper range of rubber's Young's modulus is appropriate.
- **Poisson's ratio**: Rubbers typically exhibit high Poisson's ratios close to 0.5, indicating significant lateral expansion when compressed. Given the material's flexibility and the need for good grip, a value near the upper end of the rubber range is suitable.
- **Density**: The provided density is already specified as 1100.0 kg/m^3, which aligns well with EPDM rubber.

Young's modulus: 1.0e8 Pa
Poisson's ratio: 0.49
Density: 1.1e3 kg/m^3

Figure 27: **Example Response II.** We demonstrate an example response for a segment from one of the objects from our dataset. The given object has two parts, and we show the response for the "rubber cap" part.

