# OpenReview forum: "VoMP: Predicting Volumetric Mechanical Property Fields"
_ICLR.cc/2026/Conference — ICLR 2026 Poster_

### Official Review · Reviewer_HQpe · 2025-10-29

**Soundness:** 3
**Presentation:** 3
**Contribution:** 3
**Rating:** 6
**Confidence:** 4

**Summary:**

This paper proposes VoMP, an end-to-end framework that predicts voxel-wise physical property fields from a voxelized 3D representation aggregated from unprojected 2D features. The method employs a MatVAE trained on real material databases to ensure physically plausible outputs, a Geometry Transformer that transforms the input features to predict per-voxel latent codes in the material space, and uses a decoder to reconstruct continuous volumetric material fields from these latents. To train this model, the authors construct a large-scale GVM dataset with voxel-level annotations generated via VLM guidance. Experiments across various 3D inputs demonstrate generalization and physically consistent simulation results.

**Strengths:**

- Proposes a framework for predicting voxel-level material properties from arbitrary 3D representations, defining a novel and important task in 3D physical modeling.
- Introduces a MatVAE that maps material parameters into a physically valid latent space, effectively smoothing predictions and constraining outputs to realistic ranges.
- Employs a VLM to automatically construct large-scale annotated datasets, with the resulting GVM dataset being valuable for future research and other methods.
- Demonstrates higher overall quality than prior methods, with notable improvements in physical consistency and simulation fidelity.

**Weaknesses:**

- Although interior structure plays a crucial role in physical simulation, the current voxelization approach (projecting 2D multi-view features) is not well-suited for capturing internal geometry variations, leading to limited understanding of interior geometry.
- The voxelization process may cause significant information loss in high-frequency or fine geometric regions, reducing the spatial precision of the predicted material fields.
- The transformer design uses only Swin-Attention without the hierarchical downsampling (and upsampling) scheme of a full Swin-Transformer, which restricts the receptive field and may limit the model’s ability to capture long-range structural dependencies. The VAE of TRELLIS is only for local geometry encoding and compression, and using that structure for global material property prediction is not suitable.
- The task and dataset inherently probably contain inconsistencies due to the semantic labeling process based on VLM and the task’s inherent ambiguity. A pure regression formulation may not be ideal for modeling such uncertainty, even with the help of a latent MatVAE. Would a generative or probabilistic approach better capture these ambiguities?

**Questions:**

A bit of concern around L244: Since transformers inherently support variable-length sequences, why is cropping necessary in your implementation? What specific limitation or efficiency concern motivates this design choice? The explanation that “dynamic resampling ensures the model is exposed to different parts of the asset over epochs and has a larger number of effective max voxels” seems counterintuitive. Random subsampling may lead to distribution shift during full-sequence inference, effectively making inference out-of-distribution compared to training. Moreover, if subsampling changes which spatial regions of the object are visible or centered, the geometric semantics of each cropped region could shift, potentially confusing the model. How do you ensure consistency of geometry–material correspondence under such random cropping or resampling?

---

> ### Author Response · Authors · 2025-11-20
>
> We thank you for your highly useful feedback.
>
> **[W1]** We abstractly interpret our inputs as (1) voxels give us coarse information about the object and (2) DINO features give us fine-grained information about the object, including things like small details. During inference, we do not have access to rich internal structure like present during training thus, we build this aggregation scheme that keeps the coarse geometry information from voxels. And for the fine-grained information, the surface voxels receive DINO features for multiple patches across multiple views encompassing the voxel. As we go towards the internal of the object, the voxels receive global-level features about the entire object, which would allow estimating the internal properties.
>
> **[W2]** We thank the reviewer for raising this, and we agree that the final output resolution is limited by voxelization resolution. Similar to the observations made in a recent work, TRELLIS [1] (Section 3.2, subsection “Visual feature aggregation.”), we also find empirically that this resolution is sufficient to reconstruct the original 3D asset at high fidelity especially given we have fine-grained features from DINO. We interpret our inputs as (1) voxels give us coarse information about the object and (2) DINO features give us fine-grained information about the object, including things like small details. Further, we find that with this resolution, we outperform related methods and also find this resolution to be highly-usable for high-fidelity simulations.
>
> We think this is a great idea and as one of the first papers of its kind to do volumetric prediction, like with shape generation [2, 4], we are excited to improve resolution in the future as the field progresses.
>
> **[W3]** We want to note that we don't use hierarchical downsampling however, the DINO features provide semantic information that partially compensates for limited spatial context. The reconstructed features in 3D provide the coarse geometry as well as the global and local-level DINO features (due to our feature aggregation scheme).
>
> **[W4]** We use the MTD (material triplet dataset) to guide the VLM to be more accurate. In section E.1 and Table 7, we constructed a tiny dataset consisting of complex objects that are laboriously manually annotated and compare these properties with the outputs from the VLM in Table 7. We observe that the VLM, given significant additional information (as in our annotation pipeline), performs very well in annotating complex objects and matches human performance. Based on your feedback, we added more discussion about this in Section E.1. We discussed some limitations with getting material property observations from videos (L126-129) and from real-life experimental setups for each object (L118-120).
>
> We agree, it could be possible to condition the model on input geometry instead, we have not tried this formulation. In VoMP, the MatVAE outputs a property and logvar of the property, and further the data annotation pipeline receives the MTD (material triplet dataset) to guide the annotation. This includes ranges of material properties, so the annotation pipeline can use different values for the same material based on the semantics of the object, and other parameters.
>
> **[Q1]** We currently mention that we do this only when the number of available voxels (maximum of 72k per object) is much larger than maximum voxels per object (32k). We have enhanced the clarity of the writing to mention that, subsampling voxels is only performed when the maximum number of voxels per object are exceeded. Using more max voxels per object (i.e., a higher sequence size for the transformer) leads to significantly more GPU VRAM usage. Further, we enhanced the clarity of the writing to mention that we uniformly randomly sampled these voxels to effectively not change the geometry significantly. We expect the model to still have access to the same fine-grained information (DINO features) across the same views of the object.
>
> Finally, we believe that these may not lead to a distribution shift but instead allow us to train a robust model that can handle Gaussian splats and NeRFs captured in the wild, which have some noise. We show this in Figure 8 (a) (the splat was segmented from a larger Gaussian splat scene leading to spurious points), Figure 8 (c) (the splat here was captured by a phone camera), Figure 8 (e), and Figure 9 (row 3).

---

> > ### Author Response · Authors · 2025-11-20
> >
> > ---
> >
> > [1] Xiang, Jianfeng, et al. "Structured 3d latents for scalable and versatile 3d generation." Proceedings of the Computer Vision and Pattern Recognition Conference. 2025.
> >
> > [2] He, Xianglong, et al. "Sparseflex: High-resolution and arbitrary-topology 3d shape modeling." arXiv preprint arXiv:2503.21732. Proceedings of the IEEE/CVF International Conference on Computer Vision (ICCV). 2025.
> >
> > [3] Li, Xuan, et al. "PAC-NeRF: Physics Augmented Continuum Neural Radiance Fields for Geometry-Agnostic System Identification." The Eleventh International Conference on Learning Representations.
> >
> > [4] Sun, Jingxiang, et al. "Dreamcraft3d++: Efficient hierarchical 3d generation with multi-plane reconstruction model." IEEE Transactions on Pattern Analysis and Machine Intelligence (2025).

---

### Official Review · Reviewer_6Gxe · 2025-10-31

**Soundness:** 3
**Presentation:** 3
**Contribution:** 3
**Rating:** 8
**Confidence:** 3

**Summary:**

This paper presents a novel feed-forward model (VoMP) to automatically predict the physical properties for 3D objects, which is an interesting and core problem for setting up the physical simulations requiring these properties. It proposes two important modules, MatVAE and Geometry Transformer, to map the interior voxel to mechanical values in a physically valid manner. The experiments demonstrate the effectiveness and speed of the proposed model.

**Strengths:**

1. The core design is novel and effective. The problem is decoupled into two parts, which guarantee valid physical latent space learning and fast plausible prediction. The ablation study confirms the superiority of the proposed architecture
2. The feedforward nature achieves inference in seconds compared to the previous optimization approach, making it a useful tool for scalable simulation pipeline setup
3. The paper includes a comprehensive set of experiments, including strong quantitative comparisons, ablation studies that justify the design choices.

**Weaknesses:**

1. One remaining concern is the reliance on VLM-generated ground-truth. An expert-annotated dataset is hard to get; hence, the usage of VLMs and external knowledge is an interesting point for data construction. But how to guarantee the correctness of the VLM predicted value is also crucial, especially for real future downstream task usage.
2. The Geometry Transformer is trained using a fixed $64^3$ voxel grid, which is rather low resolution for complex 3D assets. This seems like a major bottleneck that prevents the model from capturing fine-grained material details.

**Questions:**

1. The Geometry Transformer's architecture uses $64^3$ voxel grid, which seems a limitation. What are the bottlenecks preventing scaling this transformer up to $128^3$ or even $256^3$? Would this influence fine-grained detail prediction? It's interesting to analyze whether the performance could also scale up with respect to the resolution.
2. A remaining concern is the material annotation accuracy. Although there is external knowledge to help with the annotation, how can to guarantee the correctness for training and evaluation? Is there any small expert-annotated dataset including these three values that can be used to serve as an independent test of physical accuracy? This problem seems implied in Figure 6, where in (b), VoMP outperforms all previous approaches significantly, while in (c) ABO-500 benchmark, the improvement is not obvious. How to demonstrate that the model is not mimicking the VLM-predicted behavior but truly learning from physics.
3. The model allows internal structure prediction. How to deal with the situations of occlusion?

---

> ### Author Response · Authors · 2025-11-20
>
> We thank you for your highly useful feedback.
>
> **[W1]** We use the MTD (material triplet dataset) to guide the VLM to be more accurate. In section E.1 and Table 7, we constructed a tiny dataset consisting of complex objects that are laboriously manually annotated and compare these properties with the outputs from the VLM in Table 7. We observe that the VLM, given significant additional information (as in our annotation pipeline), performs very well in annotating complex objects and matches human performance. Based on your feedback, we added more discussion about this in Section E.1. We discussed some limitations with getting material property observations from videos (L126-129) and from real-life experimental setups for each object (L118-120).
>
> **[W2][Q1]** We thank the reviewer for raising this, and we agree that the final output resolution is limited by voxelization resolution. Similar to the observations made in a recent work, TRELLIS [1] (Section 3.2, subsection “Visual feature aggregation.”), we also find empirically that this resolution is sufficient to reconstruct the original 3D asset at high fidelity especially given we have fine-grained features from DINO. We interpret our inputs as (1) voxels give us coarse information about the object and (2) DINO features give us fine-grained information about the object, including things like small details. Further, we find that with this resolution, we outperform related methods and also find this resolution to be highly-usable for high-fidelity simulations.
>
> We think this is a great idea and as one of the first papers of its kind to do volumetric prediction, like with shape generation [2, 4], we are excited to improve resolution in the future as the field progresses.
>
> **[Q2 (part 1)]** We completely agree with the reviewer, and we already performed this experiment and had included, however, we placed it in appendix. Please see our response above to **[W1]** or please see section E.1 and Table 7.
>
> **[Q2 (part 2)]** We want to clarify that ABO-500 is an extremely easy benchmark that only partly measures model performance. Even for measuring density, the assets are extremely simple, and the errors are very low as we show in Figure 11 (first 4 rows are random objects and fifth row is some of the worst performing objects). We included ABO-500 for completeness, since related papers use this data. Further, we noticed that the MnRE metric other papers used, and we use too for this experiment for completeness (defined in Section D.1) can hide many errors.
>
> **[Q3]** I interpreted this question as asking how the model can figure out volumetric internal structure when certain parts of the object might be fully occluded. We have enhanced the clarity of the writing to mention that our model operates on an object level. Thus, our method requires captured objects. An object may occlude some parts of itself, as we show in qualitative results like Figure 6 (a), Figure 9, and Figure 10. However, our method is trained with richly annotated volumetric information, and predicts a property per-voxel which allows the model to estimate these occluded parts. The model is trained with ground-truth volumetric assets (Section 5.2).
>
> ---
>
> [1] Xiang, Jianfeng, et al. "Structured 3d latents for scalable and versatile 3d generation." Proceedings of the Computer Vision and Pattern Recognition Conference. 2025.
>
> [2] He, Xianglong, et al. "Sparseflex: High-resolution and arbitrary-topology 3d shape modeling." arXiv preprint arXiv:2503.21732. Proceedings of the IEEE/CVF International Conference on Computer Vision (ICCV). 2025.
>
> [3] Li, Xuan, et al. "PAC-NeRF: Physics Augmented Continuum Neural Radiance Fields for Geometry-Agnostic System Identification." The Eleventh International Conference on Learning Representations.
>
> [4] Sun, Jingxiang, et al. "Dreamcraft3d++: Efficient hierarchical 3d generation with multi-plane reconstruction model." IEEE Transactions on Pattern Analysis and Machine Intelligence (2025).

---

### Official Review · Reviewer_vFJr · 2025-11-04

**Soundness:** 3
**Presentation:** 4
**Contribution:** 3
**Rating:** 6
**Confidence:** 4

**Summary:**

VoMP predicts spatially varying volumetric mechanical properties—Young’s modulus, Poisson’s ratio, and density—from 3D assets, taking any renderable and voxelizable representation as input and outputting per-voxel fields ready for accurate simulation. It aggregates multi-view DINOv2 features into a voxel grid and uses a TRELLIS Geometry Transformer to produce per-voxel material latents. These latents are decoded by a 2D MatVAE trained on real-world triplets, which enforces physically valid material values. A data pipeline combines part segmentations, material databases, textures, and a vision–language model to create large-scale volumetric supervision and a new benchmark. Experiments show lower errors than NeRF2Physics, PUGS, Phys4DGen, and Pixie, while running in a few seconds per object and enabling realistic deformable simulations. Overall, the work delivers a fast, representation-agnostic, feed-forward approach to simulation-ready material fields via multi-view voxel features and a physically grounded latent material prior.

**Strengths:**

- Representation-agnostic, feed-forward pipeline that predicts per-voxel mechanical properties in seconds, with outputs directly usable in simulators.
- Data contribution: a VLM-based pipeline that annotates ~1.6K part-segmented 3D shapes with **volumetric** materials; unlike PIXIE’s Pixelverse (surface-biased), this provides volumetric supervision.
- Extensive quantitative and qualitative results with detailed ablations, plus realistic FEM simulations.
- Clear writing and thorough explanations.

**Weaknesses:**

- Scalability is limited by the size and diversity of the training set (≈1.3K shapes) and the need for part-segmented annotations, making it harder to scale than methods that avoid volumetric labeling.
- Most experiments use author-curated data; because the test set follows a similar distribution, generalization to independent datasets (e.g., Objaverse) is uncertain.
- Resolution is bounded by fixed-grid voxelization, which restricts fine-detail fidelity.

**Questions:**

To what extent does the method capture discontinuities at part boundaries without “bleeding” material across seams, and is there any explicit penalty encouraging jumps where appropriate?

Does the model predict uncertainties per voxel (e.g., variance over E, ν, ρ)

---

> ### Author Response · Authors · 2025-11-20
>
> We thank you for your highly useful feedback.
>
> **[W1]** We agree, collecting more high-quality data can always be helpful. However, there is a severe lack of reliable, useful ground-truth data (we discussed existing data in the first paragraph of Section 2.2). Thus, we believe our dataset is still a large useful, and extremely high-quality dataset that would be helpful to the community in the current form.
>
> **[W2]** This is a great idea. We showed this in Figure 8 (a) (the splat was segmented from a larger Gaussian splat scene leading to spurious points), Figure 8 (c) (the splat here was captured by a phone camera), Figure 8 (e), and Figure 9 (row 3). All of these are OOD, collected separately, and have real-world noise in them. Further, we also tested on an existing benchmark in Figure 6 (c) (L418-424) on the ABO-500 dataset (though this is an easy dataset). We do not compute metrics on other existing due to the severe lack of reliable, useful ground-truth data (we discussed existing data in first paragraph of Section 2.2).
>
> To further enhance the clarity of our work, we have added a large-scale real world result on a Gaussian Splat scene with many individual objects in Section A.1 and Figure 14. We believe this further clarifies that the voxelization method works well on complex real-world data.
>
> **[W3]** We thank the reviewer for raising this, and we agree that the final output resolution is limited by voxelization resolution. Similar to the observations made in a recent work, TRELLIS [1] (Section 3.2, subsection “Visual feature aggregation.”), we also find empirically that this resolution is sufficient to reconstruct the original 3D asset at high fidelity especially given we have fine-grained features from DINO. We interpret our inputs as (1) voxels give us coarse information about the object and (2) DINO features give us fine-grained information about the object, including things like small details. Further, we find that with this resolution, we outperform related methods and also find this resolution to be highly-usable for high-fidelity simulations.
>
> We think this is a great idea and as one of the first papers of its kind to do volumetric prediction, like with shape generation [2, 4], we are excited to improve resolution in the future as the field progresses.
>
> **[Q1]** We do not have a quantitative measurement of sharp discontinuities captured at part boundaries due to the many caveats in constructing a reliable metric for this. The material field is piecewise constant and so can have jumps between voxels which we observe in results like Figure 1. We demonstrate this effect qualitatively in Figure 6 (a) (where we annotate these part boundaries in predictions), Figure 8 (c), Figure 9, and Figure 10. On the second part of the question: No, there is no explicit penalty encouraging jumps, the model learns this from the supervision from richly annotated data.
>
> **[Q2]** Thanks for raising this question. We have further enhanced the clarity of our paper. Yes, that is exactly correct, the MatVAE outputs a property and logvar of the property.
>
> ---
>
> [1] Xiang, Jianfeng, et al. "Structured 3d latents for scalable and versatile 3d generation." Proceedings of the Computer Vision and Pattern Recognition Conference. 2025.
>
> [2] He, Xianglong, et al. "Sparseflex: High-resolution and arbitrary-topology 3d shape modeling." arXiv preprint arXiv:2503.21732. Proceedings of the IEEE/CVF International Conference on Computer Vision (ICCV). 2025.
>
> [3] Li, Xuan, et al. "PAC-NeRF: Physics Augmented Continuum Neural Radiance Fields for Geometry-Agnostic System Identification." The Eleventh International Conference on Learning Representations.
>
> [4] Sun, Jingxiang, et al. "Dreamcraft3d++: Efficient hierarchical 3d generation with multi-plane reconstruction model." IEEE Transactions on Pattern Analysis and Machine Intelligence (2025).

---

### Official Review · Reviewer_2WjH · 2025-11-07

**Soundness:** 3
**Presentation:** 4
**Contribution:** 3
**Rating:** 8
**Confidence:** 3

**Summary:**

The paper proposes VoMP, a feed-forward system that predicts volumetric mechanical property fields including Young's modulus, Poisson's ratio and density- for 3D objects across multiple 3D representations (meshes, Gaussian splats, SDFs, NeRFs). The pipeline first voxelizes the object, lifts multi-view DINO image features into the volume, and feeds per-voxel features to a transformer trained to output material latent codes. MatVAE is a VAE trained on a database of real-world materials so that the decoding process guarantees materials lie on a learned manifold of physically plausible parameters. The authors construct a Geometry with Volumetric Materials (GVM) dataset via a VLM-assisted annotation pipeline that leverages part segmentations, PBR textures, and real-material ranges. VoMP generalizes across 3D formats and produces simulation-ready fields that drive high-fidelity FEM / simplicits simulations without tuning, outperforming recent baselines both in accuracy and speed.

**Strengths:**

1.  New Dataset: The GVM pipeline combines part segmentation + PBR textures + VLM prompts constrained by a curated materials range table to annotate 37M voxels, a scale jump over earlier works that only annotated sparse points. This will be useful to the community and encourage future research in this under-explored area.
2. Volumetric Representation: Unlike previous models that focus heavily on surface properties or struggle with interior prediction (e.g., NeRF2Physics and PUGS due to feature field limitations), VoMP is designed to predict materials throughout the object volume. The system uses a unified multi-view rendering and voxelization pipeline, successfully handling meshes, SDFs, NeRFs, and 3D Gaussian Splats.
3. MatVAE: The introduction of MatVAE- a VAE used to learn a low-dimensional latent space of valid real-world material triplets- is a significant contribution. This component ensures that the decoded per-voxel materials are physically plausible, even during interpolation or sampling, a guarantee largely absent in prior methods that may output simulator-specific or non-physical parameters. The paper explicitly shows that interpolation in the latent space yields valid materials, unlike naive linear interpolation in the material space.

**Weaknesses:**

I do not have a major concern, several minor points:
1. Reliance on Approximate Input and Voxelization Resolution: The final output resolution is limited by the fixed-grid voxelization. This can lead to oversmoothing in highly heterogeneous regions and approximation errors when mapping results back to highly detailed input geometry, especially for thin structures or internal complexities. Also the authors do not provide a comprehensive study on the impact of input & output resolution.
2. Assumption of Isotropic Materials: The annotation pipeline and the resulting model fundamentally assume that materials at the part level are isotropic (properties are uniform in all directions). This is a critical limitation for common composite or natural materials, such as wood or carbon fiber, whose physical behavior depends strongly on direction (anisotropy). The model cannot currently capture this crucial complexity.
3. While the VLM is carefully guided by MTD and visual cues, the core physical property assignment for 37M voxels relies on the VLM's inference capabilities. Failure cases in similar VLM-based approaches (like Phys4DGen's dependency on MLLM consensus) highlight the brittleness of this approach. The use of a VLM introduces a reliance on an external, possibly inaccurate, source of physical knowledge, rather than deriving parameters purely from visual features and learned physics principles. I would like to see more discussions with regard to this.

**Questions:**

1. Generalizability of Voxelization: The feature aggregation step requires solid voxelization, especially challenging for sparse representations like NeRFs and GSs which are fundamentally surface or density fields. Given the extensive training data is derived from high-quality segmented meshes (GVM), how reliable is the voxelization scheme for novel, noisy, or incomplete real-world captured NeRFs/GSs? Does the current method of voxelizing 3D Gaussian splats via rendering depth maps and carving space introduce artifacts or inconsistencies that affect the predicted internal properties in practice?

---

> ### Author Response · Authors · 2025-11-20
>
> We thank you for your highly useful feedback.
>
> **[W1]** We thank the reviewer for raising this, and we agree that the final output resolution is limited by voxelization resolution. Similar to the observations made in a recent work, TRELLIS [1] (Section 3.2, subsection “Visual feature aggregation.”), we also find empirically that this resolution is sufficient to reconstruct the original 3D asset at high fidelity especially given we have fine-grained features from DINO. We interpret our inputs as (1) voxels give us coarse information about the object and (2) DINO features give us fine-grained information about the object, including things like small details. Further, we find that with this resolution, we outperform related methods and also find this resolution to be highly-usable for high-fidelity simulations.
>
> We think this is a great idea and as one of the first papers of its kind to do volumetric prediction, like with shape generation [2, 4], we are excited to improve resolution in the future as the field progresses.
>
> **[W2]** We agree that anisotropy is a limitation of our method, which we discussed in Section 7 of the paper. We think this is a great idea. We do want to clarify that anisotropies are in : (1) spatially varying anisotropic materials and (2) direction of measurement. A huge amount of objects in real world are spatially varying isotropic, which we support. Furthermore, we want to note, that we observe high-fidelity realistic simulations even without anisotropy in our work as we show in Figures 1, 5, 8 and video. We are excited to incorporate anisotropy in some way in the annotation pipeline in the future as the field progresses.
>
> **[W3]** We use the MTD (material triplet dataset) to guide the VLM to be more accurate. In section E.1 and Table 7, we constructed a tiny dataset consisting of complex objects that are laboriously manually annotated and compare these properties with the outputs from the VLM in Table 7. We observe that the VLM, given significant additional information (as in our annotation pipeline), performs very well in annotating complex objects and matches human performance. Based on your feedback, we added more discussion about this in Section E.1. We discussed some limitations with getting material property observations from videos (L126-129) and from real-life experimental setups for each object (L118-120).
>
> **[Q1 (part 1)]** We agree with you, voxelization is challenging for Gaussian Splats and NeRFs. However, we present a new voxelization scheme in the paper (Section 6.1 subsection “Voxelization:”) for Gaussian Splats which partly alleviate these problems. For NeRFs, we set a threshold as the density values (this is referred to as a standard technique in Section 6.1). Further, we find that voxelizing such geometries is not an unusual step in methods [1, 3].
>
> Since our training objects had rich volumetric annotations (Section 5.2) and due to our training process (Section 4.1, 4.2), we observe that though the input geometry might be surface-level as we often see in NeRFs and splats, we are able to reliably estimate the volumetric structure by predicting values on each voxel. We demonstrate this in qualitative results like in Figure 1, Figure 6 (a), Figure 8 (c), Figure 8 (e), and Figure 9 (row 3).
>
> **[Q1 (part 2)]** Thank you for the question, we will further enhance the clarity of our writing to mention this in Section A.1. Any method based on some kind of sampling can always suffer from undersampling. We do not observe any problems for convex objects. In case the objects have some concavity, usually there are enough samples to make it work well. To this extent, in our qualitative results on splats, we find that our voxelization technique works very well even for complex real world objects and noisy splats.
>
> **[Q1 (part 3)]** We answer this partly in **[Q1 (part 1)]** and **[Q1 (part 2)]**. Our training and voxel sampling (Section 5.2) allowed us to train a robust model that can handle Gaussian splats and NeRFs captured in the wild, which have some noise. We show this in Figure 8 (a) (the splat was segmented from a larger Gaussian splat scene leading to spurious points), Figure 8 (c) (the splat here was captured by a phone camera), Figure 8 (e), and Figure 9 (row 3).
>
> To further enhance the clarity of our work, we have added a large-scale real world result on a Gaussian Splat scene with many individual objects in Figure 14. We believe this further clarifies that the voxelization method works well on complex real-world data.

---

> > ### Author Response · Authors · 2025-11-20
> >
> > ---
> >
> > [1] Xiang, Jianfeng, et al. "Structured 3d latents for scalable and versatile 3d generation." Proceedings of the Computer Vision and Pattern Recognition Conference. 2025.
> >
> > [2] He, Xianglong, et al. "Sparseflex: High-resolution and arbitrary-topology 3d shape modeling." arXiv preprint arXiv:2503.21732. Proceedings of the IEEE/CVF International Conference on Computer Vision (ICCV). 2025.
> >
> > [3] Li, Xuan, et al. "PAC-NeRF: Physics Augmented Continuum Neural Radiance Fields for Geometry-Agnostic System Identification." The Eleventh International Conference on Learning Representations.
> >
> > [4] Sun, Jingxiang, et al. "Dreamcraft3d++: Efficient hierarchical 3d generation with multi-plane reconstruction model." IEEE Transactions on Pattern Analysis and Machine Intelligence (2025).

---

### Author Response · Authors · 2025-11-20

We thank reviewers for their insightful feedback. We are encouraged they found the design and task of VoMP to be novel (6Gxe, HQpe), MatVAE to be a novel and useful component (2WjH, vFJr, HQpe) for guaranteeing physical validity of outputs (2WjH, 6Gxe, HQpe, vFJr). The spatially volumetric nature of VoMP to be unlike previous works which predict properties on the surface. (2WjH). They also recognized being representation agnostic (2WjH, vFJr, HQpe) and fast inference (6Gxe, vFJr) as a strength of the method.

Reviewers also recognized our datasets to be a scale jump over earlier works (2WjH), and recognized our datasets and annotation pipeline as useful contributions (2WjH, vFJr, HQpe). Overall, reviewers also state that the datasets would be useful to the community and encourage future research (2WjH, HQpe).

Further the reviewers found the writing to be clear with thorough explanations (vFJr), and found our experiments, ablations, and results to be extensive and complete that justify our choices (6Gxe, vFJr) and demonstrate more realistic simulations (HQpe, vFJr).

Based on the feedback, we have updated our paper/appendix/supplementary to expand upon discussions of some details of the methods (individually indicated to reviewers). We also add a new experiment of running the entire VoMP method (including voxelization) on a large real-life Gaussian splat scene and simulate the scene. This scene, like some of our existing results has significant noise and complex real-world objects.

We respond to each of the reviewers individually.

---

### Meta-Review · Area_Chair_kvdC · 2025-12-31

**Summary:**

Following is a summary of the reviewers' major concerns:
1. Limited voxelization resolution (Reviewer 2WjH, vFJr, 6Gxe, HQpe):
* Limited resolution of the voxelization may limit the spatial resolution of the representation
and risk missing high-frequency details.
* The authors agree with such a limitation, but argue that DINO features will help recover
fine-grained features, and the method has performed well with the current resolution.
2. Rely on VLM-generated annotations (Reviewer 2WjH, vFJr, 6Gxe, HQpe)
* The paper relies on VLM to generate training data, and the reviewers are concerned about
the correctness and consistency of the VLM predictions.
* The authors introduce a manually annotated mini-dataset and show that VLM outputs match human annotations.
3. Whether the method can model internal structures (Reviewer 6Gxe, HQpe)
* The reviewers are concerned about whether the method can predict the properties of
internal structures.
* The authors mention that the method does not have access to the internal structure during inference,
and rely on global-level features of the entire object to estimate internal properties.
4. Evaluation on OOD data (Reviewer vFJr):
* The reviewers are concerned about whether the method can generalize to independent datasets.
* The authors reply that the paper has included several results on the OOD dataset.

**Reviewer Concerns:**

The reviewers sufficiently addressed the concerns from the reviewers (See above).
The reply to the concerns on modeling internal structures is not fully clear to me. But I think
that's not a major issue that will affect the decision on the paper.

**Reviewer Scores:**

Considering that all reviewers are leaning positive before rebuttal, and the authors have sufficiently
addressed the concerns in the rebuttal,  I would expect all reviewers to maintain their positive score
if they were able to participate in the discussion.

---

### Decision · Program_Chairs · 2026-01-26

Accept (Poster)